# Potential Role of Tocotrienols on Non-Communicable Diseases: A Review of Current Evidence

**DOI:** 10.3390/nu12010259

**Published:** 2020-01-19

**Authors:** Sok Kuan Wong, Yusof Kamisah, Norazlina Mohamed, Norliza Muhammad, Norliana Masbah, Nur Azlina Mohd Fahami, Isa Naina Mohamed, Ahmad Nazun Shuid, Qodriyah Mohd Saad, Azman Abdullah, Nur-Vaizura Mohamad, Nurul’ Izzah Ibrahim, Kok-Lun Pang, Yoke Yue Chow, Benjamin Ka Seng Thong, Shaanthana Subramaniam, Chin Yi Chan, Soelaiman Ima-Nirwana, Kok-Yong Chin

**Affiliations:** Department of Pharmacology, Faculty of Medicine, Universiti Kebangsaan Malaysia, Jalan Yaacob Latif, Bandar Tun Razak, Cheras 56000, Kuala Lumpur, Malaysia; jocylnwsk@gmail.com (S.K.W.); kamisah_y@ppukm.ukm.edu.my (Y.K.); azlina@ppukm.ukm.edu.my (N.M.); norliza_ssp@ppukm.ukm.edu.my (N.M.); norliana.masbah@ppukm.ukm.edu.my (N.M.); nurazlinamf@ukm.edu.my (N.A.M.F.); isanaina@ppukm.ukm.edu.my (I.N.M.); anazrun@ukm.edu.my (A.N.S.); qodryrz@ppukm.ukm.edu.my (Q.M.S.); azman.abdullah@ppukm.ukm.edu.my (A.A.); vaizuramohd@gmail.com (N.-V.M.); nurulizzah88@gmail.com (N.I.I.); pangkoklun@gmail.com (K.-L.P.); yokeyue95@gmail.com (Y.Y.C.); benjamin6126@live.com.my (B.K.S.T.); shaanthana_bks@hotmail.com (S.S.); chanchinyi94@gmail.com (C.Y.C.); imasoel@ppukm.ukm.edu.my (S.I.-N.)

**Keywords:** cancer, musculoskeletal disease, metabolic syndrome, osteoporosis, peptic ulcer, vitamin E

## Abstract

Tocotrienol (T3) is a subfamily of vitamin E known for its wide array of medicinal properties. This review aimed to summarize the health benefits of T3, particularly in prevention or treatment of non-communicable diseases (NCDs), including cardiovascular, musculoskeletal, metabolic, gastric, and skin disorders, as well as cancers. Studies showed that T3 could prevent various NCDs, by suppressing 3-hydroxy-3-methylglutaryl-coenzyme A reductase (HMGCR) in the mevalonate pathway, inflammatory response, oxidative stress, and alternating hormones. The efficacy of T3 in preventing/treating these NCDs is similar or greater compared to tocopherol (TF). TF may lower the efficacy of T3 because the efficacy of the combination of TF and T3 was lower than T3 alone in some studies. Data investigating the effects of T3 on osteoporosis, arthritis, and peptic ulcers in human are limited. The positive outcomes of T3 treatment obtained from the preclinical studies warrant further validation from clinical trials.

## 1. Introduction

Non-communicable diseases (NCDs) have become the leading cause of death and the growing threat to global health. They account for 71% (41 million) of the total death cases (57 million) occurred worldwide [1]. Among the NCDs, cardiovascular diseases (31%), cancers (16%), injuries (9%), chronic respiratory diseases (7%) and diabetes (3%) are the major contributors for the global deaths [1]. The escalating prevalence of behavioural and metabolic risk factors (such as overnutrition, physical inactivity, obesity, hypertension, alcohol consumption, and tobacco use) results in the predominance of disease burden by NCDs. The overwhelming inflammatory response and oxidative stress serve as both the precursor and manifestation of NCDs [2]. Alteration of the hormone balance and mevalonate pathway have also been recognized as the underlying mechanism governing the onset and progression of most diseases [3,4]. 

The currently available pharmacological agents used in treating NCDs are not free from side effects, thus necessitating the search for newer agents with better efficacy and lower side effects. Currently, the use of nutraceuticals as adjunct therapy has received considerable interest from researchers for their potential safety, nutrition, and therapeutic actions [5]. Vitamin E is an essential fat-soluble nutrient which can be subdivided into two groups, namely tocopherol (TF) and tocotrienol (T3). They share a structurally similar chromanol ring but differ in the side chain with TF possesses the long saturated (phytyl) tail whereas T3 has short unsaturated (farnesyl) tail [6]. T3 and TF exists naturally as alpha- (α-), beta- (β-), gamma- (γ-), and delta- (δ-) isomers depending on the position of the side chains on the chromanol ring. Both TF and T3 have been regarded as nutraceuticals with a wide range of biological activities against pathological complications, including dyslipidemia [6], cardiovascular [7], musculoskeletal, rheumatoid diseases [8,9,10,11,12,13], gastric [13], neurodegenerative [14], and metabolic diseases [6,15], as well as cancers [16]. The biological effects of T3 can be contributed by its potent anti-inflammatory, anti-oxidative, anti-tumor, cholesterol-lowering [through inhibition of 3-hydroxy-3-methylglutaryl-coenzyme A reductase (HMGCR) activity] activities [8,9,17]. 

This paper aims to provide a comprehensive review of the health-promoting benefits of T3, particularly on its in preventing/treating NCDs. Evidence from in vitro, in vivo, and human studies are summarized in this review to serve as guidance for the application of T3 in clinical setting. It can help to guide to bridge the research gaps currently exist in the investigations of the medicinal properties of T3. 

## 2. Pharmacokinetics of Tocotrienol

T3 is lipid-soluble, thus its absorption is highly dependent of the dietary fat in the body. Consumption of dietary fat ensures sufficient secretion of bile acids and lipase. Bile acids facilitate the emulsification of lipid aggregates to microscopic droplets and provide greater surface area to be digested by lipase and absorbed into the bloodstream. A study by Yap et al. (2001) revealed that the oral bioavailability of αT3, γT3, and δT3 was enhanced when dosed with food among healthy human volunteers. Higher and earlier peak plasma concentrations were reached in fed condition compared to fasted condition. These findings indicated that the consumption of food increased the onset and extent of T3 absorption [18]. Qureshi and colleagues reported that participants supplemented orally with annatto T3 after heavy breakfast achieved peak plasma concentration for all T3 isomers (αT3, βT3, γT3, and δT3) after 3 h of oral administration [19,20]. For parenteral administration, single subcutaneous (s.c.) injection of 300 mg/kg δT3 was given to the CD2F1 mice and the bioavailability of δT3 was assessed. The peak plasma concentration δT3 was achieved after 1 h injection, indicating that T3 absorption was enhanced when administered parentally [21]. 

In the context of distribution, previous in vivo evidence indicated that T3 and its metabolites were detected in most of the organs (including serum, lung, liver, spleen, and colon) in mouse treated with AIN76m diet containing 0.05% γT3 for 2 weeks [22]. Using another mouse model, the distribution of T3 to spleen, kidney, small intestine and colon is relatively high but the level of T3 in liver is relatively low in NCr Nu/Nu mice fed with diet supplemented with 0.1% γT3 and 0.1% δT3 for 6 weeks [22]. Multiple studies showed that T3 was preferably distributed in adipose tissue after supplementation [23,24]. The transport system for vitamin E has been traditionally focused on alpha-tocopherol transport protein (α-TTP), which transports vitamin E from the liver to the blood and has specific affinity towards αTF than other vitamin E isoforms. However, Khanna et al. (2005) found that oral supplementation of αT3 restored fertility and a significant amount of αT3 was successfully transported to vital organs in the α-TTP-deficient mice [25]. In addition, recent study has uncovered the misconception stating that the presence of TF depleted the absorption of T3. Laying hens fed with diet enriched with both αTF and T3 displayed increased uptake and higher distribution of γT3 and δT3 in various organs (including liver, kidney, fat pad, brain, oviduct, yolk, and meat) compared to the animals fed with diet enriched with T3 alone [26]. These studies show that the distribution of T3 can be achieved via mechanisms other than α-TTP. 

For metabolism, it is hypothesized that T3 is metabolized in the similar way as TF [27]. The metabolism processes include hydroxylation and oxidation by cytochrome P450. The major final metabolites are carboxyethyl hydroxychromans (CEHCs) and carboxymethylbutyl hydroxychromans (CMBHCs) [27]. These metabolites are excreted in urine and faeces [22]. Recent literature suggested the health benefits of vitamin E depend on its endogenous long chain metabolites [28]. The apparent elimination half-life of αT3, γT3, and δT3 was found to be short, approximately 4.4, 4.3, and 2.3 h respectively. Thus, a twice daily supplementation was recommended as the frequency of dosing for T3 [18].

Taken together, the oral absorption of T3 depends on bile and lipase secretions, which may be low and erratic. The poor absorption of T3 often results in poor bioavailability and efficacy. This represents a major limitation in developing T3 as a nutraceutical for human consumption because it is difficult to achieve the circulating serum concentration of T3 needed for its effects. Various methods to improve the bioavailability of T3, like using self-emulsifying drug delivery system (SEDDS), have been attempted to achieve faster rate, earlier onset, and higher peak of absorption [29].

## 3. Effects of Tocotrienol on Lipid Metabolism

Hypercholesterolemia is a condition characterized by an elevation of total cholesterol (TC) and/or low-density lipoprotein cholesterol (LDL-C) or non-high-density lipoprotein cholesterol (HDL-C) in the blood [30]. Non-HDL-C is defined as the subtraction of HDL-C from TC [30,31]. Hypercholesterolemia is a contributing factor to CVD. In 2008, the global prevalence of raised hypercholesterolemia among adults (≥5.0 mmol/L) was 39% (37% for males and 40% for females) [32]. The World Health Organization (WHO) reported that hypercholesterolemia is estimated to cause 2.6 million deaths (4.5% of total deaths) and 29.7 million disability adjusted life years (DALYS) (2.0% of total DALYS) [32]. Among the 106,527 participants (aged 35–70 years) of the Malaysia Cohort study, the prevalence of hypercholesterolemia (TC ≥ 240 mg/dL) was 44.9% [33]. By ethnic groups, the prevalence of hypercholesterolemia among Malays, Chinese, Indians, and other races were 51.0%, 40.8%, 41.6%, and 34.4%, respectively [33]. Many studies have stated that hypercholesterolemia is one of the modifiable risk factors to CVD [30,34,35,36]. Every 1.0 mmol/L decrease in LDL-C levels results in a significant reduction in cardiovascular mortality and risk of non-fatal myocardial infarction [37]. According to WHO, a 10% reduction in serum cholesterol in men aged 40 and 70 years has been reported to result in a 50% and 20% reduction in heart disease within 5 years, respectively [32]. Therefore, early screening and effective lipid management may substantially reduce the burden of hypercholesterolemia in the society.

T3 has been shown to possess the potential as a hypocholesterolemic agent [38]. It is known as an inhibitor of the mevalonate pathway responsible for the synthesis of cholesterol and other isoprenoids. The structure of T3 with its three double bonds is similar to farnesyl, the compound preceding the formation of squalene in cholesterol synthesis [39]. T3 promotes the formation of farnesol from farnesyl, thus reducing the formation of squalene [39]. Many in vitro studies examining the mechanism of action of T3 in reducing hypercholesterolemia have been conducted. Triglyceride (TG) synthesis could be lowered by T3 via modulating lipogenic gene expression based on cellular studies [40]. T3 decreased fatty acid synthase (FAS), sterol regulatory element-binding protein (SREBP), stearoyl-coenzyme A desaturase 1 (SCD1), and elevating carnitine palmitoyltransferase 1A (CPT1A) gene expression in mouse hepatocellular carcinoma (Hepa 1-6) cells. However, the cholesterol level remained unchanged [40]. In another study, supplementation of 10–15 µM γT3 suppressed cellular accumulation of TG in human hepatocellular carcinoma (HepG2) cells, especially those cultured in fat-loaded medium [41]. Parker et al. (1993) demonstrated that T3 lowered serum TC by regulating cholesterol production in HepG2 cells by post-transcriptional suppression of HMGCR [39]. Reduction and inhibition of this rate-limiting enzyme were able to reduce cholesterol synthesis in the body [42,43]. In addition, γT3 and δT3 regulated HMGCR in different ways [44]. δT3 enhanced the ubiquitination and degradation of HMGCR, as well as blocking the processing of SREBP. γT3 is more selective in promoting HMGCR degradation than blocking SREBP processing [44]. A docking study showed that inhibition of HMGCR by T3 followed the order δ > γ > β > α [45]. Using HepG2 cells, γT3 was shown to stimulate apolipoprotein B (Apo-B) degradation by decreasing its translocation into the endoplasmic reticulum (ER) lumen. This action eventually caused a reduction in the number of Apo-B in lipoprotein particles [46]. Other reports showed that γT3 and δT3 had the potential to reduce the hepatic TG synthesis and very-low-density lipoprotein (VLDL) secretion by suppressing expression of genes involved in lipid homeostasis, particularly the TG, cholesterol, and VLDL biosynthesis. Moreover, δT3 also promoted the efflux of LDL through LDL receptor expression [47]. A summary of the literature on effect of T3 supplementation on lipid profile of hypercholesterolemic model in vitro is shown in Table 1. 

Animal models of dyslipidemia have been used to investigate the effects of T3 on lipid profile. Several animals have been tested, including chicken, swine and rodents (rats, hamsters, guinea pigs). In chickens fed with a varying level of αTF and γT3, Qureshi et al. (1996) showed that αTF enhanced the inhibition of HMGCR by γT3. They further stipulated that the vitamin E mixture should contain 15–20% of αTF and approximately 60% of γT3 or δT3 for optimal anti-cholesterol effects [48]. In the subsequent study, Qureshi and Peterson (2001) demonstrated that combination of 50 ppm T3-rich fraction (TRF) and 50 ppm lovastatin was more effective in suppressing HMGCR activity compared to lovastatin alone in chickens. The combination also reduced serum TC and LDL-C, TG, Apo-B, thromboxane B_2_, and platelet factor 4 in contrast to individual treatment [49]. Using chicken supplemented with 50 ppm of δT3, Qureshi et al. (2011) further revealed that δT3 reduced TC and LDL-C besides suppressing the lipid elevating effects of dexamethasone and potentiated the TG-lowering effect of riboflavin [50]. Using genetically hypercholesterolemic swine, Qureshi et al. (1991) demonstrated the significant effect of TRF in lowering serum TC, LDL-C, Apo-B, thromboxane B_2_, and platelet factor 4 [51]. In another study involving 6-week treated genetically hypercholesterolemic swine, 50 mg of TRF rice bran, γT3, desmethyl (d-P_21_-T3) or didesmethyl (d-P_25_-T3) T3 individually reduced serum TC, LDL-C, Apo-B, thromboxane B_2_, platelet factor 4, glucose, and glucagon [52]. After termination of the treatment, the cholesterol-lowering effects persisted for 10 weeks [52]. However, the effects of TRF on normal swine were limited to reduction of TC and Apo-B. The activity of HMGCR in adipose tissue was reduced with treatment in both normal and hypercholesterolemic swine [52]. The hypocholesterolemic efficacy of vitamin E isomers was also compared. It was found that δT3 and γT3 were the most effective isomers to lower the TC and LDL-C levels, followed by TRF and αT3. However, αTF did not show lipid-lowering effect in the same study [53]. On the contrary, Hansen et al. (2015) reported that in laying hens supplemented with T3 or T3 and αTF, none of the treatment causes any significant change in TC, HDL-C, and TG levels [26]. 

Rodent models have been widely utilized to demonstrate the effect of T3 in preventing diet or chemical-induced hyperlipidemia. Isomers of αT3 and γT3 exhibited potential to act as anti-hypercholesterolemic and anti-oxidative agents in rats fed with atherogenic diets [54]. Using rats supplemented with 0.1% of αT3, T3 mix or quercetin for 3 weeks, Kaku et al. (1999) reported that TG was significantly reduced in the T3 group and quercetin group. All treatments did not cause any changes in TC and phospholipids [55]. In a similar experiment, TRF supplementation was able to reduce the lipid parameters in a dose-dependent manner with the optimum effect at the dose of 8 mg/kg/day in rats [56]. Using hypercholesterolemic rats supplemented with TRF, Cheng et al. (2017) reported that TRF had the potential to reduce TC and non-HDL-C level but not the TG level as it remained high even after the TRF treatment. Thus, TRF was more specific to cholesterol metabolism compared to TG [57]. In another study involving F344 rats on high fat diet supplemented with 5 or 10 mg/kg rice bran T3 per day for three weeks, the intervention caused a reduction in TG and phospholipid hyper peroxidases (oxidative stress marker) in the liver and plasma. However, T3 did not show any change in cholesterol level [41]. Zaiden et al. (2010) noted that treatment with γδT3 at 50 mg/kg for one month significantly reduced cholesterol and TG level in the LDLr-deficient mice [47]. In a diabetic-induced rat model, TRF supplementation for 8 weeks significantly reduced lipid profile parameters (plasma TC, LDL-C, and TG) and increased HDL-C. TRF also elevated plasma superoxide dismutase (SOD) level, reduced malondialdehyde (MDA) and 4-hydroxynonenal (4-HNE) (in plasma and aorta), oxidative damage, deoxyribonucleic acid (DNA) damage and protected the vessel wall [58]. 

Chemically-induced hypercholesterolemic animal models demonstrated that supplementation of rice bran TRF (10 mg/kg/day) effectively suppressed the elevation of plasma total and LDL-C in rats administered with 7,12-dimethylbenz[alpha]anthracene (DMBA) [59]. Moreover, it also suppressed the activity and protein mass of hepatic HMGCR [59]. In a study using Syrian hamsters with inflammation-induced hypercholesterolemia, Tocomin (a palm T3 mixture) reduced the level of plasma and lipoprotein lipids, cholesterol, Apo-B, small dense-LDL-C and LDL-C level [45]. Khor et al. (1995) demonstrated the dose-dependent effect of T3 in inhibiting HMGCR activity in guinea pig [60]. Khor and Chieng (1997) found that T3 (162 ppm) lowered the serum TC but not LDL-C, HDL-C, and TG while αT3 (72 ppm) increased TC, LDL-C, and HDL-C levels. They also reported that with the presence of 0.1% squalene in the diet, both T3 and TF further reduced the level of total and LDL-C but not HDL-C and TG [61]. In another experiment, γT3 and T3 mixture were compared to study their effects on hypercholesterolemic hamsters. After 2 weeks of treatment with T3 (263 mg/kg) and γT3 (58 and 263 mg/kg), decreased plasma TC and LDL-C were observed, but both TG and HDL-C levels were not affected [62]. The authors reported that γT3 showed a greater hypocholesterolemic effect compared to T3 mixture [62]. The effect of αTF on serum lipids profile in hamsters was also studied by Khor and Ng (2000) [63]. They showed that lower dose of αTF (30 ppm) inhibited HMGCR activity while a higher dose of αTF (81 ppm) stimulated HMGCR activity. The authors also have tested the effect of T3 in guinea pigs and they found that intraperitoneal (i.p.) injection of 10 mg T3 inhibited HMGCR activity while a mixture of 5 mg αTF with 10 mg T3 resulted less inhibition [63]. Using guinea pigs injected with palm olein triglycerides (T3 or αTF), Khor et al. (2002) found that T3 (1–5 mg) reduced the activity of liver microsomal HMGCR and cholesterol 7α-hydroxylase. The inhibitory effect of αTF was less effective and at 5 mg of αTF, the inhibitory effect was ineffective. However, at higher dose (50 mg), αTF significantly increased the activity of HMGCR and reduced the inhibitory activity of T3 on HMGCR [64]. A summary of the literature on effect of T3 supplementation on lipid profile of hypercholesterolemic chicken, swine, and rodents is shown in Table 2.

The effects of T3 supplementation on lipid profile in humans were heterogeneous, probably due to the differences in the composition of the T3 mixture used, population studied and diet of the subjects. An earlier study by Qureshi et al. (1991) showed that treatment using palm vitamin E (PVE) at 200 mg/day was able to reduce serum TC, LDL-C, Apo-B, thromboxane B_2_, platelet factor 4, and glucose in hypercholesterolemic subjects during the 4-week treatment [65]. Few years later, Qureshi et al. (1995) supplemented hypercholesterolemic subjects with PVE (a blend of vitamin E providing 40 mg αTF, 48 mg αT3, 112 mg γT3, 60 mg δT3) and 200 mg γT3 after acclimation to the American Heart Association (AHA) Step-1 dietary regimen for 4 and 8 weeks, respectively [66]. The authors observed a decrease in TC level in both groups. Besides, plasma Apo-B and ex vivo generation of thromboxane B_2_ also reduced following T3 treatment but neither preparation had impacts on HDL-C, and apolipoprotein A-1 (ApoA-1) levels [66]. In 2001, Qureshi et al. demonstrated the synergistic effect of combination TRF of rice bran and lovastatin with AHA Step-1 diet on lipid parameters (reduced TC and LDL-C) in hypercholesterolemic subjects after treatment for 25 weeks [67]. Another study in 2002 involved 90 hypercholesterolemic subjects subjected to 25, 50, 100, and 200 mg/kg TRF while placed on AHA Step-1 diet. TRF at 100 mg/kg produced maximum decrease in TC, LDL-C, Apo-B, and TG compared to baseline [68]. In another study, Zaiden et al. (2010) supplemented hypercholesterolemic subjects with γT3 and δT3 (120 mg/day) for 8 weeks and they observed that the treatment significantly reduced the serum TG, VLDL and chylomicrons in the subjects. However, TC, LDL-C, and HDL-C remained unchanged in treated and placebo groups [47]. Yuen et al. (2011) reported that consumption of 300 mg mixed T3 (containing 22.9 IU αTF, 30.8% αT3, 56.4% γT3 and 12.8% δT3) for 6 months were able to reduce serum TC and LDL-C in patients with hypercholesterolemia [69]. In another study, subjects with mild hypercholesterolemia with one additional cardiovascular risk factor were randomized to receive T3 (n = 28) or αTF (n = 16) (500 mg/day) for 4 weeks [70]. T3 group experienced a decline in TC and LDL-C but the αTF group did not experience such changes. However, this was not a randomized control trial and no placebo group was assigned [70]. In a study by Heng et al. (2015), adults with MetS aged 20–60 years old received 400 mg/day of mixed T3 (61.52 mg αT3, 112.80 mg γT3 and 25.68 mg δT3 and 61.1 mg αTF) for 16 weeks [71]. The supplementation reduced TC, LDL-C and HDL-C. However, due to the reduction of HDL-C and TC/HDL-C ratio was increased [71]. Baliarsingh et al. (2005) supplemented type 2 diabetic subjects who have hyperlipidemia with TRF (3 mg/kg) for 60 days and the authors found that TRF significantly reduced the level of total lipids, TC, and LDL-C [72]. Daud et al. (2013) demonstrated that treatment with TRF (containing 20 mg αTF, 30.18 mg αT3, 5.30 mg βT3, 41.66 mg γT3 and 12.86 mg δT3) for 16 weeks decreased plasma TG level and increased HDL-C in chronic hemodialysis patients [73]. Besides, the lipid-lowering effects of T3 were evaluated in healthy individuals. Tan et al. (1991) revealed that treatment of PVE (containing 18 mg TF, 42 mg T3, and 240 mg palm olein) for 30 days lowered serum TC and LDL-C but TG and HDL-C level were no affected in healthy volunteers [74]. Chin et al. (2011) carried out a study among young (35–49 years old) and old (>50 years old) healthy subjects [75]. The subjects received either TRF (160 mg/day, 74% T3 and 26% TF) or placebo capsules for 6 months. Blood samples were collected at baseline, third month and sixth month. The elderly subjects responded better by showing an increase in HDL-C compared to placebo group [75]. Heng et al. (2013) also showed that TRF (150 mg/day) affected the expression of ApoA-1 precursor, apolipoprotein E precursor and C-reactive protein (CRP) precursor among young and old individuals [76]. 

Tomeo et al. (1995) supplemented patients with cerebrovascular diseases with 300 mg TRF containing 16 mg αTF, 40 mg γT3 and αT3 and 240 mg palm olein for 18 months but no changes in TC, LDL-C, TG, and HDL-C were observed as compared to the placebo group [77]. In men with mildly elevated lipid concentration, supplementation of T3 (4 capsules daily for 6 weeks, each containing 35 mg T3 and 20 mg αTF) showed no effects on serum LDL-C, HDL-C, TG, lipoprotein a, lipid peroxide, platelet aggregation velocity, maximum aggregation, and thromboxane B_2_ formation [78]. In the study by O’Byrne et al. (2000), subjects followed low-fat diet for 4 weeks, then they were randomized and took placebo, αT3, γT3, δT3 acetate supplements (250 mg/day) for 8 weeks while still on the low-fat diet [79]. The study discovered that αT3 increased in vitro LDL oxidative resistance and decreased its rate of oxidation. Yet, serum TC, LDL-C, and Apo-B levels were not affected [79]. A study by Mustad et al. (2002) revealed no significant changes in lipid or glucose concentration after treatment of 200 mg T3 (mixed αT3, γT3, δT3, pure δT3 p25 complex) 28 days in hypercholesteremic men and women [80]. Rasool et al. (2006) noted that supplementation of T3 rich vitamin E at 80, 160, 320 mg/day for 2 months among healthy men did not affect arterial compliance, plasma total antioxidant status, serum TC, or LDL-C levels as compared to placebo group [81]. A summary of the literature on effect of T3 supplementation on lipid profile in human subjects is shown in Table 3. The mechanism of T3 in improving lipid profile is summarized in Figure 1. 

Overall, T3 showed is a potential hypocholesterolemic agent as evidenced by in vitro, in vivo, and human studies. The hypocholesterolemic properties of T3 seen in the animal and human studies appear to be promising. It may be recommended to be taken as a supplement by hypercholesterolemic patients to manage their cholesterol level.

## 4. Effects of Tocotrienol on Cardiovascular Diseases

CVD, a general term for conditions affecting the heart or blood vessels, is the number one cause of death globally. It includes ischemic heart disease and stroke. Hypertension is a major reversible risk factor for CVD [82]. WHO reported 17.9 million CVD-associated deaths in 2016, contributing to 31% of global mortality [83]. The pathogenesis of CVD is multifactorial, which involves a significant primary role of oxidative stress and inflammation [84,85,86,87]. Persistent increase in blood pressure (BP) will damage the blood endothelial lining [88], leading to damage of the end organs like the heart and kidneys [89,90]. On the other hand, myocardial ischemia-reperfusion injury is associated with increased oxidative stress due to massive build-up of free radicals during ischemia and reperfusion [91]. Another risk factor associated with CVD and stroke is hyperhomocysteinemia [92]. This condition may lead to endothelial cell damage and altered flexibility of vessels [92] by reducing the availability of plasma nitric oxide (NO) and promoting pro-atherosclerotic inflammatory mediators like vascular adhesion molecules as well as oxidative stress in aorta [93] and heart [94]. Additionally, hypercholesterolemia is also one of the risk factors for CVD. It renders the heart to increased oxidative stress, mitochondrial dysfunction, and inflammation triggered apoptosis. All these changes increase the susceptibility of the heart to infarction [95]. 

Several animal models are used to study the effects of T3 in various CVD (Table 4). A study by Das et al. (2012) showed that T3 isomers (αT3, γT3, and δT3) protected hearts against myocardial ischemia-reperfusion injury in rats [96]. The cardioprotection of the T3 is attributable to reduced expressions of p38 mitogen-activated protein kinase (MAPK)β, heme oxygenase-1 (HO-1), endothelial nitric oxide synthase (eNOS) with calveolin. These proteins are known as pro-survival signaling components. Besides, T3 also elevated the expression of p38 MAPKα and SRc kinase, which are anti-survival signaling components, with calveolin. Known as a cholesterol-binding protein, calveolin serves as a platform in the signal transduction process. Therefore, cardioprotection by the T3 isomers could be achieved by modifying the availability of pro-survival and anti-survival proteins in the heart. In another study by Das et al. (2008), both αT3 and γT3 increased the resistance of hypercholesterolemic to ischemic reperfusion injury in male and female adult New Zealand rabbits, tentatively by reducing several hypercholesterolemic proteins like MMP-2 and MMP-9, endothelin-1 (ET-1), and Spot 14, and upregulating transforming growth factor (TGF)-β [97]. In rats infused with xanthine/xanthine oxidase (XO) to mimic myocardial ischemia-reperfusion injury, T3 decreased the release of creatine phosphokinase, a measure for heart muscle injury into bloodstream and myocardial oxidative stress [98]. 

T3 has been demonstrated to reduce the plasma homocysteine level in a different model of rats fed with high-methionine diet. The authors postulated that it might exert a direct inhibitory effect on hepatic cystathionine β-synthase [99], which is the rate-limiting enzyme for methionine transsulfuration pathway [100]. A previous study showed that T3 treatment restored the plasma NO, as well as inhibiting the rise in oxidative stress and inflammation in rats with hyperhomocysteinemia [93,94]. These protective effects are not shared with TF [101]. TRF, in particular its γ- and δ-isomers, reduced hypertension in rats with diet-induced metabolic syndrome (MetS) [57,102,103]. It improved vascular endothelial function by promoting vascular relaxation [104] and eNOS activity that enhances NO production, a potent vasodilator [105] in spontaneously hypertensive rats (SHRs), leading to a decrease in systolic BP. 

However, no human study on the effects of T3 on CVD has been performed, except in patients with hypercholesterolemia, which has been elaborated in the previous section. The effect of T3 on BP in terms of arterial compliance was not observed in normotensive human subjects [81] as well as in normotensive non-proteinuric primgravidae with singleton pregnancies. A previous study showed that T3 did not significantly reduce the risk of development of pregnancy-induced hypertension in these subjects [106]. Another study by Mustad et al. (2002) indicated no beneficial effect on key CVD risk factors in normotensive healthy subjects [80]. On the other hand, when a self-emulsifying preparation of T3-rich vitamin E, with enhanced bioavailability, at the doses of 100 and 200 mg daily for two months was administered, reductions in augmentation index and pulse wave velocity, indicative of systemic arterial stiffness were reported in healthy men [107]. The effects of T3 on CVD in human studies are summarised in Table 5.

Overall, T3 has been shown to be protective against CVD in experimental animals [102]. However, evidence of its effects in humans is limited and inconsistent [81,107], probably due to the difference in its preparation that may affect its bioavailability. Therefore, the clinical effects of T3 on CVD could not be extensively evaluated. Further studies are required to strengthen these findings. The evidence so far serves as a basis for clinical trials to validate the beneficial effects of T3, especially in the patients at risk for CVD. The cardioprotective effects of T3 are summarized in Figure 2. 

## 5. Effects of Tocotrienol on Cerebrovascular Diseases

Cerebrovascular disease refers to a group of medical conditions or disorders that affect the blood vessel of the brain and cerebral circulation. Stroke, transient ischemic attack, aneurysm, and vascular malformation are the common types of cerebrovascular disease. Hypertension is also the major contributing risk factor for cerebrovascular diseases [108].

T3 are shown to be protective against ischemic stroke by attenuating brain lesion volume in many animal models such as dogs [109] and mice [110] (Table 6). αT3 is the most widely studied isomer in animal stroke models induced by middle cerebral artery occlusion (MCAO) [110,111]. Several isoforms of TF or T3 have been shown to prevent cerebral infarction induced by MCAO [112]. The authors postulated that the protective effects of vitamin E isomers on cerebrovascular diseases might be due to their neuroprotective properties [112]. T3 in sub-attomole protects neurons from glutamate challenge, an effect not seen with αTF [113]. Oral supplementation of αT3 (50 mg/kg) for 10 weeks prevented the loss of miR-29b expression at the infarct site, leading to reduction in hemisphere lesion volume percentage in mice with acute ischemic stroke [110]. miR-29b acts as an anti-apoptosis factor by silencing pro-apoptotic genes of BH3 family and neuronal survival factor [114]. The latter prevents neurodegeneration following stroke. Thus, it is proposed that T3 exerts its effect by elevating multidrug resistance protein 1 (MRP1) that mediates the efflux of oxidized glutathione (GSSG) from neural cells [111]. The increased intracellular level of GSSG indicates increased oxidative stress due to its conversion from its reduced glutathione, an antioxidant, which will lead to neural cells death [115]. Tocotrienol enriched supplement also promoted arteriogenesis in brain by inducing metalloprotease 1 (an arteriogenic tissue inhibitor) which subsequently attenuates the activity of matrix metalloproteinase (MMP)-2 [109]. This finding was suggestive of the ability of the T3 to induce adaptive arteriogenesis in response to focal cerebral ischemia. In another mouse stroke model, supplementation with Tocovid (200 mg/kg/day), a palm tocotrienol mixture, for a month exerted anti-inflammatory effects in ischemic mice brain by lowering the expressions of inflammatory biomarkers including tumor necrosis factor-alpha (TNF-α), monocyte chemotactic protein-1 (MCP-1) and ionized calcium-binding adapter molecule-1 (Iba-1), as well as antiapoptotic or autophagic parameters which are cleaved caspase-3 and LC3-II expressions. These changes are believed to contribute to a reduction in brain infarct volume [116,117]. 

The relation between T3 and cerebrovascular disease has been investigated in a human study. TRF at the dose of 200 mg twice daily for two years was reported to be beneficial in attenuating the progression of brain white matter lesion in volunteers with cardiovascular risk factors and MRI-confirmed white matter lesion. This lesion is regarded as manifestations of cerebral small vessel disease [118]. In a study cohort consisting of 50 patients with cerebrovascular disease, apparent carotid atherosclerotic regression was found in the TRF-supplemented subjects while some controls showed progression [77]. Collectively, it has been reported that T3 exerts potential protective effects on cerebrovascular disorders (Figure 3). 

## 6. Effects of Tocotrienol on Osteoporosis

Osteoporosis is a medical condition characterized by reduced bone mass and skeletal microstructural deterioration resulting in bone fragility and susceptibility to non-traumatic fracture. According to Oden et al. (2015), a total of 21 million men and 137 million women worldwide were at high risk of developing osteoporotic fracture in the year 2010. The number of high-risk individuals was projected to increase two-fold over the next 40 years [119]. In Malaysia, the projected incidence and direct cost of hip fracture by 2050 will be 3.55- and 3.53-fold increase respectively as compared to 2018 [120]. Since osteoporotic fractures impose a huge social and economic burden, enormous effort has been undertaken to manage this problem, particularly the development of viable preventive and therapeutic approaches for osteoporosis. Advanced age [121], sex hormone deficiency [122], the use of medication [123], alcohol consumption [124], smoking [125], as well as underlying medical complications such as hypertension [126], diabetes [127], hyperlipidemia [128] and MetS [129,130,131] are the common contributing factors for poor bone health. 

The importance of vitamin E in the maintenance of bone health was illustrated by models of vitamin E deficiency and supplementation using growing (3- to 4-month old) female rats [132,133]. It was found that vitamin E deficiency (VED) diet caused hypocalcemia in the first month, elevated parathyroid hormone (PTH) level in the second month and reduced bone calcium content in L4 vertebra in the third month in female rats [132]. Another study demonstrated similar outcomes whereby female rats fed with VED diet for 8 months had impaired bone calcification evidenced by the reduction of bone calcium content in left femur and L5 vertebra as compared to the animals fed standard rat chow [133]. The supplementation of 60 mg/kg αT3 for 3 months did not improve the changes (hypocalcemia, elevated PTH and low bone calcium content) caused by vitamin E deficiency in female rats [132]. However, supplementation of PVE for 8 months increased bone calcium content of the fifth lumbar and left femur in vitamin E-deficient rats [133].

Ovariectomized rats are the most common animal model of female osteoporosis in investigating the effects of T3 on bone loss. Ovariectomy refers to the removal of ovaries, which induces a state of oestrogen deficiency mimicking menopause and the subsequently bone loss in women. Ovariectomized female rats experience increased bone resorption, deterioration of bone microstructure and reduced bone strength as early as eight weeks after ovariectomy. Low bone mineral density (BMD) was only detected after 10 months post-surgery [134]. The effects of T3 on bone loss have been widely investigated using ovariectomized female rats (Table 7). Palm-derived T3, annatto-derived T3, pure isomers of T3 and TRF were used in previous studies. The doses of T3 tested were 30, 60, or 100 mg/kg body weight and the routes of administration were oral gavage or injection. Overall, T3 was effective in preventing bone loss in osteoporotic female model induced by ovariectomy. Most of the studies showed promising effects of T3 on bone with an optimum dose of 60 mg/kg. Palm T3 (60 mg/kg) has been demonstrated to improve bone static, dynamic, and structural histomorphometry, seen by increased double-labeled surface (dLS), mineralizing surface (MS), mineral apposition rate (MAR), bone formation rate (BFR), bone volume/total volume (BV/TV) and trabecular number (Tb.N) as well as reduced single-labeled surface (sLS), osteoclast surface (Oc.S) and trabecular separation (Tb.Sp) in oestrogen-deficient rats [135,136]. A study by Norazlina et al. (2000) showed that oral administration of 30 or 60 mg/kg PVE maintained BMD, but only 60 mg/kg PVE maintained bone calcium content in ovariectomized rats [134]. In addition, oral administration of 60 mg/kg annatto-derived T3 was found to improved bone dynamic indices (increased dLS, MS, MAR, and BFR; and reduced sLS) in female rats with oestrogen deficiency [137]. So far, only pure γT3 and δT3 were examined for their bone-protective properties using ovariectomized female rats. Injection of γT3 (100 mg/kg, s.c.) exerted protective effects against ovariectomy-induced bone loss by increasing osteoblast number (Ob.N), MAR, BFR, BMD, BV/TV, trabecular thickness (Tb.Th) and osteocalcin (OCN), as well as reducing osteoclast number (Oc.N), Tb.Sp and C-telopeptide of type I collagen (CTX-1) [138]. In other studies, the ovariectomized female rats treated with 60 mg/kg δT3 displayed relatively higher osteoblast surface (Ob.S), osteoid surface (OS), osteoid volume (OV), BV/TV, Tb.N, Tb.Th, and OCN as well as lower Oc.S, eroded surface (ES), Tb.Sp, and CTX-1 than the untreated ovariectomized rats. These changes subsequently increased femoral load, stress, strain and Young’s modulus in the T3-treated ovariectomized rats compared to the negative controls [139,140]. On the other hand, TRF and T3-enriched fraction (source of T3 not mentioned) at the dose of 60 mg/kg were able to improve bone dynamic parameters (increased dLS, MAR, BFR, and reduced sLS) and bone strength (increased stress) respectively [141,142]. 

Osteoporosis ultimately leads to fragility fracture, which causes significant morbidity and mortality worldwide. Using the ovariectomized female rat model with fracture at the metaphysis region of right tibia, Ibrahim and colleagues demonstrated that single injection of annatto T3 particles (60 mg/kg) improved only the strength (load and stress) of callus. However, the combination of annatto T3 and lovastatin particles synergistically conferred better fracture healing of osteoporotic bone by improving the mineralization (BV/TV and BMD) and strength (load and stress) of callus [143]. 

The effects of T3 on bone have also been extensively studied using male rats (Table 8). An early study showed that γT3 could improve bone structure, increased Ob.N and OS, reduced Oc.N and ES, as well as enhancing mineral deposition in normal male rats [144]. This study suggested that T3 might exhibit protective effect on male skeleton. In subsequent studies using male osteoporosis models, rats induced with osteoporosis (testosterone deficiency, nicotine, MetS, and ferric nitrilotriacetate models) were given palm T3, PVE, annatto T3, and pure isomers of T3 (γT3) at 30 mg/kg, 60 mg/kg and 100 mg/kg. Palm T3 and γT3 at 60 mg/kg were proven effective in preserving the trabecular bone structure by improving the BV/TV and Tb.Th in bone loss models induced by nicotine [145] and ferric nitrilotriacetate [146]. Supplementation of palm and annatto T3 at 60 mg/kg also exerted positive anabolic effects on bone structure, such as increasing BV/TV, Tb.N and decreasing Tb.Sp in rats with osteoporosis induced by testosterone deficiency [147] and MetS [148,149,150].

The improvement of bone structure is a result of enhanced bone mineralization process and bone cell dynamic. These can be visualized by bone dynamic and cellular parameters. Significant changes in bone dynamic parameters, indicated by increased sLS and decreased dLS, were observed in normal rats treated with γT3 for 4 months [144]. However, a diet mixed with αTF and T3 (250 mg/kg diet) produced negligible effects on bone microstructure in normal rats [151]. γT3 supplemented significantly increased Os.N, OS, OV, MAR, BFR, and decreased Oc.N and ES in normal rats [144]. γT3 was able to reverse the damaging effects of nicotine on bone structure by improving OS, MAR, and BFR while reducing Oc.S, ES significantly [145]. Annatto T3 was shown to increase Ob.N, OS, OV, and decreased Oc.N and ES in orchidectomized rats [152]. 

All of the skeletal histological changes aforementioned can lead to gross physical and mechanical changes in the bone. PVE at 30 mg/kg and 60 mg/kg supplementation successfully improved BMD in orchidectomized [153] and dexamethasone-treated rats [154] It also restored the bone calcium content and increased the femoral length of the rats in dexamethasone model [155]. T3 at 60 mg/kg also reversed the negative effect of nicotine on femoral calcium content in rats [156]. Supplementation with γT3 for 4 months significantly improved the biomechanical strength of the femur in normal rats [157]. However, in testosterone deficiency model, supplementation of annatto T3 at 60 mg/kg for eight weeks was unable to improve bone biomechanical strength of the rats, possibly due to short duration of treatment [158]. Wong et al. (2018) showed that rats supplemented with oral T3 for 12 weeks significantly improved biomechanical strength of the femur in MetS-induced osteoporotic rats [148]. Despite the abundance of positive preclinical evidence, it is uncertain that these skeletal beneficial effects will be shown in men because there is no study on the effects of T3 on bone metabolism in men to date. 

T3-mediated bone protection may involve its antioxidant, anti-inflammatory, mevalonate suppressive and bone-related gene-modulating properties. Nazrun et al. (2008) showed that palm T3 (60 mg/kg) increased glutathione peroxidase (GPx) and superoxide dismutase (SOD) activities but suppressed MDA level in the ovariectomized rats [159]. Supplementation of palm T3 orally at 100 mg/kg significantly reduced the products of lipid peroxidation of thiobarbituric acid-reactive substance (TBARS) and increased GPx activity in normal rats [160]. Pro-inflammatory cytokines play a significant role in bone resorption. T3 supplementation reversed elevation of IL-1 and IL-6 induced by oestrogen deficiency [161], nicotine [162], ferric nitrilotriacetate [146], and MetS [148,149,150]. Reduction of these cytokines in the circulation could suppress osteoclastogenesis and bone resorption. Recently, study conducted by Wong et al. (2019) demonstrated that treatment with annatto or palm T3 modulated osteocyte-related bone peptides (characterized by lowering of RANKL, sclerostin (SOST), Dickkopf-related protein (DKK)-1 and fibroblast growth factor (FGF)-23) in a rat model of bone loss induced by MetS [163]. Bone resorption markers, such as pyridinoline and deoxypyridinoline (DPD), were also reduced in nicotine [164] and ferric nitrilotriacetate-treated rats [146] concurrently. 

T3 may also affect the formation and activities of bone cells at gene levels. Utilizing nicotine-induced osteoporosis rat model, Abukhadir et al. (2012) reported that PVE increased gene expression related to bone formation such as BMP-2, a growth factors known to induce bone and cartilage formation, as well as Runt-related transcription factor *2* (Runx-2) and osterix (OSX), which play key roles in osteoblast differentiation [165]. In the testosterone deficiency model, annatto T3 enhanced the expression of gene related to bone formation and osteoblast activity, such as alkaline phosphatase (ALP), β-catenin, and collagen type I alpha 1 (COL1α1) [147]. Treatment with T3 upregulated the expression of bone morphogenetic protein (BMP)-2, osteoprotegerin (OPG), Runx-2, OSX, vascular endothelial growth factor (VEGF)-α, and downregulated receptor activator of nuclear factor kappa-B ligand (RANKL) expression in the ovariectomized animals [137,138] and ovariectomized animals with fracture [166]. A recent in vitro study further supported that annatto T3 enhanced osteoblastic differentiation-related genes such as OSX, COL1α1, ALP, and OCN in murine MC3T3-E1 pre-osteoblastic cells [167]. These effects translated to a positive influence of T3 on bone formation. In a study by Norazlina et al. (2010), supplementation of γT3 at 60 mg/kg caused an increase in OCN (a bone formation marker) in nicotine-treated rats [164]. However, most other studies revealed insignificant changes in the circulating bone formation (OS, ALP, and P1NP) and resorption markers (TRACP 5b) [152,153,162]. It was postulated that the changes in bone turnover markers occur early during osteoporosis induction, thus they are not detectable at the end of the studies. So far, only the study by Deng et al. (2014) showed that γT3 might act as HMGCR inhibitor to protect ovariectomized rats against bone loss. In their study, daily supplementation of mevalonate reversed the bone-protecting effects of T3 in the ovariectomized rats [138]. 

Recently, a double-blinded placebo-controlled trial involving 89 postmenopausal women with osteopenia was conducted to investigate the effects of T3 on bone [168]. Supplementation of annatto T3 (430 or 860 mg/day) for 12 weeks decreased bone marker levels (serum bone-specific alkaline phosphatase (BALP), urine N-terminal telopeptide (NTX), serum soluble RANKL and RANKL/OPG ratio) and oxidative stress biomarker (urinary 8-hydroxy-2′-deoxyguanosine (8-OHdG)) in postmenopausal women with osteopenia relative to the placebo group. These findings suggested that T3 exerted osteoprotective effects by suppressing bone remodeling regulators in postmenopausal women with osteopenia, which might be in part mediated by the inhibition of oxidative stress [168]. 

Overall, there is strong evidence on the skeletal benefits of vitamin E supplementation in rats, whereby positive changes in bone structure and strength are observed with supplementation. T3 and its isomer promote bone growth in rats and prevent the progression of bone loss in osteoporosis models. These effects could be mediated by the antioxidant, anti-inflammatory, and gene modulating effects of T3 (Figure 4). However, the limited number of human studies on the effects of T3 on bone health in humans hinders its use clinically. 

## 7. Effects of Tocotrienol on Arthritis

Rheumatoid arthritis is a chronic inflammatory autoimmune disease, in which the immune system mistakenly attacks the joints. T3 exhibits potent anti-inflammatory and anti-oxidative properties, thus suggesting the potential protective effects of T3 on joints in rheumatoid arthritis. Using female rats with collagen-induced arthritis (CIA), Radhakrishnan et al. (2014) revealed that the animals orally treated with γT3 (5 mg/kg) for 25 days (day 21 to day 45) had lower hind paw thickness and edema compared to untreated arthritic animals [169]. Similar outcomes were obtained from another study by Haleagrahara et al. (2014), whereby oral supplementation of δT3 (10 mg/kg) exerted significant impact in lowering both left and right hind paw edema in the CIA animals [170]. The researchers also investigated the involvement of inflammatory response and oxidative stress in this CIA model. Both γT3 and δT3 were capable of decreasing inflammation (reduced CRP and TNF-α) and oxidative stress (increased SOD and total glutathione (GSH)) in the CIA rats [169,170]. A recent study showed that annatto tocotrienol at 100 mg/kg prevented cartilage degeneration in a rat model of osteoarthritis induced by monosodium iodoacetate, as evidenced by reduced histological scoring and circulating cartilage oligomeric matrix protein level in the treated rats [171]. Taken together, findings from these studies suggested that T3 may be effective in protecting against arthritis-induced joint damage (Table 9). Further in vivo and in vitro studies are recommended to confirm the efficacy and uncover the mechanism of action of T3 against cartilage destruction caused by arthritis.

## 8. Effects of Tocotrienol on Muscle Diseases

According to European Working Group on Sarcopenia in Older People (EWGSOP), sarcopenia in older adults (age of >65 years old) can be diagnosed with low muscle mass and either low muscle strength or low physical performance (gait speed ≤ 0.8 m/s) [173]. Sarcopenia is a geriatric syndrome contributed by several factors such as immobilization, endocrine shift, chronic diseases, inflammation, and malnutrition [174]. The prevalence of sarcopenia is higher in men than women because the progressive loss of muscle strength in women was slower [175,176,177]. About 1 to 2% of skeletal muscle mass declines annually after 30 years old in both genders [178]. While in older populations, the loss of leg strength happened around 10–15% decennially before they reach 70 years old, and approximately 2 to 4% per year was observed in more advanced age [176,177]. The loss of muscle regenerative capacity is proposed as one of the possible contributory factors of this age-related muscle deterioration [179]. Studies have also shown that muscle health is dependent on an adequate supply of dietary vitamin E which includes T3 [180]. A diet deficient in vitamin E causes profuse myocyte necrosis and ultimately leads to lethal muscular dystrophy, as demonstrated in various animal model studies [181,182,183,184]. 

Skeletal muscle is well-known for its regeneration competency in restoring and maintaining muscle mass when muscle cells damaged [185]. Muscle regeneration involves four sequential and overlapping phases: degeneration, inflammation, regeneration and remodeling. Satellite cells, the precursor to skeletal muscle cells, are crucial in regenerative phase and will be triggered for proliferation and differentiation in response to stimuli. The proliferating satellite cells are known as myoblasts [186]. The regenerative functions of the resting satellite cells in geriatric mice model decline with ageing and they lose reversible quiescence by switching to an irreversible pre-senescence state [187]. Age-related transient chronic inflammation imposes direct degenerative muscular effect [188] and indirectly causes oxidative stress, promoting the progression of sarcopenia. In addition, the relationship between MetS and sarcopenia has been established [189]. Obesity increases adipocytes followed by an elevation of adipokines, which causes muscle catabolic activity and increment of inflammatory cytokines [190,191]. On the other hand, insulin resistance associated with type 2 diabetes mellitus (T2DM) can also decrease muscle strength [191].

Skeletal muscle is a sensitive site for oxidative damage due to its high consumption of oxygen. An extensive increase in reactive oxygen species (ROS) production over a long period of time may result in amplified protein catabolism and decreased cell survival [192]. The imbalance between ROS production and antioxidant defence responses may alter the expression of transcription factors, for instance, shifting protein synthesis to protein degradation and leading to muscle wasting [193,194]. The accumulation of ROS stimulates MAPK such as extracellular signal-regulated kinase (ERK) and p38 MAPK [195]. These MAPKs act as the preliminary contact between cellular surface and nucleus. ROS will activate the signaling cascade of this pathway causing a change in skeletal muscle mass and function and progressively lead to sarcopenia [196]. Thus, T3 with antioxidant and anti-inflammatory properties could be an alternative to prevent sarcopenia.

The differentiation of myogenic cells is regulated by a family of myogenic regulatory factors (MRFs) that includes MyoD, Myf5, Myogenin, and MRF4. They are also transcription factors that assist protein interactions and DNA binding to activate muscle-specific genes [14]. Lim et al. found that TRF promoted the proliferation capacity of stress-induced premature senescence myocytes [197]. Moreover, TRF treatment was suggested to modulate the renewal of satellite cells through regulation of p53 signaling, cell cycle, Wnt signaling pathway, and expression of MRF [197]. It was also shown to reverse senescence in stress-induced pre-senescence myoblasts, demonstrating the potential therapeutic effects of TRF on senescent muscle cells [198]. Furthermore, Khor et al. suggested that TRF was superior to αTF in promoting myogenic differentiation followed by prevention of replicative senescence of myoblasts [199]. 

In animal studies, TRF-treated rats were shown to swim significantly longer than the control animals [200]. The TRF-treated rats also possessed significantly higher concentrations of antioxidant enzymes such as SOD, catalase (CAT) and GPx, but lower levels of oxidative stress biomarkers, such as blood lactate, plasma and liver TBARS as well as muscle protein carbonyl. This implied that TRF was able to improve muscle health and reduce the exercise-induced oxidative stress in forced swimming rats [201,202]. 

The evidence of T3 in preventing sarcopenia in humans was limited. In a related study examining the relationship between TF and sarcopenia, vitamin E daily intake was positively correlated with knee extension strength and total physical performance score among older adults in Italy. Specifically, a strong correlation was found between γTF and muscle strength, as well as between αTF and muscle strength and physical performance [57]. In another study, a daily dose of 160 mg Tri E^®^ T3 (consisted of 74% T3 and 26% TF) given to participants aged between 37 and 78 years old for 6 months was found to reduce free radical damage as shown by a decreased percentage of DNA damage, sister chromatid exchange frequency (SCE) and urinary 8-OHdG levels [75,203]. The TRF supplementation also caused a reduction in protein damage and serum advanced glycation end products (AGEs) in adults aged above 50 years old [75]. Since increased AGEs was associated with lower muscle strength [204,205], T3 supplementation may prevent a decrease in muscle strength among elderly adults. 

In addition, the relation between potent antioxidant of vitamin E and sarcopenia could also be indicated by knee extension strength test, which is important for the quantification of muscle strength in older adults [206]. A prospective population-based study of older adults in Italy done between 1998 and 2000 reported that vitamin E daily intake was positively correlated with knee extension strength and total physical performance score. An adjusted data according to plasma concentration and daily intake showed a strong correlation between γTF and muscle strength, as well as a significant correlation between αTF and both measurements [207]. Although this study only focused on the effects two types of TF on muscle strength, several experimental studies examining the antioxidant effects of T3 have found that T3 was superior due to their better distribution in the lipid layers of the cell membrane [199,208]. Thus, T3 may act as a better antioxidant than TF. Indeed, the role of T3 on muscle mass and function warrants further investigation. Currently, there is a proposed clinical trial (NCT03708354) to investigate the effect of T3 on skeletal muscle strength and function in humans at Texas Tech University Health Sciences Center.

As a conclusion, T3 may have anti-sarcopenic properties but its effects need to be validated in human studies. This effect may be mediated by the antioxidant, anti-inflammatory, and metabolic activities of T3 (Figure 5).

## 9. Effects of Tocotrienol on Peptic Ulcers

Many studies have shown the involvement of oxidative stress and inflammation as well as diminished gastroprotective factors in the pathogenesis of gastric lesions using various animal models. Therefore, treatment with antioxidants like T3 may provide beneficial effects in attenuating the formation of gastric lesions. Both αTF and T3 reduced gastric lesion index in experimental animals exposed to ulcerogens [209,210,211,212]. Vitamin E possesses antioxidant properties, but T3 was reported to have a higher antioxidant ability compared to TF [213]. Therefore, this section discusses the use of vitamin E in the forms of palm T3 in various animal models of gastric ulcers. 

Gastric ulcer formation is influenced by many factors such as oxidative stress, endogenous gastroprotective substance and inflammation. There are many models used to explore the protective effects of vitamin E, including TF and T3, on gastric lesions development. The commonly used models are stress-induced [214], non-steroidal anti-inflammatory drugs (NSAIDs)-induced [215], ethanol-induced [211], *Helicobacter pylori*-induced [216], acid-induced [217], and pyloric ligation-induced [218]. These models induce the development of gastric ulcer via different mechanisms. 

A balance between the protective and the aggressive factors in the gastric microenvironment are known to play a crucial role in the development of gastric ulcer. Amongst the aggressive factors, elevation of gastric acidity, oxidative stress, and reduction on the mucosal blood flow have been implicated to contribute to injuries on the gastric mucosa [219]. Rats exposed to water-immersion restraint stress (WIRS) developed stress ulcers, which mimics ulcer development in critically ill patients [220]. Pre-treatment with pure T3 mixtures before exposure to WIRS for 28 days significantly reduced gastric acidity, an important aggressive factor in the pathogenesis of stress-induced gastric injuries [221]. 

Exposure to restraint stress also resulted in high gastric oxidative stress content, accompanied by the formation of stress ulcers [220,222,223]. T3 was able to significantly reduce oxidative stress product (MDA) and this was associated with improved gastric ulcers [221,223,224]. Endogenous prostaglandin, in particular prostaglandin E_2_ (PGE_2_), is important in maintaining gastric mucosal integrity. Prostaglandins maintain the gastric mucosa integrity by stimulating secretions of mucus and bicarbonate, modulating mucosa blood flow, inhibiting gastric acid secretion, and influencing the contraction of longitudinal muscle and relaxation of circular muscle [225]. Stress reduced the level of PGE_2_ in the stomach at least by half of its normal level, which was accompanied by formation of lesions [226,227,228]. Interestingly, Rodzian et al. (2013) found that the supplementation annatto T3, which mainly contains δT3 isomer, suppressed gastric PGE_2_ level [229]. On the other hand, palm pure T3 mixture was able to protect the gastric mucosa with the elevation of PGE_2_ level, which was associated with an increased in cyclooxygenase-1 (COX-1) expression [221]. Therefore, the composition of T3 mixture is important in determining the effects on PGE_2_ level. Palm T3 mixture appears to be optimal in reducing the risk of developing gastric ulcers in susceptible patients via reduction of PGE_2_ level. 

Other than the oxidative parameters, studies had also demonstrated the ability of palm T3 in reducing nitrite levels (an indicator of NO), which is mediated by the ability of T3 in downregulating the expression of inducible nitrite oxide synthase (iNOS), leading to decreased production of gastric mucosal NO [230]. Nitric oxide generated from iNOS had been shown to participate in ulcer formation through the production of oxygen-derived radicals and their cytotoxic actions [231]. Marked elevated of nitrite levels could occur due to stimulation of iNOS, which reacts with superoxide to form peroxynitrite, a potent cytotoxic oxidant causing gastric damage [232]. 

Growth factors promote cell proliferation and migration during the process of ulcer healing [233]. Endothelial growth factor (EGF) is required to maintain the integrity of gastric mucosa and promote healing of injured tissue [234]. TGF-α accelerates the replacement of the epithelium and regulates regeneration of epithelial cells in gastric tissues [235]. VEGF assists in repair of ulcers by stimulating angiogenesis and remodeling of connective tissues [233], while basic fibroblast growth factor (bFGF) is known to stimulate synthesis of local prostaglandins, which ultimately leads to increased formation of blood vessels [236] and endothelial cell proliferation [237]. This assists in maintenance and recovery of gastric tissue in the event of injury. T3 was reported to exert its effectiveness by diverse mechanisms of protection, including upregulating the expression of bFGF, TGF-α, and VEGF in a stress-induced gastric ulcer rat model [238], which ultimately prevents and repairs gastric mucosal injuries. A summary of the literature on the effects of T3 on stress-induced gastric ulcer model is shown in Table 10.

Non-steroidal anti-inflammatory drugs and its adverse effects on the stomach are yet another unresolved medical problem. A single exposure to indomethacin leads to severe gastric ulceration, which was not reduced by T3 (60 mg/kg) treatment for 4 weeks [239]. However, Nafeeza et al. (2002) showed that the ulcers induced by aspirin were significantly reduced when the rats were supplemented with PVE at 300 mg/kg food for 8 weeks [240]. These findings suggested that the dose and duration of treatment may play a determining role in the outcome of the treatment with PVE. It appears that a longer duration with a higher dose of PVE is warranted to produce a positive effect [240]. 

An increase in oxidative stress is usually measured by the formation of MDA content which is an end product of lipid peroxidation. Elevated MDA content in the rat stomach was consistently observed in various models of gastric lesion, including NSAIDs models [241]. PVE and TF were observed to reduce gastric content of MDA in aspirin-induced rats [242]. Reduced lipid peroxidation was also demonstrated in the group which received a combination of T3 and TF [239]. The free radicals can also be endogenously detoxified by reduced glutathione. In many models of gastric lesions, the gastric reduced glutathione content was significantly decreased. Treatment with αTF and/or palm T3 in these models protected against the reduction in the GSH content [243,244]. Vitamin E might spare the GSH by scavenging the free radicals before they could react with the GSH.

Another source of oxyradicals is myeloperoxidase (MPO) enzyme which is a marker for neutrophil infiltration [210]. It produces hypochlorus acid from hydrogen peroxide and chloride ion. The MPO activity was reported to be elevated in aspirin and indomethacin-induced gastric ulcer models [210,243,245]. The increase in MPO activity was attenuated by αTF administration, which also reduced gastric lesions and formation of leukocyte-generated oxygen radicals [210,243,245]. These findings suggest that αTF reduced the severity of gastric lesion via inhibition of neutrophil infiltration. However, there is a lack of studies on the effect of T3 on MPO enzyme but the TF in PVE may possibly provide a similar effect. It might retard the earlier oxidative events that lead to neutrophil infiltration. 

PVE was also shown to be effective in increasing the PGE_2_ level in rats given indomethacin [239] but not in ethanol-induced ulcers [246]. Prostaglandin is produced from arachidonic acid by the action of cyclooxygenase (COX) enzyme. COX exists in two isoforms; constitutive COX-1 and inducible cyclooxygenase-2 (COX-2) [247]. NSAIDs significantly inhibited both COX [248]. T3 pretreatment was shown to downregulate COX-2 expression in stress-induced ulcers; thus, similar observation could be possible for NSAIDs-induced gastric ulcers. It also upregulated gastric COX-1 expression, the enzyme which is involved in PGE_2_ synthesis, hence increased the protective PGE_2_ content in the stomach [221]. A summary of the literature on the effect of T3 on NSAIDS-induced gastric model subjects is shown listed Table 10.

Gastric ulcers induced by ethanol exposure in rats were significantly reduced with the supplementation of PVE at 150 mg/kg in food for 3 weeks [249,250]. The ulcers healing was also accelerated when PVE (150 mg/kg food) was given post-ulcer induction with ethanol [249]. These effects were achieved by retarding the lipid peroxidation induced by ethanol. PVE also exerts similar effects like αTF in reducing lipid peroxidation in rats exposed to ethanol [242]. The effects of PVE and T3 on gastric acidity were different according to the animal model used. PVE suppressed the increased in gastric acidity induced by stress [221] and ethanol [242] but not aspirin [242]. Since ethanol-induced gastric lesions were more severe compared to aspirin-induced lesions, a larger dose of PVE (150 mg/kg diet) was needed for ulcer prevention. Aspirin-induced lesions could be prevented by PVE in doses as low as 20 or 60 mg/kg diet. 

Another important protective factor in the gastric mucosa is the mucus covering it [251]. However, the gastric adherent mucus content was unaltered in rats fed with a diet containing various doses of TRF (60, 100, 150, and 300 mg/kg) for 4–8 weeks and administered absolute ethanol even though there was a reduction in gastric lesion index compared to the untreated control [240,242,244,246,249]. A similar finding was also observed in rats given a diet containing αTF (20, 30, 50, and 300 mg/kg) for 4–8 weeks and challenged with aspirin [240,242], which confirms that neither T3 or TF exerts any significant effect on gastric mucus production. A summary of the literature on the effect of T3 on ethanol-induced gastric model subjects was listed in Table 10.

Conclusively, palm T3 confers its protection against ulcerogenic factors primarily via its antioxidant, anti-inflammatory mechanisms and maintenance of gastric protective factors (Figure 6). Both TF and T3 have comparable gastroprotection against gastric mucosal injury. It may be recommended to be taken as supplement to avoid gastric lesions precipitated by stress or other aggressive agents or as a part of the treatment regime in critically ill patients. However, no clinical studies have been conducted on the use of T3 in patients with gastric lesions so far. Future research should be carried out in order to confirm its gastroprotective properties in humans. 

## 10. Effects of Tocotrienol on Neurodegenerative Diseases

Vitamin E, particularly αT3, has been proven to play a vital role in maintaining the integrity of the central nervous system (CNS). The brain is specifically susceptible to oxidative stress as it is an organ with high metabolic needs, hence the higher requirement of oxygen consumption. The abundance of polyunsaturated fatty acids (PUFAs) also renders the brain more susceptible to lipid peroxidation process [253]. Neurodegenerative diseases have been linked to oxidative stress-induced cell death, also known as oxytosis [254]. In normal aging process or in patients with age-associated neurodegenerative diseases, selective areas of brain neurons will undergo degeneration or cell death. For example, in Parkinson’s disease, neurons of the substantia nigra are vulnerable to cell death whereas in Alzheimer’s disease, neurons in the hippocampus and frontal cortex are prone to neurodegeneration [255]. In a pathologic brain, the generation of free radicals results in a rise of lipid peroxidation products which mediates and accelerates apoptotic neuronal cell death [253].

Parkinson’s disease is a neurodegenerative disorder which primarily affects motor function, due to the loss of dopaminergic neurons in the substantia nigra of the brain caused by factors like oxidative stress [256]. It involved the formation of Lewy bodies, an abnormal accumulation of protein develops inside neurons and Lewy neurites [257]. Parkinson’s disease can be categorized into 6 stages. Stage 1 and 2 are the pre-symptomatic stages which affect the medulla oblongata and olfactory bulb. Stage 3 and 4 progress to substantia nigra and other nuclei of forebrain and midbrain while stage 5 and 6 involve the neocortex [257]. Clinical symptoms start to develop at stage 3. Current treatments for this disease are mainly symptomatic. However, recent therapies have emphasized neuroprotection as one of the major cornerstones in treatment for Parkinson’s disease. Pathogenesis of Parkinson’s disease is strongly related to oxidative stress [258]. 

The neuroprotective effect of T3 have been tested on cellular and animal models. Nakaso et al. (2014) found that T3, especially γT3 and δT3 exerted cytoprotective effect on neuroblastoma cells through oestrogen receptor (ER)-β/phosphoinositide-3-kinase (P13K)/protein kinase B (Akt) signaling pathway [259]. In 2016, Nakaso et al. (2016) performed a study on mice model to verify the ERβ-P13k/Akt mechanism and found that δT3 protected dopamine neurons in substantia nigra induced with neurotoxicity [260]. In addition, δT3 improved motor activity in a mice model of Parkinson’s disease [260]. An earlier study by Lan et al. (1997) showed that vitamin E-pretreated mice showed reduction in oxidative stress markers. In addition, vitamin E prevented the loss of dopaminergic neurons and reduced iron accumulation in the mice brain [261]. However, in these studies, the vitamin E considered is αTF. 

Evidence from human studies reported contradicting outcomes on the relationship between vitamin E and Parkinson’s disease. An earlier multicenter, controlled clinical trial recruited untreated patients with Parkinson’s disease and assigned TF supplementation (2000 IU per day). The results indicated that TF did not delay the onset of disability associated with early, otherwise untreated Parkinson’s disease [262]. A nested case-control study, idiopathic Parkinson’s disease patients (n = 84) were identified and dietary data sets were obtained at the time of cohort enrolment using a food-frequency questionnaire and a 24 h photograph-assisted dietary recall. The results found no conclusive evidence that individuals with and without Parkinson’s disease differed with regards to vitamin E intake [263]. In the same year, Hellenbrand et al. (1996) conducted a case-control study to compare the past dietary habits on αTF intake between Parkinson’s disease patients and control subjects, but no difference was found among the two groups [264]. Another case-control study was performed in subsequent year by Scheider et al. (1997). The authors reported that long-term dietary vitamin E intake did not provide protective effects on Parkinson’s disease [265]. Similar outcomes were obtained by Anderson et al. (1999) whereby there was no association between past vitamin E intake and risk of Parkinson’s disease [266]. 

On the other hand, several previous studies displayed paradoxical findings. A population-based case-control study enrolled patients diagnosed with Parkinson’s disease between year 1991 to 1995 (n = 126) and a standardized food frequency questionnaire was used to collect data for the foods they consumed within the past year. This study provided evidence that vitamin E was potentially protective against Parkinson’s disease [267]. A cohort study by Zhang et al. (2002) suggested that food rich in vitamin E might be protective against Parkinson’s disease, as shown by their findings that the reduction in risk of was associated with high intake of vitamin E [268]. In addition, a cross-sectional study by de Rijk et al. (1997) found a significant inverse relationship between vitamin E intake and Parkinson’s disease [269]. In year 2005, a meta-analysis was done using data from five case-control studies [263,264,265,266,267], one cohort study [268], and one cross-sectional study [269]. In this analysis, Etminan et al. (2005) studied the effects of vitamin E intake and the risk of Parkinson’s disease and reported that moderate to high dietary intake of naturally-occurring vitamin E diminishes the risk of developing Parkinson’s disease [270]. A summary of the literature on the effects of vitamin E on Parkinson’s Disease is listed in Table 11 and the mechanism is presented in Figure 7. Despite the abundant evidence of vitamin E on Parkinson’s Disease in humans, studies on T3 are lacking. 

Alzheimer’s disease is an age-associated inflammatory neurodegenerative disease marked by the presence of pathognomonic amyloid plaques and neurofibrillary tangles [271]. As the plaques and tangles accumulate in the brain, they destroy the synapses that mediate memory and cognitive function [272]. Amyloid-beta (Aβ) plaques also exacerbate production of ROS, such as hydrogen peroxides and hydroxyl radical, which worsens the Alzheimer’s disease condition [273]. It is postulated that increased oxidative stress and reduced antioxidant levels may play an important role in the development and progression of Alzheimer’s disease [274]. 

Several in vitro studies have unambiguously demonstrated the neuroprotective effects of T3. Huebbe et al. (2007) revealed that pre-treatment of human neuroblastoma (SH-SY5Y) cells with αTF, γTF, αT3, or γT3 counteracted the cytotoxicity of tert-butyl hydroperoxide (tBHP) with the highest protection seen in αTF-treated cells. Only αTF, γTF, and αT3 exerted protective effects on buthionine sulfoximine (BSO)-induced cytotoxicity [275]. Saito and co-authors reported that αTF and αT3 attenuated cell death induced by glutamate using primary cortical neurons isolated from cerebral cortex of Sprague-Dawley rats fetuses [276]. Another in vitro study using human neuroblastoma (SH-SY5Y) cells found that αTF and αT3 decreased cholesterol and ROS like hypochlorite (-OCl) and free hydroxyl radical (OH) as vitamin E had been proposed to have cholesterol-lowering and antioxidant effects [277]. However, the study also revealed enhanced Aβ production and reduced Aβ degradation in αT3-treated cells [277]. This side effect of T3 should be considered when initiating treatment in Alzheimer’s disease subjects. To counteract mitochondrial dysfunction which occurs early in Alzheimer’s disease, neuron culture was treated with rice bran extract consisting of TF and T3 and their mitochondrial function was improved [278]. Selvaraju et al. (2014) discovered the prophylactic and recovery effects of TRF on astrocytes towards glutamate toxicity. TRF was shown to exhibit antioxidant effect as it reduced lipid peroxidation and increased cell survival [279]. 

Animal studies also supported the antioxidant activity of T3 on Alzheimer’s disease. TRF modulated antioxidant enzymes activity and diminished DNA damage in Alzheimer’s disease mouse model [280]. Another study on rice bran extract using mice model showed similar results, whereby it enhanced mitochondrial functions such as mitochondrial respiration and membrane potential [281]. Tiwari et al. (2009) demonstrated that T3 improved learning and memory in male Wistar rats [282]. Level of acetylcholinesterase (AChE), which degrades acetylcholine associated with learning and memory, decreased in vitamin E isomers treated groups whereas antioxidant like glutathione increased [282]. Complexation of annatto-derived T3 with γ-cyclodextrin in diet also enhanced mitochondrial membrane potential and adenosine triphosphate (ATP) concentration in aged mice [283].

A population-based cohort study reported that higher plasma concentration of vitamin E at baseline predicted a lower risk for developing Alzheimer’s disease later in life [284]. Studies further demonstrated an inverse relationship between risk of cognitive impairment and serum levels of T3 isomers in the elderly [285]. The study of Mangialasche et al. (2013) revealed that lower concentration of vitamin E isomers in the plasma of Alzheimer’s disease patients compared to the controls [286]. Besides, magnetic resonance imaging (MRI) results in combination with T3 levels predicted Alzheimer’s disease more accurately than MRI alone. In addition, Mangialasche et al. (2012) showed reduced total TF, T3, and vitamin E in mild cognitive impairment (MCI) and Alzheimer’s disease subjects [287]. Dietary intake of vitamin E (T3 and TF) was also associated with lower risk of developing dementia, whereby a study found that subjects with higher vitamin E intake had lower long-term risk to develop dementia and Alzheimer’s disease [288]. A summary of the literature on the effects of T3 on Alzheimer’s disease is shown in Table 12 and the mechanism is presented in Figure 8. 

The neuroprotective role of T3 in preventing and attenuating the damages associated with neurodegenerative disorders have been established by previous studies. The main mechanism through which T3 attenuates the neurodegenerative changes of these disorders is via its antioxidant action, either by reducing the generation of ROS or decreasing lipid peroxidation by-products. However, this evidence largely focused on two main neurodegenerative disorders, namely Parkinson’s disease and Alzheimer’s disease. There are still gaps in knowledge as to how T3 can assist in other neurodegenerative condition, which can be further explored in future studies. 

## 11. Effects of Tocotrienol on Wound Healing

Wounds are injuries that damage or disrupt the structure and function of the skin [290]. They can occur superficially on the surface of the tissue resulting in the loss of integrity in the epithelial layers. The acute wound healing process takes place in a dynamic and orderly manner, involving a coordinated series of homeostasis; coagulation; inflammatory reactions; proliferation of connecting tissues and parenchymal cells; accumulation of extracellular matrix proteins; as well as tissue and parenchymal cells remodeling [291]. Disorders in the acute wound healing process contribute to long periods of healing of the tissue, known as chronic wound healing, associated with hard-to-heal ulcers, and excessive scarring [292]. Delayed-to-heal wounds can be quite a burden to patients, emotionally and financially. Abnormal wound healing can be seen in patients who are immunocompromised, those with underlying diseases such as diabetes, cancer, infection, malnutrition, and even old age [293]. 

Evidence on the effects of T3 on wound healing is largely preclinical (Table 13). Unlike tocopherols, there are no clinical trials examining T3 and wound healing per se. However, there was a prospective double-blinded clinical study looking at the efficacy of topical T3 in the prevention of hypertrophic scars [294]. Most of the animal studies were carried out in diabetic rodents inflicted with excisional wounds. T3 was administered orally in all but one study. In the study by Xu et al. (2017), the researchers used topical application of mono-epoxy-T3-α (MeT3α), an αT3 with one epoxide moiety, on excisional wounds in diabetic mice [295]. Previous studies have shown that modification of T3 with one or more epoxide moieties on their polyisoprene side-chains resulted in an increase in the synthesis of endogenous lipid soluble antioxidant co-enzyme Q (CoQ) [296]. It is interesting to note that both topical and oral administration of T3 produced positive results on wound healing properties [295,297,298,299,300]. Accelerated healing rate resulted in earlier wound contraction. This was accompanied by increased serum protein synthesis and elevated protein content in the wounds, indicating accelerated production and migration of new cells to the wound area. In addition to accelerated wound repair, the diabetic animals had better glycemic control and improvement in the renal status. T3 administration also resulted in an increase in endogenous antioxidant profile with a reduction in the oxidative stress markers [299]. All these were made possible as T3 was shown to promote angiogenesis, epithelization, granulation, and collagen production [295,301]. T3 may have a role in scar management as it was shown to inhibit hypertrophic scar fibroblast [302]. However, when translated to human studies, the outcome of topical application was paradoxical [294]. This could probably be due to poor absorption of T3 by the scarred skin. As a conclusion, T3 has a good potential to be used as an agent to expedite wound repair (Figure 9). Its role in preventing hypertrophic scar can be explored further if modifications can be made for it to be absorbed better when administered topically.

## 12. Effects of Tocotrienol on Obesity, Diabetes, and Their Complications

Obesity and diabetes mellitus (DM) is the major global health issue nowadays. More than 1.9 billion adults (39%) are overweight and obese in 2016 [1] and an estimated 422 million adults were suffering from DM in 2014 [304]. Obesity is associated with low-grade inflammation with reduction in adiponectin level, activation of nuclear factor-kappa B (NF-κB), and release of pro-inflammatory mediators [305]. The pro-inflammatory state subsequently affects the insulin expression and signaling, which give rise to the risk of DM [305]. In addition, DM, including Type 1 DM (T1DM) and T2DM, is characterized by hyperglycemia associated with the increase of oxidative stress [306,307] which contributes to the diabetic CVD, diabetic retinopathy, neuropathy, and nephropathy [308,309,310].

The anti-inflammatory activities of T3 were well-reported from in vitro studies [311,312,313,314,315,316,317], obese animals [315,316,317], diabetic animals [318,319], and patients with diabetes [320,321]. The effect of T3 on the obese population remains unknown. T3 from rice bran extracts [314], muscadine grape seed oil [317], TRF [317,319], annatto oil [313], and purified γT3 [311,315,316,318] were reported to suppress the release and expression of pro-inflammatory mediators including TNF-α, granulocyte colony-stimulating factor, CCAAT-enhancer-binding proteins β, leptin hormone, IL-1β, IL-6, IL-8, iNOS, and MCP-1. In addition, annatto oil T3 mixture upregulated IL-10 (anti-inflammatory cytokine) mRNA level in the adipose tissue of obese C57BL/6J mice [313]. Annatto T3 [313] and γT3 supplement (0.05%) [316] also suppressed the macrophage infiltration in the adipose tissue of obese mice. Besides, γT3 was reported to suppress NF-κB activation and downstream gene expression from TGFβ-activated kinase 1, receptor-interacting protein, tumor necrosis factor, phorbol myristate acetate, okadaic acid, lipopolysaccharide, cigarette smoke condensate, IL-1β, and epidermal growth factor induction [312]. Furthermore, electrophoretic mobility shift assay also revealed that γT3 is the most potent NF-κB inhibitor as compared to other T3 isoforms [312]. The mechanistic study identified that γT3 inhibited NF-κB activation by inducing the degradation of tumor necrosis factor receptor-associated factor 6 (positive regulator of NF-κB) [318] and inhibiting nuclear factor-κB inhibitor type α kinase (IKK) signaling, Akt signaling, and nuclear factor-κB inhibitor type α (IκBα) degradation [311,312,315,316]. Moreover, γT3 was also demonstrated to inhibit caspase-1 activation and interleukin-1β secretion via suppression of nucleotide-binding oligomerization domain-like receptor protein 3 (NLRP3) inflammasome and AMPK activation [318]. From DM patient study, supplementation of T3-enriched canola oil (200 mg/day) for 8 weeks significantly reduced serum high sensitivity CRP among T2DM patients [321]. Another recent human study, however, reported that oral administration of palm oil TRF (552 mg/day) did not improve the inflammatory status of T2DM patients [320].

Apart from anti-inflammatory activities, T3 exhibited antioxidant activities by reducing total NO, MDA, and 4-HNE as well as restoring non-thiols protein level and antioxidant enzymes activities in diabetic animal models [58,319,322,323,324,325,326,327,328]. Moreover, TRF showed radical scavenging capacity via 1-diphenyl picrylhydrazyl assay [104]. Furthermore, T3 protected leukocytes of diabetic rats from oxidative DNA damage [58,328]. In DM patients, supplementation of T3-enriched canola oil (200 mg/day) significantly reduced serum MDA level with an increase of total antioxidant capacity [329]. Similarly, the supplementation of T3-rich extract, *Palmvitee* (1800 mg/day) also significantly reduced the MDA level of T2DM patients [330]. However, the group receiving the vehicle (refined palm oil with <0.1% T3) also experienced a reduction in the MDA level. Thus, this put the beneficial effects of *Palmvitee* in doubts [330].

The anti-obesity effects of T3 have also been demonstrated in in vitro studies [331,332,333] and obese animal models [41,102,103,314,316,334]. T3 suppressed adipogenesis with the decrease of intracellular TG level and lipid droplets in mouse hepatoma Hepa 1-6 cells [40], human hepatoma HepG2 cells [41,47], 3T3-L1 murine preadipocytes [331,335], 3T3-F442A murine preadipocytes [336], and primary human adipose-derived stem cells [317,332]. The potency of anti-adipogenic activities of T3 follows the rank order of γ > δ > β > α [331,332]. From obese animal studies, T3 was reported to reduce body weight, total abdominal, epididymal, and mesenteric fat mass in high fat diet-fed obese rodents [41,102,103,314,316]. However, human study on the effect of T3 in obesity is limited.

T3s (especially γT3 and δT3) were demonstrated to improve glycemic control in in vitro studies [316,318], obese animals [102,103,313,316], diabetic animals [58,318,319,322,323,324,325,327,328,337,338,339,340], and human population [329,341,342,343]. There is a lack of study reported on the hypoglycemic effect of T3 on the obese population. γT3 restored the glucose uptake, insulin sensitivity and Akt signaling in differentiated primary mouse adipocytes [318] and primary human adipocyte stem cells [316]. Palm TRF, δT3-enriched palm olein, annatto oil T3 mixture, and purified T3 also significantly improved the glucose tolerance in obese rodents [102,103,313,316]. Apart from this, T3 was demonstrated to improve insulin synthesis (insulinotropic activity) and enhance insulin sensitivity in diabetic animals [58,318,319,322,323,324,325,327,328,337,338,339,340]. δT3 is the most potent insulinotropic agent, followed by γT3 and αT3 [340]. Moreover, the combination treatment of TRF and insulin resulted in better glycemic control in diabetic rats [322,323,324]. Besides, γT3 suppressed the progression of diabetes in BKS.Cg-*Dock*7^m+/+^ Lepr^db^/J diabetic mice with an increase of adiponectin level, reduced the loss of pancreatic β-cells and increased the average islet size, size distribution, and insulin-positive area with lesser immune cell infiltration [318]. Furthermore, T3s, including palm TRF [319,339] and purified T3 isoforms [339,340], were reported to activate peroxisome proliferator-activated receptor (PPAR) to improve insulin sensitivity. αT3, γT3, and δT3 were reported to upregulate PPAR-δ and PPAR-γ mRNA levels in glucose-stimulated primary normal rat pancreatic β-islet cells [340]. The molecular analysis also revealed that αT3 and γT3 were served as PPAR-α-selective agonist while δT3 was a pan-PPAR agonist [339]. Molecular docking analysis also demonstrated that δT3 exhibited higher affinity and formed greater hydrogen bonding interaction with PPAR-δ and PPAR-γ which explaining its pan-PPAR agonist properties [340]. Furthermore, δT3 was also demonstrated with the greater potency as direct PPAR-α agonist by enhancing PPAR-α and PPAR-γ coactivator-1α protein interaction as compared to others [340]. In like manner, δT3 did not possess any direct antidiabetic activity where it only weakly reduced the accumulation of AGEs in a cell-free anti-AGEs assay [344]. On the other hand, T3 supplement, *Tocomin*^®^ was unable to improve glycemic control in obese rats [334,345]. This might due to the composition of *Tocomin*^®^ and the stable glycemic status of these rats [334].

The anti-diabetic activities of T3 on human studies are heterogeneous [72,320,329,330,341,342,343]. A Finnish Mobile Health Examination Survey with 23-years of the follow-up period on 2285 men and 2019 women (free from DM in starting) identified that dietary intake of βT3 was significantly associated with the lower risk of T2DM [343]. The lack of association between the disease and other T3s may due to their extremely low dietary intake or the small DM case size among the subjects [343]. Parallel with this, another randomized, double-blind and placebo-controlled cohort study, Alpha-Tocopherol, Beta-Carotene Cancer Prevention Study with 10 years follow-up period on 29,133 male Finnish smokers also reported similar finding, whereby dietary βT3 was positively associated with the lower risk of T2DM [342]. This association, however, was found to be not significant upon multivariate demographic factors adjustment [342]. This cohort study was limited by recall bias and dietary changes during the study period [342]. In addition, supplementation of T3-enriched canola oil (200 mg/day) significantly reduced the fasting blood glucose level among T2DM patients [329]. Similarly, the Vitamin E in Neuroprotection Study (VENUS) also revealed that consumption of oral mixed T3 (400 mg/day) capsule for 12 months resulted in slightly better glycemic control with no adverse effect among DM patients, but failed to reduce the diabetic neuropathic symptoms [341]. On the other hand, Wan Nazaimoon et al. reported that *Palmvitee* did not improve glycemic control of T1DM patients [330]. Similarly, TRF treatment (6 mg/kg/day) for 60 days also failed to reduce the glycated hemoglobin (HbA1c), fasting and postprandial glucose levels among T2DM patients with hyperlipidemia [72]. Moreover, oral administration of palm oil TRF (552 mg/day) for 8 weeks also did not improve the glycemic status of 86 T2DM patients [320]. The negative anti-diabetic effect of TRF in these two human studies [72,320] might be because of the stable baseline of glycemic markers among these patients.

T3 was reported to improve liver function in obese animals [41,102,103,313]. Palm TRF [102], αT3, γT3 and δT3-enriched palm olein (85 mg/kg/day) [103] were demonstrated to reduce liver injury in obese rats with lower liver injury markers including plasma aspartate transaminase (AST) and alanine transaminase (ALT). In addition, annatto oil T3 mixture [313], TRF, αT3, γT3, and δT3 [102,103] were reported to suppress hepatic steatosis and inflammation in obese rodent models. Moreover, T3 also reduced lipid abnormalities in obese animals [41,47,102,103,313] and diabetic animals [58,327,338,346]. γT3-enriched rice bran oil diet was reported to increase fecal neutral sterols and bile acid excretion in diabetic rats [338,346] with the upregulation of hepatic cholesterol 7α-hydroxylase, low-density lipoprotein receptor and HMGCR expression [338]. Yet, the effects of T3 on liver and lipid profiles of obese population remain unknown. TRF was shown to reduced serum total lipids, TC and LDL-C levels among T2DM patients with hyperlipidemia [72]. On the other hand, oral administration of palm oil TRF (552 mg/day) [320] and *Palmvitee* (1800 mg/day) [330] did not improve the lipid profile among T2DM patients.

Oral administration of *Tocomin*^®^ [334], palm TRF [102], γT3 and δT3 (but not αT3) [103] significantly improved the cardiovascular function of obese rodents by normalizing the systolic BP, improving the systolic function and contractile response, and reducing myocardial inflammation. Besides, T3 was showed to reduce cardiovascular complications in diabetic animals by protecting the thoracic aorta of diabetic rats from vascular wall alterations and degeneration [58]. Moreover, TRF was also found to be beneficial in hypertension management by enhancing the acetylcholine-induced relaxation of aorta rings in SHRs and streptozotocin (STZ)-induced diabetic Wistar Kyoto rats [104]. The effect of T3 on cardiovascular function among the obese population is yet to be determined. The only human study so far had reported that TRF did not improve the vascular function, systolic and diastolic pressures among T2DM patients [72,320].

Nephroprotective effects of T3 were also reported where palm oil and rice bran oil TRF [325,327] and T3 mixture [324] significantly improved the renal function of diabetic animals by increasing the urea and creatinine clearance, and reducing diabetes-related polyuria, hyperalbuminuria, hypercreatinuria, proteinuria, hypercreatinemia, and high blood urea nitrogen (BUN) levels. Besides, T3 also suppressed diabetes-mediated oxidative stress, inflammation, and cellular damage in kidneys [324,325,327]. Yet, the nephroprotective effects of T3 on DM patient remain scarce. Supplementation of T3-enriched canola oil (200 mg/day) significantly reduced urine microalbumin but did not reduce serum NO, urine creatinine level, and urine volume among 44 T2DM patients [321].

There were several animal studies conducted to investigate the effect of T3 on diabetic cataractogenesis [347,348]. Topical application of T3 mixture in microemulsion formulation or liposomal formulation (0.02% and 0.03%) slightly slowed down the diabetic cataractogenesis in galactose diet-induced diabetic rats by reducing ocular opacity index and normalizing the total lens protein, soluble lens protein, insoluble lens protein, and soluble to insoluble lens protein ratio [347]. There is no significant difference in treatment efficacy between microemulsion and liposomal formulation [347]. In addition to this, 0.2% microemulsion formulation was reported not beneficial in preventing diabetic cataract progression, possibly due to its pro-oxidant activities [347]. Besides, the anti-cataractogenic effect of microemulsion formulation was found to be associated with normalization of the redox status, lenticular ATP level, sorbitol level, ATPases activities, calpain 2 activity and lens aldose reductase activity, and the suppression of NF-κB activation [347,348].

In addition, T3 mixture was demonstrated to reduce DM-related neuromuscular disorders in diabetic rats [319,322,323]. Oral administration of palm oil TRF (100 and 300 mg/kg bw) for 12 weeks significantly reduced the DM-related muscle atrophy by increasing mitochondrial biogenesis, at the same time, suppressing oxidative stress, inflammation, and apoptosis in skeletal muscle of diabetic mice [319]. In addition, consumption of T3 mixture (25, 50, and 100 mg/kg/day) also dose-dependently improved the behavior and memory performances of diabetic rats by suppressing the cerebral cortex AChE activity, inflammation, and cell death events [322]. Furthermore, the T3 mixture (100 mg/kg body weight/day) also improved the nociceptive threshold on thermal hyperalgesia, mechanical hyperalgesia, and tactile allodynia by reducing the oxidative stress, inflammation, and cellular damage in the sciatic nerves of diabetic rats [323]. However, oral mixed T3 (400 mg/day) supplementation for 1 year did not significantly reduce neuropathic symptoms among 229 DM patients with diabetic peripheral neuropathy syndromes as reported from VENUS human study [341]. Overall, the effects of T3 on energy metabolism are summarized in Figure 10.

## 13. Effects of Tocotrienol on Cancers

Cancer is one of the leading causes of death worldwide. A total of 14.1 million new cases and 8.2 million deaths were reported in 2012 [349]. In 2017, 1,688,780 new cancer cases and 600,920 cancer deaths were projected to occur in United States [350]. In Asia alone, cancer cases were estimated to increase from 1.6 million in 2008 to 10.7 million in 2030, with the highest incidence of lung cancer in men and breast cancer in women [349].

The anti-neoplastic effects of T3 have been extensively studied using tumor cells of mammary gland [351,352,353,354,355,356], lung [357,358,359,360,361,362], liver [363,364,365,366,367] pancreas [368], prostate [362,369,370,371,372], cervix [373,374], skin [362,375,376,377], brain [357,358,359,378], stomach [379], and colon [378,380]. Besides that, T3 tends to accumulate in the liver and pancreatic tissues as well as tumor tissues with an unknown mechanism [367,381]. T3 possesses anti-cancer properties through the intracellular signaling pathways, such as the inhibition of cell proliferation, induction of apoptosis, induction of cell cycle arrest, suppression of angiogenesis and inhibition of metastasis [16]. Apoptosis is a programmed cell death crucial for defending a host against cancer initiation. The mechanisms of apoptosis in cancer cells involve two major pathways: the intrinsic (mediated by mitochondrial pathway) and extrinsic (mediated by death receptor pathway) pathways [382].

The intrinsic pathway (mitochondrial pathway) of apoptosis depends on mitochondrial membrane permeability. Under the influence of various stressors, a critical apoptotic event known as mitochondrial outer membrane permeabilization (MOMP) occurs, which is triggered by decreased levels of B-cell lymphoma 2 (Bcl-2) and B-cell lymphoma-extra large (Bcl-xl, the anti-apoptotic proteins) as well as increased levels of Bcl-2-associated X protein (Bax) and Bcl-2-antagonist/killer 1 (Bak) (the pro-apoptotic proteins) [383]. The disruption of the membrane permeability, cytochrome c is released from the mitochondria to the cytosol to form an apoptosome complex consisting apoptosis protease activating factor-1 (Apaf-1) and cofactor deoxyadenosine triphosphate (dATP). Subsequently, caspase-9 is recruited to activate downstream caspase-3/6/7 and promote apoptosis [384]. T3 induced ER stress in cancer cells, which is evidenced by the increased levels of ER stress-related proteins, such as binding immunoglobulin protein (BiP), eukaryotic Initiation Factor 2α (eIF2α), protein kinase R-like ER-localized *eIF2α* kinase (PERK), inositol-requiring enzyme 1α (IRE1α) and C/EBP homologous protein (CHOP) [371,374,377,385]. T3s, including αT3, γT3, and δT3—but not αTF—could induce an early intracellular calcium ion release from ER lumen as the initial signal to trigger ER stress [374]. Both γT3 and δT3 were more potent in inducing ER stress than αT3 with significant caspase-12 activation and upregulation of X-box binding protein-1 and glucose-regulated protein 78 [374]. However, only δT3 is specific in inducing IRE1α-(but not PERK and ATF6) mediated ER stress pathway with an unknown mechanism [374].

ER stress leads to the activation of Bax and Bak, followed by signal transmission from ER to mitochondria and execution of death. Several groups of researchers found that T3 increased the ratio of Bax/Bcl-2 (an indicator for MOMP), cleavage of caspase-3 and poly (ADP-ribose) polymerase (PARP) as well as release of cytochrome c, thus inhibiting the DNA repair and contributing to cell death [359,373,377]. In addition, anti-tumor activity of δT3 was evaluated in vivo, whereby female immunodeficient CD1-nu/nu mice were injected (s.c.) with A375 cells and orally treated with δT3 (100 mg/kg) for 35 days. Reduced tumor volume, tumor weight, and delayed tumor progression were observed in the T3-treated group compared to the control group [377]. Together, both in vitro and in vivo data suggested the pro-apoptotic effects of T3 were mediated through the intrinsic apoptosis pathway.

On the other hand, the extrinsic apoptotic pathway is initiated outside the cells by the interaction between death ligands (TNF-α, first apoptosis signal ligand (Fas-L) and TNF-related *apoptosis*-inducing ligand (TRAIL)) and death receptors (TNFR, Fas and TRAIL-R). The binding recruits Fas-associated protein with death domain (FADD) adapter molecule and activates caspase-8. Caspase-3 are the downstream molecule of caspase-8 involved in dismantling of the cell components leading to apoptosis [386]. Indeed, the extrinsic pathway directly affects the amplification of intrinsic pathway as caspase-8 cleaves BH3 interacting domain death agonist (Bid) to promote MOMP [387]. Abubakar et al. demonstrated the combined treatment of δT3 and jerantinine B triggered caspase-8 and caspase-3 activities in human brain glioblastoma and colon adenocarcinoma [378]. Similarly, another study indicated that βT3 induced caspase-8 activity, increased levels of Bid and Bax protein, as well as altered mitochondrial permeability and levels of cytochrome c in lung adenocarcinoma and brain glioblastoma [358]. Overall, the previous evidence theorized that the induction of apoptosis in cancer cells by T3 involved the interplay between both extrinsic and intrinsic pathways.

Cell cycle arrest is a regulatory process that halts the duplication and division processes in the cell cycle [388]. In an in vitro pancreatic cancer model, δ-T3 raised the proportion of pancreatic cancer cells (PANC-1) in G1 phase and lowered the proportion of cells in S phase, implying the cells were arrested in G1 phase. This finding was evident by the increased level of p21, a cyclin-dependent kinase (CDK) inhibitor that suppresses complexes of cyclin/CDK and subsequently inhibits the S phase-promoting E2F-1 transcription factor [368]. In the same year, Ananthula et al. reported the oxazine derivatives of γT3 and δT3 inhibited mammary gland tumor growth in female BALB/c mice implanted with +SA cells (breast cancer cells). Mechanistically, the outcomes were associated with reduced cell cycle progression (cyclin D1, CDK2, CDK4, and CDK6) and increased in cell cycle arrest proteins (p21 and p27) [389]. Furthermore, a naturally occurring T3 mixture (containing 8.3 g αT3, 1.5 g βT3, 11.4 g γT3, 4.6 g δT3, and 6.0 g αTF per 100 g of product) suppressed prostate tumor growth in 8-week-old male NCr (−/−) nude mice by promoting the CDK inhibitors p21 and p27 [390].

Angiogenesis is one of the hallmarks for the progression of cancer cells, defined as the formation of new blood vessel to provide adequate oxygen and nutrients [391]. The anti-angiogenesis property of T3 has been elucidated in several studies outlined below. A recent study by Husain et al. pinpointed that δT3 inhibited biomarkers of tumor angiogenesis (VEGF and MMP-9) in pancreatic cancer cells (MiaPaCa-2 and L3.6pl). In female athymic nude mice implanted with luciferase-expressing L3.6pl cells, oral administration of 200 mg/kg δT3 for 4 weeks reduced cluster of differentiation 31 (CD31) expression, an indicator for the degree of tumor angiogenesis [392]. Hypoxia promotes angiogenesis, as the proliferating tumor cells require new vascular network for sustained delivery of oxygen [393]. Hypoxia-inducible factor (HIF)-1 facilitates the mitogenic and migratory activities of endothelial cells, thus is an important regulator of hypoxia-induced angiogenesis [394]. Significant reductions of HIF-1α and HIF-2α were detected in prostate cancer stem cells (PC-3) following exposure to δT3, however, with an unclear mechanism of action [372].

The spreading of cancerous tumor from the primary to secondary site is termed as tumor metastasis. The anti-tumor effects of T3 have been reported in part mediated through the suppression of metastatic phenotype. The growth inhibitory effects of γT3 on breast cancer cells (MDA-MB-231 and T-47D) were attributable to the suppression of Wnt signaling, reversal of epithelial-to-mesenchymal transition (EMT) and significant reduction in cancer cell migration [355]. Another study was performed using mouse +SA and human MDA-MB-231 breast cancer cell lines. The authors deciphered that γT3 caused significant reductions in cell invasion and migration. The suppression of Ras-related C3 botulinum toxin substrate 1 (Rac1)/Wiskott–Aldrich Syndrome protein-family verprolin-homologous protein 2 (WAVE2) signaling was proposed as the possible underlying mechanism (characterized by reduced expression of WAVE2, Arp2, Arp3, and reduced phosphorylation of Rac1/Cdc42) [395]. All these signaling molecules are essential for cytoskeletal reorganization, formation of membrane protrusions and raffles that promote cancer cell mobility [396]. The unknown mechanism of action in T3-inhibited Rac1/WAVE2 signaling renders further investigation.

A wide array of preclinical scientific evidence strongly supports the notion on the suppressive effects of T3 on cancer cell growth, proliferation, migration, and invasion. The potential anti-cancer property of T3 may be accredited to its ability to promote apoptosis and cell cycle arrest as well as inhibit angiogenesis and metastasis. Targeting all these mechanisms of action may be an important approach in searching cancer therapy (Figure 11). Nonetheless, there are several potential challenges in the effort of discovering anti-cancer drugs. Firstly, treatment of cancer is complicated by drug resistance. Hypoxia has been shown to influence stem-like cells properties of cancer cells which lead to multidrug resistance in leukemic cells [397]. Transcription factor HIF-1α has been found to be activated in hypoxia condition which lead to regulation of drug transporter, ATP-binding cassette super-family G member 2 (ABCG2), and implicated in drug resistance [398]. Secondly, T3 may act differently when exist in combination with other compounds due to drug interaction. For instance, γT3 interferes with the generation of ROS by BSO which is beneficial for cancer cell death [399].

Despite the abundance of cell and animal studies investigating the anticancer role of T3 [400], evidence on the effects of T3 on cancer is limited and most trials are at the preliminary stage. In a Phase 1 clinical trial involving patients (n = 25) with pancreatic ductal neoplasia, γT3 (200–3200 mg for 13 days prior to surgery and one dose on the day of surgery) was shown to be safe and effective in increasing the cleaved caspase caspase-3 level in the patients, indicating apoptosis of the cancer cells [401]. In a Phase II clinical trial involving patients with ovarian cancers (n = 23), the combination of γT3 (900 mg daily) and bevacizumab increased the rate of disease control, progression free survival and overall survival significantly [402]. In a Phase III pilot clinical trial, the combination of TRF and tamoxifen did not increase the breast cancer specific survival and 5-year disease free survival, nor reduce cancer mortality significantly of women with tumor node metastases stage I or II breast cancer and oestrogen receptor positive tumors (n = 240) [403]. Hence, more clinical trials are warranted to examine the safety, efficacy, and tolerability of T3 as a novel chemopreventive or adjuvant agent.

## 14. Anti-Inflammatory Properties of Tocotrienol

Inflammation has a close relationship with non-communicable diseases addressed in this review. It is a complex immune response playing a role to protect the body against harmful assaults. However, the outcome of chronic inflammation is deleterious causing the underlying injury. The capability of T3 in modulating inflammation has been extensively explored by researchers. T3 acts as regulator for the expression of pro- and anti-inflammatory mediators including TNF-α, IL-1, IL-2, IL-6, IL-8, IL-10, IL-23, IFN-γ, iNOS, and COX-2 [8,404]. The most implicated signal transduction pathways involved in orchestrating inflammatory response are NF-κB, MAPK, signal transducer and activator of transcription (STAT) and Toll-like receptor (TLR) signaling [8,405]. Docking studies showed that αT3 blocks the assess of arachidonic acid to the catalytic site of 12-lipoxygenase, thereby achieving its anti-inflammatory effects [406]. From electrophoretic mobility shift assay, γT3 was the most potent NF-κB inhibitor compared to other T3 isoforms [312]. However, further investigation is required to confirm the structural role of chromanol ring in NF-κB inhibition.

In addition, recent studies reported that metabolites of vitamin E, like 13-((2R)-6-hydroxy-2,5,7,8-tetramethylchroman-2-yl)-2,6,10-trimethyltridecanoic acid (α-T-13′-COOH), possessed inhibitory activities on 5-lipoxygenase, and production of inflammatory mediators derived from it in vitro and in vivo, subsequently suppressed inflammation and hyperactivity in mouse models of peritonitis and asthma [28]. Another study showed that long-chain carboxychromanols resulted from vitamin E metabolism inhibited the activities of COX. In particular, 13′-carboxychromanol bound more strongly with COX-1 compared to 9′-carboxychromanol and exerted more potent COX-inhibitory effects [407]. Another vitamin E metabolite, α-(13′-hydroxy)-6-hydroxychroman inhibited inflammation of RAW264.7 murine macrophages by regulating activities of COX and iNOS and transcription of inflammatory cytokines [408]. To what extent these vitamin E metabolites contribute to the anti-inflammatory activities and protective effects of T3 or vitamin E in general remains elusive.

Considering the promising potential of T3 against various medical complications as well as its potent anti-inflammatory properties, the current state of knowledge deserves further elucidation on the molecular action of T3. Such effort is essential for developing potential drug targets for alleviating inflammation in NCDs.

## 15. Comparison between the Effects of Tocotrienol and Tocopherol

Vitamin E possesses protective effects on a variety of diseases in varying degree. The differences can be observed among each isomer of vitamin E. HMGCR inhibition is one of the biological actions of T3 which could not be observed with TF, suggesting the crucial role of phytyl tail in this action. As discussed in the earlier section, inhibition of HMGCR by T3 was following the order of δ > γ > β > α from an in-silico docking analysis [45]. Song et al. had identified that only γT3 and δT3 potently mimicked 25-hydroxycholesterol in stimulating degradation and ubiquitination of HMGCR and other vitamin E isoforms, including TF, had no effect at all [44]. Additionally, δT3 induced both HMGCR degradation and interference of SREBP processing while γT3 is more selective in promoting HMGCR degradation than blocking SREBP processing [44]. The mechanism underlying δT3 and γT3-mediated SREBP processing is currently unknown [44] which may be due to the structural differences of T3s.

Antioxidant activity is one of the main molecular actions of T3 in protecting against NCDs. Mechanistically, both T3 and TF could scavenge the free radicals directly by donating the phenolic hydrogen of the chromanol ring [409,410]. T3 has better membrane antioxidant activity as compared to TF, which could due to the uniform distribution of T3 in the membrane bilayers and stronger disordering of membrane lipids structure [411]. A study from Palozza et al. showed that δT3 exerted greater inhibition on lipid peroxidation and ROS production than γT3 and αT3 [202]. This could due to the decreased methylation of the chromanol ring in δT3, which allow better incorporation of T3 into the cellular membranes [202].

The hypocholesterolemic actions between T3 and TF were compared in several studies. Raederstorff et al. (2002) provided evidence that γT3 had higher efficacy than mixed T3 (containing 9.9% γTF) in lowering TC and LDL-C [62]. It was also found that αTF is less effective in improving lipid metabolism as compared to TRF (containing a mixture of T3 and TF), γT3, and δT3 [41,53]. In humans, T3 but not αTF reduced TC and LDL-C in subjects with mild hypercholesterolemia with one additional cardiovascular risk factor [70]. The distinct effects of T3 and TF on lipid metabolism might be elicited through their differential regulatory role in the suppression of HMGCR in mevalonate pathway [44]. The plausible explanation is that the absence of three double bonds in the phytyl tail of TF gives rise to its inferior hypocholesterolemic action [412].

In CVD, the results obtained by Das and colleagues illustrated that γT3 exhibited the best cardioprotective effect in myocardial ischemia-reperfusion injury model, followed by αT3 and δT3 [96,97]. Recently, Wong et al. (2017) compared the effects of αTF, αT3, γT3 and δT3 in rats fed with HCHF diet to induce MetS. Among these isomers, δT3 was capable of improving more metabolic abnormalities than γT3, but αT3 and αTF were the least effective [103]. Using a hypertensive rat model, Muharis et al. (2010) revealed that the effects of T3 were as effective as TF in promoting radical scavenging capability and acetylcholine relaxation in aortic ring [104]. Intravenous (i.v.) administration of αTF, αT3, or γTF was able to reduce infarct volume induced by middle cerebral artery occlusion (MCAO) better than γT3, δTF, and δT3.

With regards to bone health, previous studies showed that two different forms of vitamin E (palm T3 mixture and αTF) were comparable in preserving bone microarchitecture and BMD in postmenopausal osteoporosis rat model [134,136]. Otherwise, some studies reported better outcomes on the skeleton when normal [144,157] or osteoporotic animals [146,164] were supplemented with T3 compared to αTF. In fractured rat model, T3 was also better than αTF in improving bone strength of the fractured callous in oestrogen-deficient rats [142]. In another experimental model, supplementation of PVE but not αTF alone maintained bone calcium content in femur and fifth lumbar of normal rats provided with VED diet [133]. Mechanistically, T3 was shown to have more potent anti-inflammatory [146] and anti-oxidative [159,160] properties, which were responsible for its superior skeletal-promoting effects in animals.

The ability of PVE, TRF, and αTF to protect against gastric mucosal injury induced by stress and NSAIDs were not significantly different [223,240,242,252]. The gastroprotective effects of vitamin E were modulated through the reduction of inflammatory response and oxidative stress, whereby T3 had more prominent anti-inflammatory [215] and anti-oxidative [224] properties than TF. For neurodegenerative diseases, some in vitro studies reported similar cytoprotective effects between T3 and TF against glutamate injury [276,279]. Whereas in an in vivo study, Nakaso et al. (2016) evaluated the cytoprotective effects of αT3 and δT3 equally conferred protection in preventing neurotoxicity and motor deficit in the MPTP-induced mouse model of Parkinson’s disease, but they were more effective than αTF and δTF [260].

In terms of metabolic regulation, Kuhad and co-authors found that T3 was more effective than αTF at similar dose of 100 mg/kg in normalizing body weight and glucose level in STZ-induced diabetic rats [323,324]. These observations might be attributed to the higher efficacy of T3 in reducing inflammatory response and lipid peroxidation [323,324]. Another group of researchers further confirmed that palm TRF with higher content of T3 (77%) had greater efficiency in reducing FBG, HbA1c, TC, and LDL-C via suppression of lipid peroxidation and increase of anti-oxidant activities in STZ-induced diabetic rats as compared to rice bran TRF with lower content of T3 (60.8%) [325,327]. Besides, αT3, γT3, and δT3 could improve the insulin synthesis, wherein δT3 is the most potent isoforms, followed by γT3 and αT3 [340]. In-silico docking analysis and reporter assay revealed that α-T3, γ-T3, and δ-T3 serve as PPARα agonist and only δ-T3 could activate PPARδ and PPARγ by exhibiting a higher binding affinity [339]. Additionally, the anti-adipogenic activities of each T3 were reported by following the rank order of γ > δ > β > α [331,332]. The upstream molecular mechanism of anti-adipogenic activities of T3 is remains elusive, however the anti-adipogenic effect of δT3 was found to be independent of HMGCR inhibition [336].

In anticancer studies, T3 displayed significantly greater anticancer activities than their corresponding TF isoforms [413]. Structural-activity relationship study revealed the importance of chromanol ring and phytyl carbon tail in cancer cell apoptosis induction [414]. The absence of methyl groups on the chromanol ring was associated with an increase of anti-proliferative potency of T3s, with the order of δ > γ> α [413]. T3s were reported to cross the phospholipid bilayers readily and accumulate intracellularly via an α-TTP-independent manner [25,415,416]. Each T3 isoform had different cellular accumulation tendency with the similar order of δ > γ > α [366,367,417,418] which might explain the poor anticancer properties of αT3.

Taken together, vitamin E displays beneficial properties in preventing various NCDs in general even though the distinct effects between the isomers of T3 and TF do exist. The plausible explanation could be the variation of dose, composition, treatment duration, route of administration, experimental model, and study design between studies.

## 16. Safety of Tocotrienol and Tocopherol

Vitamin E is relatively safe. It has been reported in animal studies that vitamin E is not mutagenic, carcinogenic, or teratogenic [419]. A 13-week study was performed using male and female Fischer rats to explore the oral toxicity of T3. The T3 preparation given is composed of 21.4% αT3, 3.5% βT3, 36.5% γT3, 8.6% δT3, 20.5% αTF, 0.7% βTF, 1.0% γTF, and 0.5% δTF. In this study, the no-observed-adverse-effect level (NOAEL) of T3 was concluded as 120 mg/kg for male rats and 130 mg/kg for female rats [43]. Subacute and subchronic toxicity assessment of PVE reported that at dose ≥500 mg/kg, the bleeding and clotting time of the mice were increased [420]. Although the dose is well above the usual supplemental dose, the concurrent use of T3 and TF with other anticoagulants should be cautioned. This is not a new finding because vitamin E supplementation has been associated with hemorrhagic stroke in a meta-analysis [421]. Another human study indicated that oral consumption of high dose of vitamin E (3200 IU/day) is well-tolerated by adults [419]. Since vitamin E, αTF, and mixed T3 are widely available as over-the-counter supplements, it is hard to regulate their use. It is important for medical professions to recommend the appropriate doses for vitamin E based on scientific evidence, taking into account the medical conditions and concurrent medication or supplement use of the consumers. Although vitamin E showed promising health benefits, it should not replace standard medical care. Consultation from a medical practitioner is recommended, particularly for the long-term use of vitamin E.

## 17. Research Gaps and Future Perspective

Several research gaps remain to be filled by investigators. Since overwhelming inflammatory response and oxidative stress are the common causative factors for these NCDs, the underlying mechanisms involved in governing these conditions should be targeted to develop potential adjunct therapy. For instance, toll-like receptor (TLR) and nuclear factor erythroid 2-related factor 2 (Nrf2) signaling are the most implicated mechanisms involved in the induction of inflammation and oxidative stress respectively. Hence, the role of T3 in these pathways should be investigated. Apart from that, there is a lack of human trial on the effects of T3 in most of the non-communicable diseases including arthritis, sarcopenia, peptic ulcer, Alzheimer’s disease, and wound healing. Clinical trials are only available for hyperlipidemia, CVD, osteoporosis, Parkinson’s disease, diabetes, and cancers. There is also a lack of conclusive evidence on the beneficial effects T3 in these trials. Further placebo-controlled trials are recommended to justify the direct effects of T3 supplementation in patients with non-communicable diseases. As aforementioned, different isoforms of T3 and TF exert different effects in different diseases, which may be due to their chemical and physical properties. The potency of individual vitamin E isoforms needs to be further characterized, particularly in both preclinical and clinical settings. Further investigations are also required to elucidate the upstream molecular mechanism underlying initial biological activities of individual vitamin E isoforms. This effort aims to evaluate the best regimen of T3 and TF—either in single isomer or combination—in preventing these diseases with greater potency, efficacy, and effectiveness.

## 18. Conclusions

Tocotrienol can tackle many aspects of NCDs, mainly by alleviating inflammatory response, oxidative stress, mitochondrial dysfunction, and abnormal cholesterol synthesis. The anti-hyperlipidemic, anti-osteoporotic, anti-hyperglycemic, anti-inflammatory, anti-oxidative, neuroprotective, gastroprotective, and cardioprotective effects of T3 seem to be more superior or as effective compared to TF. Further validation on the clinical utilities of T3 and TF entails much effort from researchers.

## Figures and Tables

**Figure 1 nutrients-12-00259-f001:**
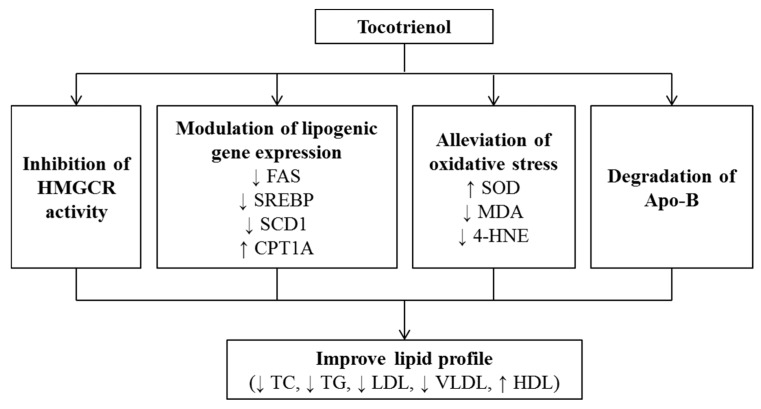
Effects of T3 on lipid metabolism.

**Figure 2 nutrients-12-00259-f002:**
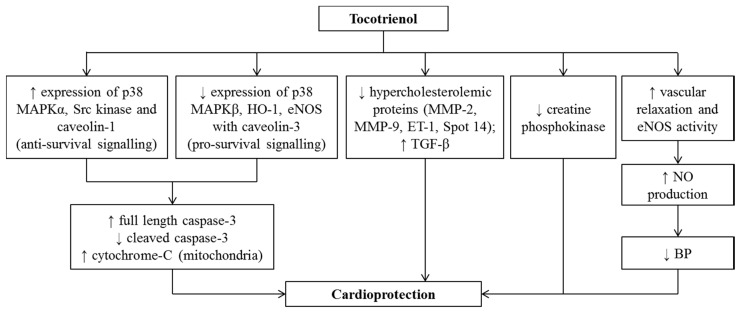
Cardioprotective effects of T3.

**Figure 3 nutrients-12-00259-f003:**
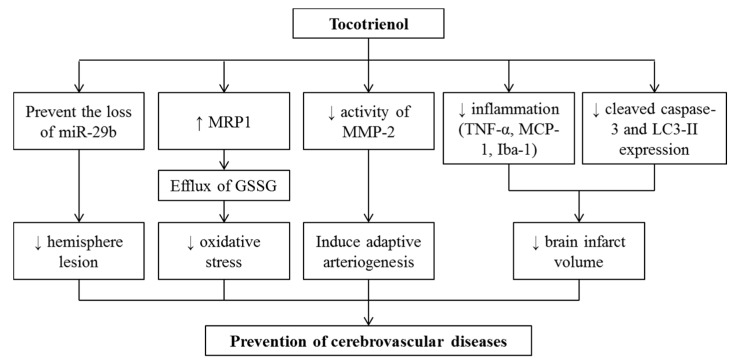
Effects of T3 on cerebrovascular diseases.

**Figure 4 nutrients-12-00259-f004:**
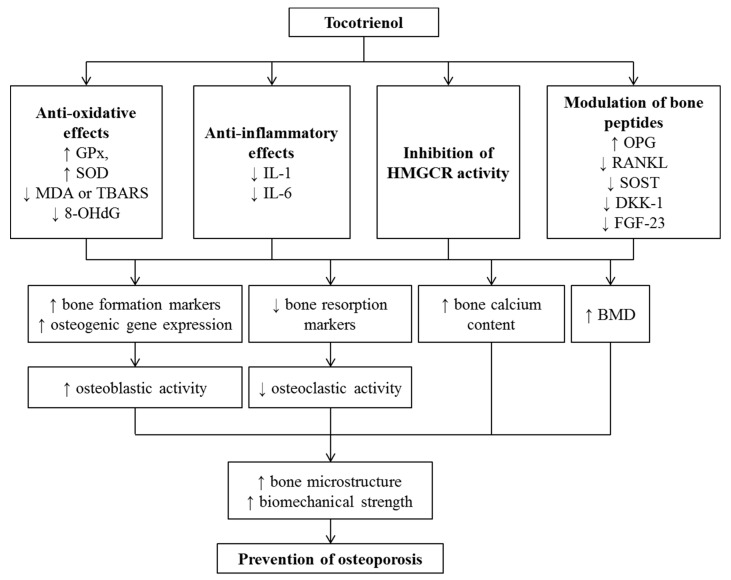
Bone-protecting property of T3.

**Figure 5 nutrients-12-00259-f005:**
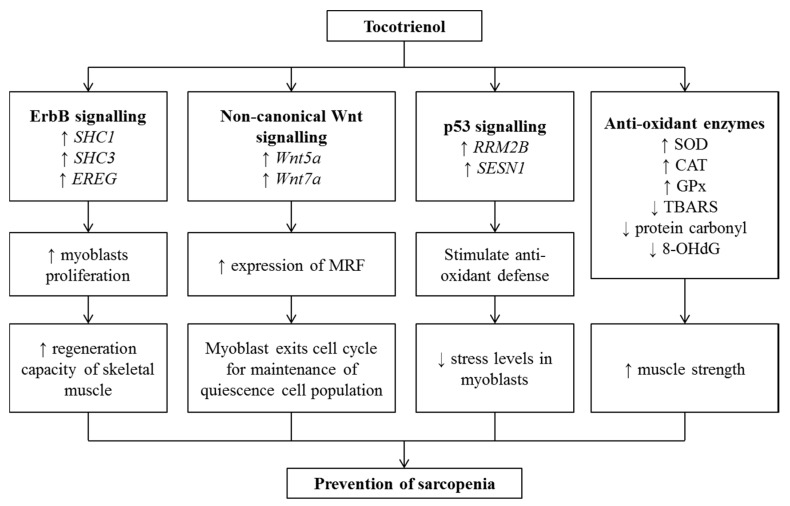
Effects of T3 on muscle health.

**Figure 6 nutrients-12-00259-f006:**
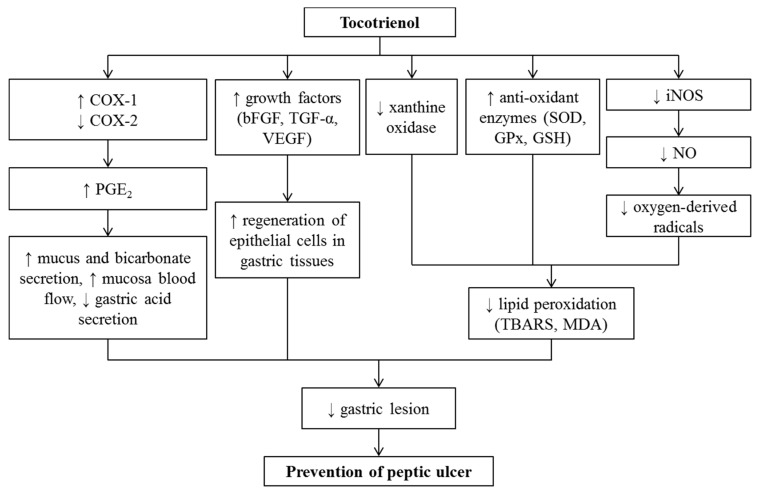
Effects of T3 on peptic ulcer.

**Figure 7 nutrients-12-00259-f007:**
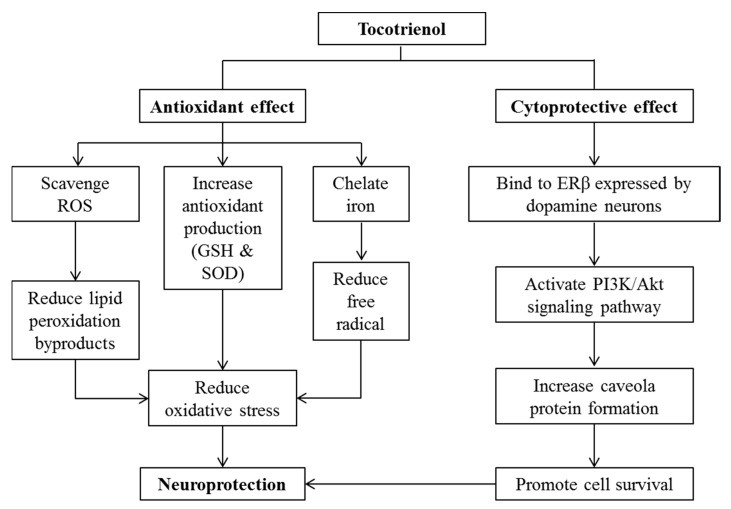
Neuroprotective effect of T3 on Parkinson’s disease.

**Figure 8 nutrients-12-00259-f008:**
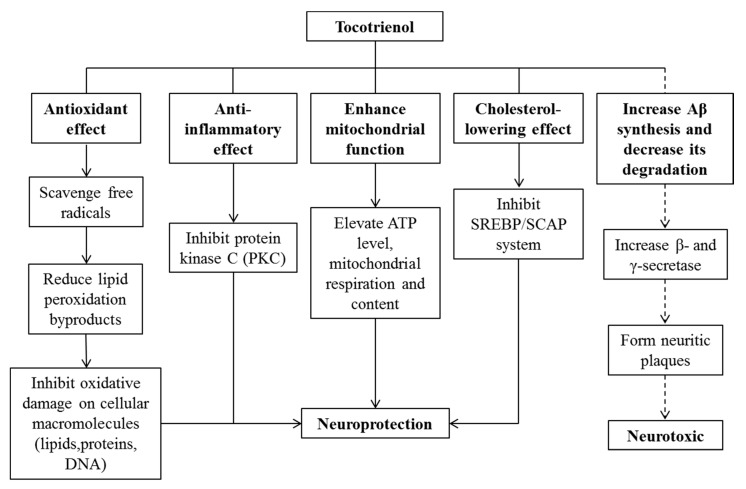
Effects of T3 on Alzheimer’s disease. Solid lines represent advantageous effects of T3 while the dotted lines represent possible harmful effects of T3.

**Figure 9 nutrients-12-00259-f009:**
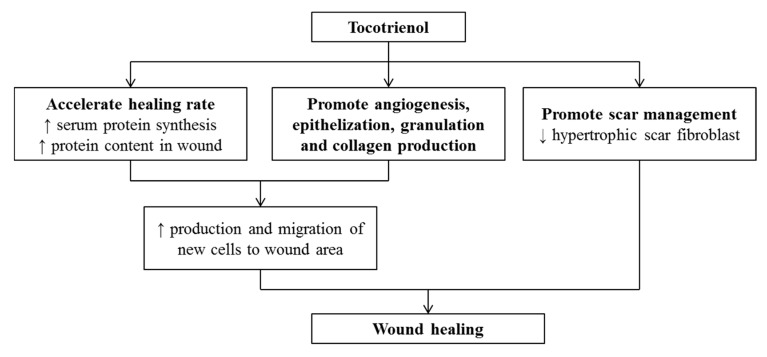
Effects of T3 on wound healing.

**Figure 10 nutrients-12-00259-f010:**
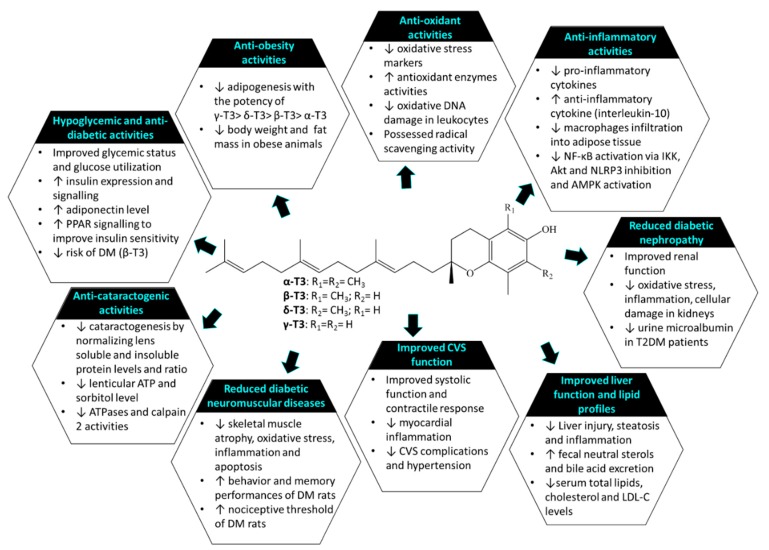
Beneficial effects of T3 on obesity and diabetes.

**Figure 11 nutrients-12-00259-f011:**
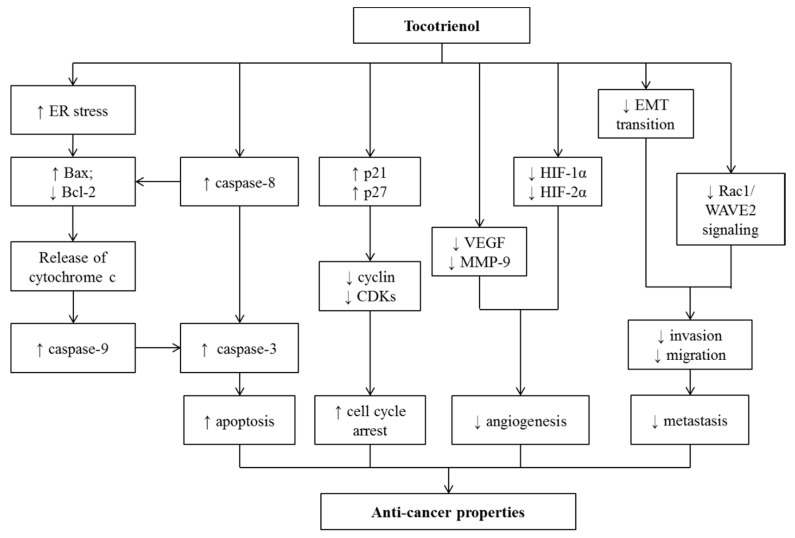
Anti-cancer mechanism of T3.

**Table 1 nutrients-12-00259-t001:** Effects of T3 on lipid metabolism in vitro.

Researcher (Year)	Cell Type	Treatment & Dose	Treatment Duration	Findings
Parker et al. (1993) [39]	Mammalian cells (HepG2 cells) supplemented with 7% lipoptotein-depleted serum in RPMI 1640 for 16 h to induce cholesterol genesis	T3 (10 µM)	4 h	HMGCR: ↓
Burdeos et al. (2013) [40]	Hepa 1-6 cells	γT3 (15 µM)	24 h	TG: ↓, FAS: ↓, SREBP: ↓, SCD1: ↓, CPT1A: ↑, TC: ↔
Burdeos et al. (2012) [41]	HepG2 cells	γT3 (10–15 µM)	24 h	Cellular accumulation of TG (especially in the fat-loaded medium): ↓
Song & DeBose-Boyd (2006) [44]	SV 589 cells (an immortalized line of human fibroblasts expressing SV-40 large T-antigen)	αT3 (10 µM)	5 h	Did not have the function as γT3 and δT3
		γT3 (10 µM)		More selective than δT3
		δT3 (10 µM)		Ubiquitination and degradation of HMGCR: ↑, SREBP processing: ↓
Theriault et al. (1999) [46]	HepG2 cells (HB 8065) supplemented with 7% lipoptotein-depleted serum in RPMI-1640 for 16 h to induce cholesterol genesis	γT3	6 h	Apo-B degradation: ↑, Apo-B translocation into the ER lumen: ↓, number of secreted Apo-B containing lipoprotein particles: ↓
Zaiden et al. (2010) [47]	HepG2 cells supplemented with 7% lipoptotein-depleted serum in RPMI-1640 for 16 h to induce cholesterol genesis	γδT3 (20 µM)	16 h	Diacylglycerol O-acyltransferase 2 (DGAT2), APOB100, SREBP1/2 and HMGCR: ↓, biosynthesis of TG, TC and VLDL: ↓, efflux of LDL through LDL receptor expression: ↑

HMGCR, 3-hydroxy-3-methylglutaryl-coenzyme A reductase; FAS, fatty acid synthase; SREBP, sterol regulatory element-binding protein; SCD1, stearoyl-coenzyme A desaturase 1; CPT1A, carnitine palmitoyltransferase 1A; SOD, superoxide dismutase; MDA, malondialdehyde; 4-HNE, 4-hydroxynonenal; TC, total cholesterol; TG, Triglyceride; LDL, low-density lipoprotein; VLDL, very-low-density lipoprotein; HDL, high-density lipoprotein.

**Table 2 nutrients-12-00259-t002:** Effects of T3 on lipid metabolism in vivo.

Researcher (Year)	Animal Model	Treatment & Dose	Mode	Treatment Duration	Findings
**Chicken model**					
Qureshi et al. (1996) [48]	White Leghorn female chicken (*n* = 10, 2 weeks old)	120 nmol/g αTF	Oral	26 days	αTF attenuated the inhibition of HMGCR by γT3 Effective preparation of T3 mixture to exhibit anti-cholesterol effects contains 15–20% αTF and approximately 60% γT3 or δT3 Less effective preparation contains 30% αTF and approximately 45% γT3 or δT3
		γT3 (60 nmol/g) + αTF (60 nmol/g)			
		γT3 (72 nmol/g) + αTF (48 nmol/g)			
		γT3 (84 nmol/g) + αTF (36 nmol/g)			
		γT3 (96 nmol/g) + αTF (24 nmol/g)			
		γT3 (120 nmol/g)			
Qureshi & Peterson (2001) [49]	White Leghorn female chickens (*n* = 32, 2 weeks old)	TRF_25_ (d-P_21_-T3 + d-P_25_-T3) (50 ppm)	Oral	4 weeks	TC: ↓, TG: ↓, LDL-C: ↓, HDL-C: ↓, HMGCR activity: ↓, ApoA-1: ↔, Apo-B: ↓, thromboxane B_2_: ↓, platelet factor 4: ↓
		TRF_25_ (d-P_21_-T3 + d-P_25_-T3) (50 ppm) + lovastatin (50 ppm)			TC: ↓, TG: ↓, LDL-C: ↓, HDL-C: ↔, HMGCR activity: ↓, ApoA-1: ↔, Apo-B: ↓, thromboxane B_2_: ↓, platelet factor 4: ↓
Yu et al. (2006) [53]	White Leghorn female chickens fed with cholesterol-free corn soy diet (5 weeks old)	TRF, γT3 or δT3 (50–2000 ppm)	Oral	4 weeks	TC: ↓, LDL-C: ↓
		αTF (50–2000 ppm)			TC: ↔, LDL-C: ↔
		αT3 (50–500 ppm)			TC: ↓, LDL-C: ↓
Qureshi et al. (2011) [50]	White Leghorn female chicken fed with corn-soy diet (*n* = 24; one day old)	δT3 (50 ppm/kg diet)	Oral	4 weeks	TNF-α: ↓, NO: ↓, TC: ↓, LDL-C: ↓, lipid elevating impact of dexamethasone: ↓, TG lowering impact of riboflavin: ↑
Hansen et al. (2015) [26]	48 Hy-Line W-36 laying hens	2000 mg/kg T3	Oral	7 weeks	TC: ↔, HDL-C: ↔, TG: ↔
		2000 mg/kg T3 + 200 mg/kg αTF			
		2000 mg/kg T3 + 1000 mg/kg αTF			
**Swine model**					
Qureshi et al. (1991) [51]	Normolipemic swine	TRF (10–20% αTF, 15–20% αT3, 30–35% γT3, 20–25% δT3) (50 μg/g)	Oral	42 days	TC: ↓, HDL-C: ↔, LDL-C: ↓, ApoA-1: ↔, Apo B: ↓, thromboxane B_2_: ↓, platelet factor 4: ↓, HMGCR activity in adipose tissue: ↓
	Genetically hypercholesterolemic				
Qureshi et al. (2001) [52]	Genetically hypercholesterolemic swine	TRF_25_ (50 µg)	Oral	6 weeks	TC: ↓, HDL-C: ↔, LDL-C: ↓, ApoA-1: ↔, Apo B: ↓, TG: ↓, thromboxane B_2_: ↓, platelet factor 4: ↓, glucose: ↓, glucagon: ↓, insulin: ↑, HMGCR activity: ↓, cholesterol 7α-hydroxylase: ↔
		γT3 (50 µg)			
		d-P_21_-T3 (50 µg)			
		d-P_25_-T3 (50 µg)			
**Rat model**					
Watkins et al. (1993) [54]	Male Wistar rats on atherogenic diet	γT3 (50 mg/kg)	Oral	6 weeks	TC: ↓, LDL-C: ↓, VLDL: ↓, TG: ↓, TBARS: ↓, fatty acid hydroperoxides: ↓
		αTF (500 mg/kg)			
Kaku et al. (1999) [55]	Male Sprague-Dawley rats (4 weeks old)	Mixed T3 (18.9% αT3, 5.5% βT3, 55.9% γT3, 16.7% δT3) (0.002% of diet)	Oral	3 weeks	TG: ↓, TC: ↔, phospholipids: ↔
Iqbal et al. (2003) [59]	Rats treated with chemical carcinogen DMBA to induce mammary carcinogenesis and hypercholesterolemia	TRF (10 mg/kg)	Oral	6 months	TC: ↓, LDL-C: ↓, HMGCR activity & protein mass: ↓, severity & extent of neoplastic transformation in mammary glands: ↓, plasma & mammary ALP activity: ↓, GST activity: ↓
Minhajuddin et al. (2005) [56]	Male albino rats fed with atherogenic diet	TRF (0–50 mg/kg)	Oral	1 weeks	TG: ↓, TC: ↓, LDL: ↓, TBARS: ↓, conjugated dienes: ↓, HMGCR: ↓, optimum dose: 8 mg/kg
Budin et al. (2009) [58]	Male Sprague-Dawley rats induced by STZ	TRF (200 mg/kg)	Oral	8 weeks	Glucose: ↓, HbA1c: ↓, TC: ↓, LDL-C: ↓. TG: ↓, HDL-C: ↑, SOD: ↑, MDA & 4-HNE (plasma and aorta): ↓, DNA damage: ↓
Zaiden et al. (2010) [47]	LDLr-deficient mice (strain B6; 129s7-Ldlr^tm1Her^/J, LDLr^−/−^)	γδT3 (50 mg/kg)	Oral	4 weeks	TC ↓, TG: ↓, HDL-C: ↔, LDL-C: ↓
		γδT3 + αTF (50 mg/kg)			
Burdeos et al. (2012) [41]	Male F344 rats fed with high fat diet	αTF (10 mg/kg)	Oral	3 weeks	TG, TC, phospholipid & PLOOH (liver & plasma): ↔
		Rice bran T3 (1.9% αTF, 2.1% γTF, 2.0% δTF, 31.4% αT3, 50.5% γT3, 0.4% δT3) (5 mg/day)			TG, TC, phospholipid & PLOOH (liver & plasma): ↔
		Rice bran T3 (1.9% αTF, 2.1% γTF, 2.0% δTF, 31.4% αT3, 50.5% γT3, 0.4% δT3) (10 mg/day)			TG, phospholipid & PLOOH (liver & plasma): ↓, TC (liver & plasma): ↔
Cheng et al. (2017) [57]	Male post-weaning Sprague-Dawley rats fed with high fat diet (8 weeks) to induce MetS	TRF (60 mg/kg)	Oral	4 weeks	TC: ↓, non-HDL-C: ↓, TG: ↔
**Guinea pigs**					
Khor et al. (1995) [60]	Male albino Harley guinea pigs	T3 isolated from palm fatty acid distillate (5 or 8 mg/day)	i.p.	6 days	HMGCR: ↓
		T3 isolated from palm fatty acid distillate (10 mg/day)			HMGCR: ↔
Khor & Ng (2000) [63]	Male albino guinea pigs	T3 (10 mg/kg/day)	i.p.	6 days	HMGCR: ↓
		T3 (10 mg/kg/day) + αTF (5 mg/kg/day)			HMGCR: ↓ (lesser)
Khor et al. (2002) [64]	Male and female albino Harley guinea pigs	PVE (1 mg T3) (200 µL)	i.p.	6 days	HMGCR: ↓, cholesterol 7α-hydroxylase: ↓
		PVE (3 mg T3) (200 µL)			
		PVE (5 mg T3) (200 µL)			
		PVE (1 mg αTF) (200 µL)			HMGCR: ↓, cholesterol 7α-hydroxylase: ↓ (less effective than T3)
		PVE (3 mg αTF) (200 µL)			
		PVE (5 mg αTF) (200 µL)			HMGCR: ↓, cholesterol 7α-hydroxylase: ↓
		PVE (20 mg αTF) (200 µL)			HMGCR: ↔, cholesterol 7α-hydroxylase: ↔
		PVE (50 mg αTF) (200 µL)			HMGCR: ↑
Khor & Chieng (1997) [61]	Male Golden Syrian hamsters	Palm oil triacylglycerol fraction +72 ppm TF	Oral	45 days	TC: ↑, LDL-C: ↑, HDL-C: ↑
		POTG + 162 ppm T3			TC: ↓, LDL-C: ↔, HDL-C: ↔, TG: ↔
		POTG + 1000 ppm T3			TC: ↑, LDL-C: ↑, HDL-C: ↑
Raederstorff et al. (2002) [62]	Male Golden Syrian hamsters	Mixed T3 [29.5% αT3, 3.3% βT3, 41.4% γT3, 0.1% δT3, 25.1% others (mainly 9.9% γTF)] (39 mg/kg/day)	Oral	4 weeks	Plasma TC: ↓, LDL-C: ↓ (starting from 2 weeks), TG: ↔, HDL-C: ↔
		Mixed T3 [29.5% αT3, 3.3% βT3, 41.4% γT3, 0.1% δT3, 25.1% others (mainly 9.9% γTF)] (263 mg/kg/day)			Plasma TC: ↓, LDL-C: ↓ (reduced significantly after 4 weeks), TG: ↔, HDL-C: ↔
		γT3 [0.6% αT3, 7.0% βT3, 86.1% γT3, 0.1% δT3, 6.2% others (mainly 1.2% γTF)] (23 mg/kg/day)			Plasma TC: ↔, LDL-C: ↔, TG: ↔
		γT3 [0.6% αT3, 7.0% βT3, 86.1% γT3, 0.1% δT3, 6.2% others (mainly 1.2% γTF)] (58 mg/kg/day)			Plasma TC: ↓, LDL-C: ↓ (reduced significantly after 4 weeks), TG: ↔, HDL-C: ↔
		γT3 [0.6% αT3, 7.0% βT3, 86.1% γT3, 0.1% δT3, 6.2% others (mainly 1.2% γTF)] (263 mg/kg/day)			Plasma TC: ↓, LDL-C: ↓ (reduced significantly after 4 weeks), TG: ↔, HDL-C: ↔
Khor & Ng (2000) [63]	Male hamsters	30 ppm αTF	Oral	45 days	HMGCR: ↓
		81 ppm αTF			HMGCR: ↑
Salman Khan et al. (2011) [45]	Male Golden hamsters	Tocomin (6.4% αT3, 1.0% βT3, 10.2% γT3, 3.2% δT3, 5.7% αTF) (10 mg/day)	Oral	10 days	Plasma lipids: ↓, lipoprotein lipids: ↓, TC: ↓, Apo-B: ↓, small dense-LDL: ↓, LDL-C: ↓, inhibition of HMG-CoA reductase: δT3 > γT3 > βT3 > αT3

**Table 3 nutrients-12-00259-t003:** Effects of T3 on lipid metabolism in humans.

Researcher (Year)	Study Population	Treatment & Dose	Mode	Treatment Duration	Findings
**Positive effect**					
Qureshi et al. (1991) [65]	Hypercholesterolemic subjects (*n* = 25; aged 30–60 years)	PVE (15–20% TF, 12–15% T3, 35–40% γT3, 25–30% δT3) (200 mg/day)	Oral	4 weeks	TC: ↓, LDL-C: ↓, Apo-B: ↓, thromboxane B_2_: ↓, platelet factor 4: ↓, glucose: ↓
Qureshi et al. (1995) [66]	Hypercholesterolemic subjects on AHA Step-1 diet for 4 or 8 weeks (*n* = 36; aged 40.6 ± 12.1 years)	PVE (40 mg αTF, 48 mg αT3, 112 mg γT3, 60 mg δT3) (220 mg)	Oral	4 weeks	TC: ↓, ApoA-1: ↔, HDL-C: ↔
		γT3 (200 mg)	Oral	4 weeks	TC: ↓, Apo-B: ↓, ex vivo thromboxane B_2_: ↓, ApoA-1: ↔, HDL-C: ↔
Qureshi et al. (2001) [67]	Hypercholesterolemic subjects on AHA Step-1 diet since days 36–70) (*n* = 14; aged 40.14 ± 8.98 years)	Lovastatin (10 mg/day)	Oral	Day 71–105	TC ↓, LDL-C: ↓, Apo-B: ↓, TG: ↓
		Lovastatin (10 mg/day) + TRF (50 mg/day)		Day 106–140	TC: ↓, LDL-C: ↓, Apo-B: ↓, TG: ↓
		Lovastatin (10 mg/day) + αTF (50 mg/day)		Day 141–175	TC: ↔, LDL-C: ↔, HDL-C/LDL-C ratio: ↑
	Hypercholesterolemic subjects on AHA Step-1 diet since days 36–70) (*n* = 14; aged: 42.57 ± 6.36 years)	TRF_25_ (50 mg/day)	Oral	Day 71–105	TC ↓, LDL-C: ↓, Apo-B: ↓, TG: ↓
		Lovastatin (10 mg/day) + TRF_25_ (50 mg/day)		Day 106–140	TC: ↓, LDL-C: ↓, Apo-B: ↓, TG: ↓
		Lovastatin (10 mg/day) + αTF (50 mg/day)		Day 141–175	TC: ↔, LDL-C: ↔, HDL-C/LDL-C ratio: ↑
Qureshi et al. (2002) [68]	Hypercholesterolemic subject on AHA Step-1 diet starting day 36 (*n* = 90; men: aged <50 years, women: aged <40 years)	TRF_25_ (25 mg/day)	Oral	35 days	TC: ↓, LDL-C: ↓
		TRF_25_ (50 mg/day)			TC: ↓, LDL-C: ↓, Apo-B: ↓, TG: ↓, Apo A-1: ↑, HDL-C: ↑
		TRF_25_ (100 mg/day)			TC: ↓, LDL-C: ↓, Apo-B: ↓, TG: ↓ (maximum decrease), Apo A-1: ↑, HDL-C: ↑
		TRF_25_ (200 mg/day)			TC: ↔, LDL-C: ↔, Apo-B: ↔, TG: ↔
Zaiden et al. (2010) [47]	Hypercholesterolemic subjects (*n* = 20; aged 25–55 years)	γδT3 (120 mg/day)	Oral	8 weeks	TG: ↓, VLDL: ↓, chylomicron: ↓. TC: ↔, LDL-C: ↔, HDL-C: ↔
Yuen et al. (2011) [69]	Hypercholesterolemic subjects (*n* = 32; aged 31–53 years)	Mixed T3 (22.9 IU αTF, 30.8% αT3, 56.4% γT3, 12.8% δT3) (300 mg/day)	Oral	6 months	TC ↓, LDL-C: ↓
Ajuluchukwu et al. (2007) [70]	Patients with hypercholesterolemia and one additional cardiovascular risk factor (*n* = 34)	T3 (500 mg/day)	Oral	4 weeks	Serum TC: ↓, LDL-C: ↓, HDL-C: ↔, TG: ↔
		αTF (500 mg/day)			Serum TC: ↔, LDL-C: ↔, HDL-C: ↔, TG: ↔
Heng et al. (2015) [71]	Adults with MetS (*n* = 70, aged 20–60 years)	Mixed T3 (61.1 mg αTF, 61.52 mg αT3, 112.80 mg γT3, 25.86 mg δT3) (400 mg/day)	Oral	16 weeks	Serum TC: ↓, LDL-C: ↓, HDL-C: ↓, TG: ↔, FBG: ↔, IL-6: ↓, TNF-α: ↓, TC/HDL-C ratio: ↑
Baliarsingh et al. (2005) [72]	Type 2 diabetic subjects (*n* = 19)	TRF (3 mg/kg)	Oral	60 days	Total lipid: ↓, TC: ↓, LDL-C: ↓
Daud et al. (2013) [73]	Subjects undergoing chronic hemodialysis (*n* = 81)	TRF (20 mg αTF, 30.18 mg αT3, 5.30 mg βT3, 41.66 mg γT3, 12.86 mg δT3) (90 mg/day)	Oral	16 weeks	Normalized plasma TG: ↓, normalized plasma HDL-C: ↑, Apo-A 1: ↑, CETP: ↓
Tan et al. (1991) [74]	Healthy volunteers (Preliminary: *n* = 9, follow up: *n* = 22)	PVE (18 mg TF, 32 mg T3, 240 mg palm olein)	Oral	30 days	TC: ↓, LDL-C: ↓, TG: ↔, HDL-C: ↔
Chin et al. (2011) [75]	Young healthy volunteers (*n* = 31; aged 35–49 years)	TRF (48 mg αTF, 70.4 mg αT3, 4.8 mg βT3, 57.6 mg γT3, 33.6 mg δT3) (160 mg/day)	Oral	6 months	TC: ↔, TG: ↔, HDL-C: ↔, LDL-C: ↔, HDL-C/TC: ↔, SOD: ↔, GPx: ↔, CAT: ↔, protein carbonyl contents: ↔, AGEs: ↔, MDA: ↔
	Old healthy volunteers (*n* = 31; aged >50 years)				TC: ↔, TG: ↔, HDL-C: ↑, LDL-C: ↔, HDL-C/TC: ↑, SOD: ↓, CAT: ↓, GPx: ↑, CAT: ↔, protein carbonyl contents: ↓, AGEs: ↓, MDA: ↔
Heng et al. (2013) [76]	Young (*n* = 6; aged 34.6 ± 0.8 years) and old (*n* = 8; aged 49.5 ± 0.9 years) subjects	TRF (22% TF, 78% T3) (150 mg/day)	Oral	6 months	Apo-A 1 precursor: ↑, Apo-E precursor: ↑, CRP precursor: ↓
**No effect**					
Tomeo et al. (1995) [77]	Subjects with cerebrovascular disease (*n* = 50; aged 49–83 years)	TRF (16 mg αTF, 40 mg αT3, 40 mg γT3, 240 mg PVE) (300 mg)	Oral	18 months	Apparent carotid atherosclerotic regression shown, TBARS: ↓, TC: ↔, LDL-C: ↔, TG: ↔, HDL-C: ↔
Mensink et al. (1999) [78]	Men with mildly elevated lipid concentration (*n* = 40; aged 21–61 years)	Vitamin E concentrate rich in T3 (20 mg αTF, 35 mg T3)	Oral	6 weeks	Serum LDL-C: ↔, HDL-C: ↔, TG: ↔, lipoprotein a: ↔, lipid peroxide: ↔, platelet aggregation velocity: ↔, maximum aggregation: ↔, thromboxane B_2_ formation: ↔
O’Bryne et al. (2000) [79]	Healthy volunteers on AHA Step-1 diet for 12 weeks (*n* = 51)	α-tocotrienyl acetate (148 mg tocotrienyl acetate, 134 mg free T3) (250 mg/day)	Oral	8 weeks (still on low fat diet)	In vitro LDL oxidative resistance: ↑, rate of LDL oxidation: ↓, serum LDL-C: ↔, Apo-B: ↔
		γ-tocotrienyl acetate (140 mg tocotrienyl acetate, 127 mg free T3) (250 mg/day)			
		δ-tocotrienyl acetate (119 mg tocotrienyl acetate, 108 mg free T3) (250 mg/day)			
Mustad et al. (2002) [80]	Healthy subjects on AHA Step-1 diet for 7 weeks (*n* = 68, aged 25–65 years)	Mixed αT3 and γT3 (200 mg/day)	Oral	28 days	Serum T3: ↑, lipid: ↔, glucose: ↔
		Pure γT3 (200 mg/day)			
		P25 complex T3 (200 mg/day)			
Rasool et al. (2006) [81]	Healthy men (*n* = 36; aged 21–30 years)	T3-rich vitamin E (80 mg/day)	Oral	2 months	Arterial compliance (pulse wave velocity & augmentation index): ↔, TC: ↔, LDL-C: ↔, aortic systolic BP: ↔, plasma total antioxidant status: ↔
		T3-rich vitamin E (160 mg/day)			Arterial compliance (pulse wave velocity & augmentation index): ↔, TC: ↔, LDL-C: ↔, aortic systolic BP: ↓, plasma total antioxidant status: ↔
		T3-rich vitamin E (320 mg/day)			Arterial compliance (pulse wave velocity & augmentation index): ↔, TC: ↔, LDL-C: ↔, aortic systolic BP: ↓, plasma total antioxidant status: ↑

**Table 4 nutrients-12-00259-t004:** Effects of T3 on CVD in vivo.

Researcher (Year)	Animal Model	Treatment & Dose	Mode	Treatment Duration	Findings
**Myocardial ischemia perfusion rats model**					
Das et al. (2008) [96]	Male Sprague-Dawley rats	αT3 (0.3 mg/kg)	Oral	30 days	Cytochrome-C in mitochondrial fraction: ↑, p38 MAPKα binding with caveolin-1: ↔, Src and caveolin-1 interaction: ↔, caveolin-1 and HO-1 interaction: ↔, eNOS and caveolin-1 interaction: ↔, post ischemic 26S proteasome activity: ↔, Rpt5 and α3 subunit protein: ↔
		δT3 (0.3 mg/kg)			Cytochrome-C mitochondrial fraction: ↔, p38 MAPKα binding with caveolin-1: ↔, Src and caveolin-1 interaction: ↔, caveolin-1 and HO-1 interaction: ↔, eNOS and caveolin-1 interaction: ↑, post ischemic 26S proteasome activity: ↔, Rpt5 and α3 subunit protein: ↔
		γT3 (0.3 mg/kg)			Cytochrome-C mitochondrial fraction: ↑, p38 MAPKα binding with caveolin-1: ↑, Src and caveolin-1 interaction: ↔, caveolin 1 and HO-1 interaction: ↓, eNOS and caveolin-1 interaction: ↔, post ischemic 26S proteasome activity: ↔, Rpt5 and α3 subunit protein: ↔
Das et al. (2012) [97]	Male and female adult New Zealand rabbits	αT3 (20 µmol/kg)	Oral	4 weeks	TC: ↓, coronary flow: ↑, aortic flow: ↑, left ventricular developed pressure: ↑, infarct size: ↓, TGF-β, MMP-2, MMP-9: ↓, p-Akt: ↑, ER-α: ↑: ER-β: ↔, Spot 14: ↓, Forkhead box O4: ↑
		γT3 (20 µmol/kg)			
		δT3 (20 µmol/kg)			TC: ↔, coronary flow: ↔, aortic flow: ↔, left ventricular developed pressure: ↔, infarct size: ↔, TGF-β, MMP-2, MMP-9: ↓ (female), p-Akt: ↑, ER-α: ↑: ER-β: ↔, Spot 14: ↔, Forkhead box O4 (female): ↑
**Hyperhomocysteinemia rats model**					
Kamisah et al. (2015) [99]; Norsidah et al. (2013) [93]; Norsidah et al. (2013) [94]	Male Wistar rats fed with high methionine diet (*n* = 42)	TRF (40.62% αTF acetate, 21.62% αTF, 10.71% αT3, 3.46% γTF, 18.08% γT3, and 5.49% δT3) (30 mg/kg diet)	Basal diet	5 weeks	Plasma homocysteine level: ↔; S-adenosyl methionine (SAM)/S-adenosyl homocysteine (SAH) ratio: ↔; hepatic cystathionine: ↓; liver TBARS: ↔; aortic TBARS count: ↓; NO: ↔; aortic vascular cell adhesion protein 1 (VCAM-1) expression: ↔; intima-media thickness and intima media ratio: ↔; systolic BP: ↔; heart TBARS content: ↔; GPx: ↔; SOD: ↔; CAT: ↔
		TRF (40.62% αTF acetate, 21.62% αTF, 10.71% αT3, 3.46% γTF, 18.08% γT3, and 5.49% δT3) (60 mg/kg diet)			Plasma homocysteine level: ↔; SAM/SAH ratio: ↓; hepatic cystathionine: ↓; liver TBARS: ↓; aortic TBARS count: ↓; NO: ↓; aortic VCAM-1 expression: ↔; intima-media thickness and intima media ratio: ↔; systolic BP: ↔; heart TBARS content: ↓; GPx: ↑; SOD: ↔; CAT: ↔
		TRF (40.62% αTF acetate, 21.62% αTF, 10.71% αT3, 3.46% γTF, 18.08% γT3, and 5.49% δT3) (150 mg/kg diet)			Plasma homocysteine level: ↓; SAM/SAH ratio: ↓; hepatic cystathionine: ↓; liver TBARS: ↓; aortic TBARS count: ↓; NO: ↓; aortic VCAM-1 expression: ↓; intima-media thickness and intima media ratio: ↔; systolic BP: ↔; heart TBARS content: ↓; GPx: ↑; SOD: ↔; CAT: ↔
**MetS rat model**					
Cheng et al. (2017) [57]	Male post-weaning Sprague-Dawley rats fed with high-fat diet (60% kcal) for 8 weeks (*n* = 21, aged 3 weeks old)	TRF—Tocovid (23.5% αT3, 43.2% γT3, 9.8% δT3, and 23.5% αTF) (60 mg/kg)	Oral	4 weeks	Visceral adiposity: ↔, systolic and diastolic BP: ↓, oral glucose tolerance test (OGTT): ↔, antioxidant capacity: ↑, MPO hyperactivity: ↓, expression of receptor for advanced glycation end product (RAGE): ↓, HBA1c: ↓, AGEs: ↓, TG: ↑, TC: ↓, non-HDL-C: ↓, HDL-C: ↔, non-esterified fatty acid (NEFA): ↔, hepatic TG deposition: ↓, liver total lipid: ↔, PPARs: ↔, PPAR-α and PPAR-γ expression: ↔
Wong et al. (2012) [102]	Male Wistar rats fed with high carbohydrate, high fat (*n* = 32, aged 9–10 weeks old)	TRF (31.9% αT3, 2.1% βT3, 24.8% γT3, 18.3% δT3 and 22.9% αTF) (120 mg/kg)	Oral	8 weeks	Ventricular function: ↑, attenuated cardiac stiffness: ↑, glucose and insulin tolerance: ↑, left ventricular collagen deposition: ↓, liver enzymes (ALT and AST): ↓, inflammatory cell infiltration: ↓, fat vacuoles: ↓, balloon hepatocytes: ↓, plasma free fatty acid: ↓, TG: ↓
Wong et al. (2017) [103]	Male Wistar rats fed with HCHF diet (*n* = 80, aged 9–10 weeks old)	91.6% αTF (85 mg/kg)	Oral	8 weeks	Collagen deposition: ↓, inflammatory cell infiltration: ↓, TC: ↔, NEFA: ↔, TG: ↔, liver enzymes (ALT and AST): ↓, lipid droplet: ↓, fasting plasma glucose: ↔, total fat mass: ↔, abdominal circumference: ↔, adiposity index: ↔, retroperitoneal and epididymal fat pads: ↔
		90.7% αT3 (85 mg/kg)			
		95% γT3 (85 mg/kg)			Collagen deposition: ↓, inflammatory cell infiltration: ↓, normalised systolic BP, TC: ↔, NEFA: ↓, TG: ↔, liver enzymes (ALT and AST): ↓, lipid droplet: ↓, fasting plasma glucose: ↔, total fat mass: ↔, abdominal circumference: ↔, adiposity index: retroperitoneal and epididymal fat pads: ↔
		90% δT3 (85 mg/kg)			Collagen deposition: ↓, inflammatory cell infiltration: ↓, normalised systolic BP, TC: ↓, NEFA: ↓, TG: ↓ abdominal adiposity: ↓, liver enzymes (ALT and AST): ↓, lipid droplet: ↓, fasting plasma glucose: ↓, total fat mass: ↓, abdominal circumference: ↓, adiposity index: ↓, retroperitoneal and epididymal fat pads: ↓
**Hypertension rats model**					
Muharis et al. (2010) [104]	Male SHRs and male Wistar Kyoto rats injected with STZ (75 mg/kg, i.p.) for 8 weeks (aged 18 weeks old)	αTF (0.01 mg/mL)	Aortic rings incubated with extracts	20 minutes	Radical scavenging capability: ↑, acetylcholine relaxation (aortic ring): ↑
		TRF (0.01 mg/mL)			
Newaz et al. (2003) [105]	Male SHRs (*n* = 33, aged 8–10 weeks old) and male Wistar Kyoto (WKY) rats (*n* = 17, aged 8–10 weeks old)	90% γT3 (15 mg/kg)	Oral	3 months	Systolic BP: ↓, nitrite level: ↑, MDA: ↓, nitric oxide synthase (NOS) activity negatively correlated with BP
		90% γT3 (30 mg/kg)			Systolic BP: ↓, nitrite level: ↑, MDA: ↓, NOS activity: ↑, NOS activity negatively correlated with BP
		90% γT3 (150 mg/kg)			

MAPK, mitogen-activated protein kinase.

**Table 5 nutrients-12-00259-t005:** Effects of T3 on CVD in humans.

Researcher (Year)	Study Design	Treatment & Dose	Mode	Treatment Duration	Findings
Mahdy et al. (2013) [106]	Randomized placebo-controlled double-blind clinical trial (*n* = 151; mean age: 25.94)	TRF (100 mg/day)	Oral	12–16 weeks pregnancy until delivery	Relative risk of PIH = 0.36 (95% CI = 0.12–1.09): ↔, Relative risk of preeclampsia = 0.20 (95% CI = 0.02–1.66): ↔, clinical severity: ↔, hematological or biochemical indices: ↔, blood loss at delivery: ↓
Rasool et al. (2006) [81]	Randomized blinded end-point, placebo-controlled clinical trial (36 males, aged 21–30 years old)	T3 rich vitamin E (26.2% αTF, 34.6% αT3, 24.63 γT3, 15% δT3) (80 mg/day)	Oral	2 months	Plasma vitamin E level: ↑, augmentation index: ↔, plasma total antioxidant status: ↔, TC: ↔, LDL-C: ↔
		T3 rich vitamin E (26.2% αTF, 34.6% αT3, 24.63 γT3, 15% δT3) (160 mg/day)			Plasma vitamin E level: ↑, augmentation index: ↔, plasma total antioxidant status: ↔, TC: ↔, LDL-C: ↔
		T3 rich vitamin E (26.2% αTF, 34.6% αT3, 24.63 γT3, 15% δT3) (320 mg/day)			Plasma vitamin E level: ↑, plasma total antioxidant status: ↑, TC: ↔, LDL-C: ↔
Mustad et al. (2002) [80]	Randomized, double-blind parallel design (68 healthy subjects: 39 males and 29 females, aged 25–65 years old)	αT3 plus γT3 extract from palm oil (Tocovid) (200 mg/day)	Oral	7 weeks	Serum concentrations of T3: ↔, fasting total: ↔, LDL-C: ↔, glucose: ↔, no significant correlation between serum T3 concentrations and changes in fasting serum lipids or glucose and urinary 8-epi-PGF_2α._
		High γT3 extract from rice bran oil (Nutriene) (200 mg/day)			Serum concentrations of T3: ↑, fasting total: ↔, LDL-C: ↔, glucose: ↔, no significant correlation between serum T3 concentrations and changes in fasting serum lipids or glucose and urinary 8-epi-PGF_2α._
		P25-complex extract from rice bran oil (EvolvE) (200 mg/day)			Serum concentrations of T3: ↔, fasting total: ↔, LDL-C: ↔, glucose: ↔, no significant correlation between serum T3 concentrations and changes in fasting serum lipids or glucose and urinary 8-epi-PGF_2α._
Rasool et al. (2008) [107]	Randomized placebo controlled, blinded end point clinical study (36 healthy male, aged <40 years old)	Tocovid (23.5% αTF, 23.54% αT3, 23.54% γT3, 43.16% δT3) (50 mg/day)	Oral	2 months	Pulse wave velocity: ↔, augmentation index: ↓, BP: ↔, TC: ↔ LDL-C: ↔.
		Tocovid (23.5% αTF, 23.54% αT3, 23.54% γT3, 43.16% δT3) (100 or 200 mg/day)			Pulse wave velocity: ↓, augmentation index: ↓, BP: ↔, TC: ↔ LDL-C: ↔.

**Table 6 nutrients-12-00259-t006:** Effects of T3 on cerebrovascular diseases in vivo.

Researcher (Year)	Animal Model	Treatment & Dose	Mode	Treatment Duration	Findings
Rink et al. (2011) [109]	Mongrel canines (acute ischemic stroke induced by MCAO) (*n* = 20, aged 2.4 ± 0.9 years)	TRF (200 mg twice per day)	Oral (gel capsule)	10 weeks	Cytotoxic edema at 1 h: ↓; protect fibre tract projection (hemisphere); relative connectivity of fibre tracts between the internal capsule and corona radiate: ↑; cerebrovascular collateral circulation: ↑; artery territory collateral score: ↑; gene expression of chloride intracellular channel 1 and tissue inhibitor of metalloproteinase-1 (TIMP-1): ↑; MMP-2 (cerebral cortex): ↓.
Khanna et al. (2013) [110]	C57BL/6 male mice, Harlan (transient 90 minutes focal cerebral ischemic induced by MCAO (*n* = 12, aged 5 weeks old)	αT3 (50 mg/kg)	Oral	10 weeks	Rescued stroke-induced loss of miR-29b and minimized lesion size.
Park et al. (2011) [111]	C57BL/6 male Harlan Indianapolis rats (transient focal cerebral ischemia induced by MCAO) (*n* = 41, aged 5 weeks old)	αT3 (50 mg/kg)	Oral	13 weeks	Brain αT3 level: ↑; attenuated hemispherical infarct volume; abundance of MRP1 positive cells at infarct hemisphere: ↑; miR199a-5p in infarct hemisphere: ↓; attenuated stroke-induced neurodegeneration; 4-HNE-positive cells: ↓;
Mishima et al. (2003) [112]	Male ddY mice with focal cerebral infarction induced by MCAO (*n* = 80)	αTF (0.2 & 2 mM)	i.v.	24 h	Infarct volume: ↓, body temperature: ↔, BP; ↔
		αT3 (0.2 & 2 mM)			
		γTF (0.2 & 2 mM)			
		γT3 (0.2 & 2 mM)			Infarct volume: ↔, body temperature: ↔, BP; ↔
		δTF (0.2 & 2 mM)			
		δT3 (0.2 & 2 mM)			
Jiao et al. (2018) [116]; Shang et al. (2018) [117]	Male ICR mice with transient MCAO for 60 minutes (*n* = 119, aged 6 weeks old)	Tocovid (11.3% αTF, 12.4% αT3, 2.5% βT3, 19.2% γT3, 6.3% δT3) (200 mg/kg/day)	Oral	1 month	Mice neurobehaviors: ↑, infarct volume: ↓, inflammatory markers (TNF-α, MCP-1, Iba-1): ↓, neurovascular units (MMP-9, immunoglobulin G and collagen IV): ↑, rotarod time: ↑, infarct volume: ↓, 4-HNE, nitrotyrosine and 8-OhdG: ↓, RAGE, chaperone-mediated autophagy and CML expressions: ↓, Nrf2 and MRP1: ↑, GSSG/GSH ratio: ↓, caspase-3 and LC3-II expressions: ↓

**Table 7 nutrients-12-00259-t007:** Effects of T3 on osteoporosis in female rats.

Researcher (Year)	Animal Model	Treatment & Dose	Mode	Treatment Duration	Findings
Soelaiman et al. (2012) [135]	Ovariectomized female Sprague-Dawley rats (*n* = 32, aged 4 months old)	Palm T3 (24.67% αT3, 38.955% γT3, 4.55% δT3, 20.11% αTF) (60 mg/kg)	Oral	8 weeks	dLS/BS: ↑, sLS/BS: ↓, MS: ↑, MAR: ↑, BFR/BS: ↑, OCN: ↔, CTX-1: ↓
Aktifanus et al. (2012) [141]	Ovariectomized female Sprague-Dawley rats (*n* = 32, aged 4 months old)	TRF (24.67% αT3, 38.955% γT3, 4.55% δT3, 20.11% αTF) (60 mg/kg)	Oral	2 months	OCN: ↔, CTX-1: ↔, sLS: ↓, dLS: ↑, MS: ↔, MAR: ↑, BFR: ↑
Chin et al. (2017) [137]	Ovariectomized female Sprague-Dawley rats (*n* = 40, aged 3 months old)	Annatto T3 (10% γT3, 90% δT3) (60 mg/kg)	Oral	8 weeks	sLS: ↓, dLS: ↑, MS: ↑, MAR: ↑, BFR: ↑, BMP-2: ↑
Muhammad et al. (2012) [136]	Ovariectomized female Wistar rats (*n* = 40, aged 3 months old)	Palm T3 (37.2% αT3, 39.1% γT3, 22.6% δT3) (60 mg/kg)	Oral	4 weeks	BV/TV: ↑, Tb.Sp: ↓, Tb.N: ↑, Tb.Th: ↔, Oc.S: ↓, Ob.S: ↔
		αTF (60 mg/kg)			
Deng et al. (2014) [138]	Ovariectomized female C57BL/6 mice (*n* =56, aged 8 weeks old)	γT3 (100 mg/kg)	s.c.	3 months	BMD: ↑, BV/TV: ↑, Tb.Th: ↑, Tb.N: ↑, Tb.Sp: ↓, Oc.N: ↓, Ob.N: ↑, MAR: ↑, BFR: ↑, OCN: ↑, CTX-1: ↓, RANKL: ↓, OPG: ↑, Osterix: ↑, Runx-2: ↑
Abdul-Majeed et al. (2012) [139]	Ovariectomized female Sprague-Dawley rats (*n* =48, aged 3 months old)	δT3 (60 mg/kg)	Oral	8 weeks	OCN: ↑, CTX-1: ↓, Ob.S: ↑, Oc.S: ↓, ES: ↓, OS: ↑, OV: ↑
		δT3 (60 mg/kg) + lovastatin (11 mg/kg)			
Abdul-Majeed et al. (2015) [140]	Ovariectomized female Sprague-Dawley rats (*n* = 48, aged 3 months old)	δT3 (60 mg/kg)	Oral	8 weeks	BV/TV: ↑, Tb.N: ↑, Tb.Th: ↑, Tb.Sp: ↓, load: ↑, stress: ↑, strain: ↑, Young’s Modulus: ↑
		δT3 (60 mg/kg) + lovastatin (11 mg/kg)			
Mohamad et al. (2012) [142]	Ovariectomized female Sprague-Dawley rats (*n* = 32, aged 3 months old)	αTF (60 mg/kg)	Oral	2 months	Stress: ↔, load: ↔, strain: ↔, Young’s Modulus: ↔
		T3-enriched fraction (73.9 mg/g αTF, 167.1 mg/g αT3, 41.1 mg/g βT3, 165.2 mg/g γT3, 98.5 mg/g δT3) (60 mg/kg)			Stress: ↑, load: ↔, strain: ↔, Young’s Modulus: ↔
Norazlina et al. (2000) [134]	Ovariectomized female Sprague-Dawley rats (*n* = 40, aged 3 months old)	αTF (30 mg/kg)	Oral	8 months	BMD: ↔, calcium (femur & lumbar): ↔, ALP: ↑, tartrate-resistant acid phosphatase (TRAP): ↓
		PVE (24.82% αTF, 20.73% αT3, 26.68% γT3, 13.32% δT3) (30 mg/kg)			BMD: ↔, calcium (femur & lumbar): ↓, ALP: ↑, TRAP: ↔
		PVE (24.82% αTF, 20.73% αT3, 26.68% γT3, 13.32% δT3) (60 mg/kg)			BMD: ↔, calcium (femur & lumbar): ↔, ALP: ↔, TRAP: ↔
Muhammad et al. (2013) [161]	Ovariectomized female Wistar rats (*n* = 32, aged 3 months old)	Palm T3 (37.2% αT3, 39.1% γT3, 22.6% δT3) (60 mg/kg)	Oral	4 weeks	IL-1: ↓, IL-6: ↓, OCN: ↓, pyridinoline (serum): ↔
Nazrun et al. (2008) [159]	Ovariectomized female Wistar rats (*n* = 64, aged 3 months old)	αTF (60 mg/kg)	Oral	8 weeks	MDA: ↔, GPx: ↔, SOD: ↑, load: ↔, stress: ↔, stiffness: ↔, Young’s Modulus: ↔
		Palm T3 (30.7% αT3, 55.2% γT3, 14.1% δT3) (60 mg/kg)			MDA: ↓, GPx: ↑, SOD: ↑, load: ↔, stress: ↔, stiffness: ↔, Young’s Modulus: ↔
Ibrahim et al. (2014) [143]	Ovariectomized female Sprague-Dawley rats with fracture at metaphysis region of right tibia (*n* =48, aged: 3 months old)	Annatto T3 particles (60 mg/kg)	Single injection	-	BV/TV_callus_: ↔, BMD_callus_: ↔, load: ↑, stress: ↑, strain: ↔, Young’s Modulus: ↔
		Annatto T3 particles (60 mg/kg) + lovastatin particles (750 µg/kg)			BV/TV_callus_: ↑, BMD_callus_: ↑, load: ↑, stress: ↑, strain: ↔, Young’s Modulus: ↔
Ibrahim et al. (2015) [166]	Ovariectomized female Sprague-Dawley rats with fracture at metaphysis region of right tibia (*n* = 48, aged 3 months old)	Annatto T3 particles (60 mg/kg)	Single injection	-	OCN: ↔, BMP-2: ↔, VEGF-α: ↔, Runx-2: ↔, bone sialoprotein (BSP): ↔, TGF-β2: ↔, TGF-β3: ↔, collagen type II alpha 1 (*COL2α1*): ↔, *FGF-23*: ↔
		Annatto T3 particles (60 mg/kg) + lovastatin particles (750 µg/kg)			OCN: ↑, BMP-2: ↑, VEGF-α: ↑, Runx-2: ↑, BSP: ↔, TGF-β2: ↔, TGF-β3: ↔, *COL2α1*: ↔, *FGF-23*: ↔
Norazlina et al. (2002) [133]	Female Sprague-Dawley rats fed with VED diet (*n* = 40, aged 4 months old)	αTF (30 mg/kg)	Oral	8 months	calcium (5th lumbar & femur): ↔
		PVE (24.82% αTF, 20.73% αT3, 26.68% γT3, 13.32% δT3) (30 mg/kg)			calcium (5th lumbar): ↑, calcium (femur): ↔
		PVE (24.82% αTF, 20.73% αT3, 26.68% γT3, 13.32% δT3) (60 mg/kg)			calcium (5th lumbar): ↑, calcium (femur): ↑
Norazlina et al. (2004) [132]	Female Sprague-Dawley rats fed with VED diet (*n* = 40, aged 3 months old)	αT3 (60 mg/kg)	Oral	3 months	PTH (serum): ↑, calcium (serum): ↔, calcium (femoral): ↔, DPD (urine): ↔, OCN: ↔
		αTF (60 mg/kg)			

**Table 8 nutrients-12-00259-t008:** Effects of T3 on osteoporosis in male rats.

Researcher (Year)	Animal Model	Treatment & Dose	Mode	Treatment Duration	Findings
Maniam et al. (2008) [160]	Normal male Sprague-Dawley rats (*n* = 56, aged 3 months old)	αTF (30 mg/kg)	Oral	4 months	TBARS: ↔, GPx: ↔, SOD: ↔
		αTF (60 mg/kg)			
		αTF (100 mg/kg)			
		Palm T3 (30 mg/kg)			
		Palm T3 (60 mg/kg)			
		Palm T3 (100 mg/kg)			TBARS: ↓, GPx: ↑, SOD: ↔
Mehat et al. (2010) [144]	Normal male Sprague-Dawley rats (*n* = 32, aged 3 months old)	αTF (60 mg/kg)	Oral	4 months	Oc.N: ↓, N.Ob: ↑, ES: ↓, OS: ↑, sLS/BS: ↓, dLS/BS: ↑, BFR: ↑, OV: ↔, MS: ↔, MAR: ↔
		δT3 (60 mg/kg)			Oc.N: ↓, N.Ob: ↑, ES: ↓, OS: ↑, sLS/BS: ↓, dLS/BS: ↑, BFR: ↑, OV: ↑, MS: ↑, MAR: ↑
		γT3 (60 mg/kg)			
Shuid et al. (2010) [157]	Normal male Sprague-Dawley rats (*n* = 24, aged 3 months old)	αTF (60 mg/kg)	Oral	4 months	BV/TV: ↑, Tb,Th: ↑, Tb.N: ↑, Tb.Sp: ↓, displacement: ↑, stress:↑, strain: ↑, load: ↔, stiffness: ↔
		γT3 (60 mg/kg)			BV/TV: ↑, Tb.Th: ↑, Tb.N: ↑, Tb.Sp: ↓, displacement: ↑, stress: ↑, strain: ↑, load: ↑, stiffness: ↑
Tennant et al. (2017) [151]	Male Sprague-Dawley rats (*n* = 19, aged 11 weeks old)	Adequate dietary αTF	Oral	18 weeks	Bone mineral content, bone area & BMD: ↔, cross-sectional volume: ↔. cortical volume: ↔, Ct.Th: ↔, BV/TV: ↔, Tb.Th: ↔, Tb.N: ↔, Tb.Sp: ↔, mineralizing perimeter: ↔ MAR: ↔, BFR: ↔,OCN (serum): ↔, Runx-2: ↔, Sp7: ↔, OCN (gene): ↓, osteopontin: ↔, ALP: ↔, RANKL: ↔, CSF-1: ↔
		High dietary αTF (500 mg/kg)			
		High dietary mixed Tocomin (250 mg/kg diet)			
Norazlina et al. (2007) [162]	Nicotine-induced osteoporotic male Sprague-Dawley rats (*n* = 24, aged 3 months old)	Palm T3 (60 mg/kg)	Oral	3 months	IL-1: ↔, IL-6: ↓, OCN: ↔, calcium (femur): ↔
		αTF (60 mg/kg)			IL-1: ↑, IL-6: ↑, OCN: ↔, calcium (femur): ↔
Hermizi et al. (2009) [145]	Nicotine-induced osteoporotic male Sprague-Dawley rats (*n* = 49, age: 3-month-old)	T3-enhanced fraction (43% αT3, 31% γT3, 14% δT3, 12% other oils) (60 mg/kg)	Oral	2 months	BV/TV: ↑, Tb.Th: ↑, Tb.N: ↔, MAR: ↑, BFR: ↑, sLS: ↓, Ob.S: ↑, Oc.S: ↓, ES: ↓
		γT3 (60 mg/kg)			BV/TV: ↑, Tb.Th: ↑, Tb.N: ↑, MAR: ↑, BFR: ↑, sLS: ↓, Ob.S: ↑, Oc.S: ↓, ES: ↓
Norazlina et al. (2010b) [156]	Nicotine-induced osteoporotic male Sprague-Dawley rats (*n* = 32, aged 3 months old)	T3 mixture (60 mg/kg)	Oral	12 weeks	calcium (femoral): ↑, OPG and RANKL (serum): ↔, calcium (4th lumbar): ↔
		αTF (60 mg/kg)			
Norazlina et al. (2010a) [164]	Nicotine-induced osteoporotic male Sprague-Dawley (*n* = 35, aged 3 months old)	αTF (60 mg/kg)	Oral	4 months	IL-1: ↓, PYD: ↔, OCN: ↔
		T3-enriched fraction (60 mg/kg)			
		γT3 (60 mg/kg)			IL-1: ↓, PYD: ↓, OCN: ↑
Abukhadir et al. (2012) [165]	Nicotine-induced osteoporotic male Sprague-Dawley rats (*n* = 32, aged 3 months old)	PVE (60 mg/kg)	Oral	4 months	*BMP-2:* ↑, *OSX:* ↑, Runx-2: ↑
Ima-Nirwana & Fakhrurazi (2002) [154]	Adrenalectomized & dexamethasone-induced osteoporotic male Wistar rats (*n* = 70, age: 3-month-old)	PVE (20.73% αT3, 26.68% γT3, 13.32% δT3, 24.82% αTF) (60 mg/kg)	Oral	8 weeks	BMD (whole body & regional): ↑, femoral length: ↑, calcium content: ↑
Ima-Nirwana & Suhaniza (2004) [155]	Adrenalectomized & dexamethasone-induced osteoporotic male Sprague-Dawley rats (*n* = 42, age: 4-month-old)	γT3 (60 mg/kg)	Oral	8 weeks	BMD: ↔, calcium content (femur): ↔, calcium content (4th lumbar): ↑
Ahmad et al. (2005) [146]	Ferric nitrilotriacetate (2 mg/kg, i.p.)-induced osteoporotic male Wistar rats (*n* = 32, age: 4-week-old)	αTF (10 mg/kg)	Oral	8 weeks	IL-1: ↔, IL-6: ↔, OCN: ↔
		αTF (30 mg/kg)			IL-1: ↔, IL-6: ↓, OCN: ↔, DPD: ↔
		αTF (60 mg/kg)			IL-1: ↔, IL-6: ↓, OCN: ↔, DPD: ↔
		αTF (100 mg/kg)			Ob.N: ↔, Oc.N: ↔, ES: ↔, BFR: ↔, BV/TV: ↔, Tb.Th: ↔, Tb.N: ↔, OCN: ↓, DPD: ↓, IL-1: ↔, IL-6: ↓
		Palm T3 (30.7% αT3, 55.2% γT3 and 14.1% δT3) (10 mg/kg)			IL-1: ↔, IL-6: ↔, OCN: ↔, DPD: ↔
		Palm T3 (30 mg/kg)			IL-1: ↔, IL-6: ↓, OCN: ↔, DPD: ↔
		Palm T3 (60 mg/kg)			IL-1: ↓, IL-6: ↓, OCN: ↔, DPD: ↓
		Palm T3 (100 mg/kg)			Ob.N: ↑, Oc.N: ↔, ES: ↓, BFR: ↑, BV/TV: ↑, Tb.Th: ↑, Tb.N: ↔, OCN: ↓, DPD: ↓, IL-1: ↓, IL-6: ↓
Ima-Nirwana et al. (2000) [153]	Orchidectomized male Wistar rats (*n* = 30, age: 3-month-old)	PVE (21.6% αT3, 27.7% γT3, 11% δT3, 15.3% palm olein, 24.4% αTF) (30 mg/kg)	Oral	8 months	BMD: ↑, bone calcium content: ↔, ALP: ↔, TRAP: ↔
Chin et al. (2014) [152]	Orchidectomized male Sprague-Dawley rats (*n* = 40, age: 3-month-old)	Annatto T3 (10% γT3, 90% δT3) (60 mg/kg)	Oral	8 weeks	P1NP: ↔, TRACP 5b: ↔, Ob.S: ↑, Oc.S: ↓, ES: ↓, OS: ↑, OV: ↑
Chin & Ima-Nirwana (2014) [147]	Orchidectomized male Sprague-Dawley rats (*n* = 40, age: 3-month-old)	Annatto T3 (10% γT3, 90% δT3) (60 mg/kg)	Oral	8 weeks	OCN: ↔, CTX-1: ↔, BV/TV: ↑, Tb.N: ↑, Tb.Th: ↔, Tb.Sp: ↓, sLS: ↓, dLS: ↑, MS: ↔, MAR: ↔, BFR: ↔, *ALPL*: ↑, bone gamma-carboxyglutamate protein (*BGLAP*): ↔, *COL1α1*: ↑, β-catenin: ↑, *RUNX-2*: ↔, *SP7*: ↔, *SPARC*: ↔, integrin binding sialoprotein (*IBSP*): ↔, *OPG*: ↔
Chin et al. (2016) [158]	Orchidectomized male Sprague-Dawley rats (*n* = 30, age: 3-month-old)	Annatto T3 (10% γT3, 90% δT3) (60 mg/kg)	Oral	8 weeks	Calcium (serum): ↓, phosphate (serum): ↔, calcium (bone): ↑, load: ↔, stress: ↔, strain: ↔, extension: ↔
Wong et al. (2018) [148]; Wong et al. (2018) [150]; Wong et al. (2019) [163]	MetS-induced osteoporotic male Wistar rats (*n* = 30, aged 12 weeks old)	Palm T3 (21.9% αTF, 24.7% αT3, 4.5% βT3, 36.9% γT3, 12.0% δT3) (60 mg/kg)	Oral	20 weeks	Abdominal circumference: ↔, systolic & diastolic pressure: ↓, FBG: ↓, OGTT: ↔, TG: ↓, TC: ↓, HDL-C: ↔, LDL-C: ↓, fat mass: ↔, lean mass: ↔, percentage of fat: ↔, BV/TV: ↑, Tb.N: ↑, Tb.Sp: ↓, SMI: ↓, Tb.Th: ↔, Conn.D: ↔, bone calcium (femur): ↔, load: ↑, displacement: ↔, stiffness: ↑, stress: ↔, strain: ↔, Young’s Modulus: ↑, Ob.S: ↑, Oc.S: ↔, ES: ↔, OS: ↑, OV: ↔, sLS: ↓, dLS: ↔, MS: ↔, MAR: ↔, BFR: ↔, OCN: ↔, CTX-1: ↔, leptin: ↓, adiponectin: ↔, insulin: ↔, IL-6: ↓, IL-1α: ↔, IL-1β: ↔, TNF-α: ↔, OPG: ↔, RANKL: ↓, SOST: ↔, DKK-1: ↔, FGF-23: ↓, PTH: ↔
		Palm T3 (21.9% αTF, 24.7% αT3, 4.5% βT3, 36.9% γT3, 12.0% δT3) (100 mg/kg)			Abdominal circumference: ↔, systolic & diastolic pressure: ↓, FBG: ↓, OGTT: ↓, TG: ↓, TC: ↓, HDL-C: ↑, LDL-C: ↓, fat mass: ↔, lean mass: ↔, percentage of fat: ↔, BV/TV: ↑, Tb.N: ↑, Tb.Sp: ↓, SMI: ↓, Tb.Th: ↔, Conn.D: ↔, bone calcium (femur): ↔ load:↑, displacement: ↔, stiffness: ↔, stress: ↔, strain: ↔, Young Modulus: ↑, Ob.S: ↑, Oc.S: ↔, ES: ↑, OS: ↑, OV: ↔, sLS: ↔, dLS: ↔, MS: ↔, MAR: ↔, BFR: ↔, OCN: ↔, CTX-1: ↔, leptin: ↔, adiponectin: ↔, insulin: ↔, IL-1α: ↓, IL-1β: ↔, IL-6: ↓, TNF-α: ↔, OPG: ↔, RANKL: ↓, SOST: ↓, DKK-1: ↓, FGF-23: ↓, PTH: ↔
Wong et al. (2018) [149]; Wong et al. (2019) [163]	MetS-induced osteoporotic male Wistar rats (*n* = 30, aged 12 weeks old)	Annatto T3 (16% γT3, 84% δT3) (60 mg/kg)	Oral	20 weeks	Abdominal circumference: ↔, systolic & diastolic pressure: ↓, FBG: ↓, OGTT: ↔, TG: ↓, TC: ↓, HDL-C: ↔, LDL-C: ↔, fat mass: ↔, lean mass: ↔, percentage of fat: ↔, BV/TV: ↑, Tb.N: ↑, Tb.Sp: ↓, SMI: ↓, Tb.Th: ↔, Conn.D: ↓, bone calcium (femur): ↔, load: ↔, displacement: ↔, stiffness: ↔, stress: ↔, strain: ↔, Young’s modulus: ↔, Ob.S: ↑, Oc.S: ↔, ES: ↔, OS: ↔, OV: ↔, sLS: ↓, dLS: ↔, MS: ↔, MAR: ↑, BFR: ↔, OCN: ↔, CTX-1: ↔, leptin: ↔, adiponectin: ↑, insulin: ↔, IL-1α: ↓, IL-1β: ↔, IL-6: ↓, TNF-α: ↔, OPG: ↔, RANKL: ↓, SOST: ↔, DKK-1: ↔, FGF-23: ↓, PTH: ↔
		Annatto T3 (16% γT3, 84% δT3) (100 mg/kg)			Abdominal circumference: ↔, systolic & diastolic pressure: ↓, FBG: ↓, OGTT: ↔, TG: ↓, TC: ↓, HDL-C: ↔, LDL-C: ↓, fat mass: ↔, lean mass: ↔, percentage of fat: ↔, BV/TV: ↑, Tb.N: ↑, Tb.Sp: ↓, SMI: ↓, Tb.Th: ↔, Conn.D: ↓, bone calcium content: ↔, load: ↑, displacement: ↔, stiffness: ↔, stress: ↔, strain: ↓, Young’s modulus: ↑, Ob.S: ↑, Oc.S: ↔, ES: ↔, OS: ↔, OV: ↔, sLS: ↔, dLS: ↔, MS: ↔, MAR: ↔, BFR: ↔, OCN: ↔, CTX-1: ↔, leptin: ↓, adiponectin: ↑, insulin: ↔, IL-1α: ↓, IL-1β: ↔, IL-6: ↓, TNF-α: ↔, OPG: ↔, RANKL: ↓, SOST: ↓, DKK-1: ↓, FGF-23: ↓, PTH: ↔

**Table 9 nutrients-12-00259-t009:** Effects of T3 on arthritis in vivo.

Researcher (Year)	Animal Model	Treatment & Dose	Mode	Treatment Duration	Findings
Radhakrishnan et al. (2014) [169]	CIA in female dark Agouti rats (*n* = 24, aged 10 weeks old)	γT3 (5 mg/kg)	Oral	24 days	Hind paw thickness and edema: ↓, CRP: ↓, TNF-α: ↓, SOD: ↑, GSH: ↑
Haleagrahara et al. (2014) [170]	CIA in female dark Agouti rats (*n* = 24, aged 6–10 weeks old)	δT3 (10 mg/kg)	Oral	25 days	Paw edema (left & right hind limb joint): ↓, collagen-stimulated lymphocytes (proliferation): ↓, CRP: ↓
Zainal et al. (2019) [172]	CIA in female dark Agouti rats (*n* = 30, aged 4–5 weeks old)	TRF (25% αTF, 75% T3) (30 mg/kg)	Oral	18 days	Ankle circumference: ↓, swelling and redness of the joints: ↓, cartilage was smooth, number of cartilage cells: ↑, CRP: ↓, TNF-α: ↓, IL-1β: ↓, IL-6: ↓, BMD: ↑
Chin et al. (2019) [171]	Sprague-Dawley male rats (*n* = 40, aged 3 months old) induced with monosodium iodoacetate	Annatto T3 (50, 100, 150 mg/kg)	Oral	5 weeks	Joint histology scoring: ↓, cartilage oligomeric matrix protein in serum: ↓

**Table 10 nutrients-12-00259-t010:** Effect of T3 on peptic ulcer in vivo.

Researcher (Year)	Animal Model	Treatment & Dose	Mode	Treatment Duration	Findings
**Stress-induced gastric ulcer model**					
Ibrahim et al. (2011) [252]	WIRS-induced male Sprague-Dawley rats (*n* = 60)	PVE (60 mg/kg)	Oral	28 days	Plasma acetylcholine: ↓, gastric contraction: ↓, amplitude of contraction: ↓, lesion index: ↓
		αTF (60 mg/kg)			
Fahami et al. (2011) [224]	WIRS-induced male Sprague-Dawley rats (*n* = 24)	TF (60 mg/kg)	Oral	28 days	MDA: ↓, GPx: ↔, SOD: ↔, GSH/GSSG: ↑
		T3 (60 mg/kg)			MDA: ↓, GPx: ↓, SOD: ↔, GSH/GSSG: ↑
Kamisah et al. (2011) [223]	WIRS-induced male Sprague-Dawley rats (*n* = 60)	TRF (21% αTF, 4% γTF, 17% αT3, 33% γT3, 24% δT3) (60 mg/kg/day)	Oral	28 days	Gastric lesions index: ↓, XO activity: ↓, TBARS: ↓
		αTF (60 mg/kg)			
Rodzian et al. (2013) [229]	WIRS-induced male Sprague-Dawley rats (*n* = 14)	T3 (90% δT3, 10% γT3) (60 mg/kg)	Oral	28 days	Gastric lesions score: ↓, MDA: ↑, PGE_2_: ↔
Nur Azlina et al. (2013) [221]	WIRS-induced male Wistar rats (*n* = 28)	T3 (44.8% d-γT3, 29.4% d-αT3, 10.8% d-δT3, and 5% d-βT3 (60 mg/kg)	Oral	28 days	Gastric lesion index: ↓, gastric acid concentration: ↓, PGE_2_: ↑, COX-1: ↑, COX-2: ↓
Nur Azlina et al. (2015) [230]	WIRS-induced male Wistar rats (*n* = 28)	T3 (60 mg/kg)	Oral	28 days	Gastric lesion index: ↓, MDA: ↓, SOD activity: ↑, iNOS expression: ↓, NO: ↓, TNF-α: ↓, IL-1β: ↓
Nur Azlina et al. (2017) [238]	WIRS-induced male Wistar rats (*n* = 28)	T3 (60 mg/kg)	Oral	28 days	Gastric lesion index: ↓, VEGF: ↑, EGF: ↔, TGF-α: ↑, bFGF: ↑
**NSAIDs-induced gastric ulcer model**					
Nafeeza et al. (2002) [240]	Aspirin-induced male Sprague-Dawley rats (*n* = 36)	TF (300 mg/kg diet)	Oral	8 weeks	Gastric lesion index: ↓, gastric MDA content: ↓, adherent mucous concentration: ↔, gastric acid concentration: ↔
		TRF (80% T3 and 20% TF) (300 mg/kg diet)			
Saad et al. (2002) [244]	Indomethacin-induced male Sprague-Dawley rats (*n* = 48)	TRF (150 mg/kg diet)	Oral	8 weeks	MDA: ↔, GSH: ↑, PGE_2_: ↔, gastric acid: ↔, gastric mucous concentration: ↔
Jaarin et al. (2002) [242]	Aspirin-induced male Sprague-Dawley rats (*n* = 96)	PVE (60, 100, or 150 mg/kg diet)	Oral/orogastric	4 weeks	Gastric lesion index: ↓, gastric MDA: ↓, gastric acid concentration: ↔
		TF (20, 30, or 50 mg/kg diet)			
Nafeeza & Kang (2005) [239]	Indomethacin-induced male Sprague-Dawley rats (*n* = 48)	Combination of TF-T3, TF-ubiquinone or T3-ubiquinone)	Oral	28 days	Gastric lesion: ↔, lipid peroxidation: ↓, PGE_2_: ↑, gastric acid concentration: ↔, GSH/GSSG ratio: ↔
Ohta et al. (2006) [245]	Compound C48/80-induced male Wistar rats (aged 6 weeks)	αTF (50 mg/kg)	Oral	0.5 h after treatment with compound C48/80	Gastric mucosal lesions: ↔, serotonin: ↔, histamine: ↔, gastric mucosal blood flow: ↔, vitamin E content: ↑, TBARS: ↓, MPO activity: ↔, XO activity: ↔, GPx activity: ↔, ascorbic acid & hexosamine: ↔
		αTF (100 mg/kg)			Gastric mucosal lesions: ↓, serotonin: ↔, histamine: ↔, gastric mucosal blood flow: ↔, vitamin E content: ↑, TBARS: ↓, MPO activity: ↓, XO activity: ↓, GPx activity: ↑, ascorbic acid & hexosamine: ↑
		αTF (250 mg/kg)			
Fesharaki et al. (2006) [210]	Aspirin-induced male Wistar rats (*n* = 90)	Vitamin E (75 units)	Intragastric	3, 6, 9, & 24 h	Gross & histological lesion: ↓, PGE_2_: ↑, MPO activity: ↓, XO activity: ↔, SOD activities: ↑, GSH: ↑, GPx: ↑
Odabasoglu et al. (2008) [243]	Indomethacin-induced male Sprague-Dawley rats (*n* = 102)	αTF (100 mg/kg)	Oral	6 h	Ulcer area: ↓, CAT: ↓, SOD: ↑, GST: ↓, glutathione reductase: ↑, GSH: ↑, MPO: ↓
Jiang et al. (2009) [215]	Carrageenan-induced air-pouch inflammation male Wistar rats	Aspirin + γTF (33 mg/kg)	Oral	3 days	PGE_2_: ↓, exudate volume: ↓, number of infiltrating immune cells: ↔, LDH: ↓, plasma & exudate γTF: ↑, 8-isoprostane: ↓
		Aspirin + αTF (33 mg/kg)			PGE_2_: ↔, exudate volume: ↔, number of infiltrating immune cells: ↔, LDH: ↔, plasma & exudate αTF: ↑, 8-isoprostane: ↔
**Ethanol-induced gastric ulcer model**					
Jaarin et al. (1999) [249]	Ethanol-induced male Sprague-Dawley rats (*n* = 56)	PVE (150 mg/kg diet)	Oral (PVE), i.p. (ranitidine)	3 weeks	Mean ulcer index: ↓, gastric acid concentration: ↔, MDA: ↓, gastric mucous concentration: ↔
		PVE (150 mg/kg diet) + ranitidine (30 mg/kg)			Mean ulcer index: ↓, gastric acid concentration: ↓, MDA: ↓, gastric mucous concentration: ↔
Ismail et al. (1999) [250]	Ethanol-induced male Sprague-Dawley rats	PVE (150 mg/kg diet)	Oral	3 weeks	Gastric acid concentration: ↔, MDA: ↓, PGE_2_: ↔
Jaarin et al. (2000) [246]	Ethanol-induced male Sprague-Dawley rats (*n* = 42)	PVE (150 mg/kg diet)	Oral	3 weeks	Mean ulcer index: ↓, MDA: ↓, gastric acid concentration: ↔, PGE_2_: ↔

**Table 11 nutrients-12-00259-t011:** Effect of T3 on Parkinson’s disease.

**In Vitro Study**					
**Researcher (Year)**	**Cell Type**	**Treatment & Dose**		**Findings**	
Nakaso et al. (2014) [259]	Human dopaminergic neuroblastoma (SH-SY5Y) cells	αT3 (1 µM)		Cell viability: ↑, p-Akt: ↔, p-ERK: ↔	
		βT3 (1 µM)		Cell viability: ↑, p-Akt: ↑, p-ERK: ↔	
		γT3 (1 µM)		Cell viability: ↑, p-Akt: ↑, p-ERK: ↑	
		δT3 (1 µM)		Cell viability: ↑, p-Akt: ↑, p-ERK: ↑	
**In Vivo Studies**					
**Researcher (Year)**	**Animal Model**	**Treatment & Dose**	**Mode**	**Treatment Duration**	**Findings**
Lan et al. (1997) [261]	1-methyl-4-phenyl-l,2,3,6-tetrahydropyridine (MPTP)-induced neurodegeneration in male and female weanling C57BL/6 mice (21 ± 3 days old)	Iron (25 g/kg diet) + vitamin E (7 g/kg diet)	Oral	30 days	Striatal iron concentration: ↓, GSSG: ↓, GSH: ↔, GSSG/GSH: ↓, MDA: ↓
Nakaso et al. (2016) [260]	MPTP-induced neurodegeneration in C57BL/6 mice	αTF (100 μg/kg)	Oral	4 days (day 2–6)	Wheel running activity: ↔, loss of dopaminergic neurons: ↓, time on rotarod: ↔
		δTF (100 μg/kg)			
		αT3 (100 μg/kg)			Wheel running activity: ↔, loss of dopaminergic neurons: ↓, time on rotarod: ↑
		δT3 (100 μg/kg)			Wheel running activity: ↑, loss of dopaminergic neurons: ↓, time on rotarod: ↔
**Human Studies**					
**Researcher (Year)**	**Study Design**	**Study Population**	**Data Collection**		**Findings**
Parkinson study group (1993) [262]	Multicenter, controlled clinical trial	Untreated patients with Parkinson’s disease (n = 202)	Supplementation of TF (2000 IU/day)		TF did not delay the onset of disability associated with early otherwise unwanted Parkinson’s disease (HR = 0.91, 95% CI = 0.74–1.12)
Moren et al. (1996) [263]	Nested case control study	Patients with idiopathic Parkinson’s disease (n = 84; aged 46–65 years)	Intake of dietary vitamin E		No beneficial effect of prior vitamin E intake on Parkinson’s disease occurrence (OR = 0.64, 95% CI = 0.35–1.17)
Hellenbrand et al. (1996) [264]	Case control study	Patients with Parkinson’s disease (n = 342; mean age: 56.2 ± 6.7 years)	Past dietary habits		No association between Parkinson’s disease and vitamin E intake (OR = 0.92, 95% CI = 0.46–1.83)
Scheider et al. (1997) [265]	Case control study	Male subjects with two cardinal signs of Parkinson’s disease (n = 126; aged 45–79 years)	Usual dietary intake (vitamin E) 20 years ago		No reduction in Parkinson’s disease risk was associated with higher vitamin E intakes (OR = 1.18, 95% CI = 0.47–2.98).
Anderson et al. (1999) [266]	Case control study	Newly diagnosed idiopathic Parkinson’s disease in men and women (n = 103; aged 40–89 years)	Dietary habits during adult life		Intake of food containing vitamin E was not significantly related with Parkinson’s disease risk (OR = 1.21, 95% CI = 0.69–2.11)
Johnson et al. (1999) [267]	Case control study	Patients with Parkinson’s disease (n = 126; aged ≥50 years)	Usual dietary intake		Weak positive association of Parkinson’s disease with vitamin E (OR = 1.25, 95% CI = 0.71–2.22)
Zhang et al. (2002) [268]	Cohort study	Patients with Parkinson’s disease (n = 371; aged 30–55 years)	Dietary information on how often and amount of food		The risk of Parkinson’s disease was significantly reduced with high intake of dietary vitamin E (OR = 0.69, 95% CI = 0.49–0.98)
de Rijk et al. (1997) [269]	Cross sectional study	Individual without dementia and participants with Parkinson’s disease (n = 5342; aged ≥55 years)	Intake of dietary antioxidants		High intake of dietary vitamin E protected against the occurrence of Parkinson’s disease (OR = 0.50, 95% CI = 0.20–0.90)

**Table 12 nutrients-12-00259-t012:** Effect of T3 on Alzheimer’s disease.

**In Vitro Studies**					
**Researcher (Year)**	**Cell Type**	**Treatment & Dose**	**Findings**		
Huebbe et al. (2007) [275]	Human neuroblastoma (SH-SY5Y) cells	αTF (25 µM)	Cell viability: ↑ (after treated with tBHP and BSO)		
		γTF (25 µM)			
		αT3 (25 µM)			
		γT3 (25 µM)	Cell viability: ↑ (after treated with tBHP)		
Saito et al. (2010) [276]	Primary cortical neurons isolated from cerebral cortex of Sprague-Dawley rat fetuses	αTF (0.25–2.5 µM)	Glutamate-induced cytotoxicity: ↓, cellular GSH: ↔, ROS: ↓, lipid hydroperoxide formation: ↓		
		αT3 (0.25–2.5 µM)			
Grimm et al. (2016) [277]	Human neuroblastoma (SH-SY5Y) cells & neuro 2a (N2a) cells	αTF (10 µM)	TC: ↓, free cholesterol: ↓, ROS: ↔, hydrogen peroxide-induced ROS: ↓		
		αT3 (10 µM)	TC: ↓, free cholesterol: ↓, ROS: ↓, hydrogen peroxide-induced ROS: ↓, Aβ production: ↑, Aβ degradation: ↓		
Hagl et al. (2014) [278]	PC12 (from pheochromocytoma of rat adrenal medulla) and HEK293 (from human embryonic kidney) cells	Rice bran extract (0.02–0.5 mg/mL)	ATP level: ↑, mitochondrial membrane potential: ↑, mitochondrial respiration: ↑, citrate synthase activity: ↑, peroxisome proliferator-activated receptor gamma coactivator 1-alpha (PGC1α): ↑, Mfn1: ↓, Drp1: ↑, fis1: ↑, Opa1: ↔		
Selvaraju et al. (2014) [279]	Human glioblastoma (DBTRG-05MG) cells	TRF (100, 200 and 300 ng/mL)	Astrocytes’ viability: ↑, mitochondrial membrane potential: ↑, MDA level: ↓		
		αTF (100, 200 and 300 ng/mL)			
**In Vivo Studies**					
**Researcher (Year)**	**Animal Model**	**Treatment & Dose**	**Mode**	**Treatment Duration**	**Findings**
Damanhuri et al. (2016) [280]	Double transgenic male mice B6C3-Tg (APPswe, PS1dE9)85Dbo/Mmjax with C57BL/6J genetic background	TRF (24% αTF, 27% αT3, 4% βT3, 32% γT3, 14% δT3) (200 mg/kg)	Oral	6 months	SOD: ↓, GPx: ↔, CAT: ↑, DNA damage: ↓
		αTF (200 mg/kg)			SOD: ↓, GPx: ↔, CAT: ↔, DNA damage: ↓
Hagl et al. (2013) [289]	Male Dunkin Hartley guinea pigs	Rice bran extract (50 or 150 mg/kg)	Oral	30 days	Respiration in brain mitochondria: ↑, resistance of brain cells against mitochondrial dysfunction: ↑, Drp1: ↑, fis1: ↑, Mfn1: ↔, Opa1: ↔, citrate synthase activity: ↑, cardiolipin: ↑, phosphatidylglycerol: ↑
Hagl et al. (2016) [281]	Aged (18-month-old) female NMRI (Naval Medical Research Institute) mice	Rice bran extract (86 μg/g αTF, 71 μg/g βTF, 288 μg/g γTF, 93 μg/g δTF, 55 μg/g αT3, 2226 μg/g γT3, 266 μg/g δT3) (340 mg/kg)	Oral	3 weeks	ATP concentration: ↑, basal mitochondrial membrane potential: ↑, complex I respiration: ↑, citrate synthase activity: ↑, PGC1α: ↑, complex V: ↑, Drp1: ↔, fis1: ↓, Mfn1: ↑, Opa1: ↔
Tiwari et al. (2009) [282]	STZ-induced cognitive impairment and oxidative-nitrosative stress in male Wistar rats	αTF (100 mg/kg)	Oral	20 days	Learning performance: ↑, time spent in target quadrant: ↑, retention transfer latency: ↓, spontaneous locomotor activity: ↔, AChE activity: ↓, MDA: ↓, GSH: ↑, SOD: ↑, CAT: ↑, nitrite: ↓
		T3 (mixture of αT3, βT3, γT3) (50 and 100 mg/kg)			
Schloesser et al. (2015) [283]	Male C57BL/6J mice	T3/γ-cyclodextrin complex diet (100 mg/kg diet) + αTF (20 mg/kg diet)	Oral	24 weeks	ATP concentration: ↑, mitochondrial membrane potential: ↑, transcription factor A mitochondrial (TFAM) protein: ↑, PGC1α: ↔, Sod2: ↔, Hmox1: ↔, Gclm: ↔, Gpx4: ↔, Cat: ↔, proteasomal activity: ↔, BACE1: ↔
**Human Studies**					
**Researcher (Year)**	**Study Design**	**Study Population**	**Intervention**	**Duration**	**Findings**
Mangialasche et al. (2010) [284]	Population-based cohort study	Dementia-free subjects (*n* = 232; aged 80+ years)	Plasma levels of vitamin E (αTF, βTF, γTF, δTF, αT3, βT3, γT3, δT3)	6 years follow-up	High plasma levels of vitamin E were associated with a reduced risk of Alzheimer’s disease in advanced age
Devore et al. (2010) [288]	Population-based cohort study	Dementia-free subjects (*n* = 5395; aged 55+ years)	Dietary information at study baseline	9.6 years follow-up	High intake of foods rich in vitamin E reduced long-term risk of dementia and Alzheimer’s disease.
Mangialasche et al. (2013) [286]	Longitudinal study	Patients with Alzheimer’s disease (*n* = 81; aged 75.1 ± 5.7 years) and MCI (*n* = 86; aged 74.6 ± 5.2 years)	Plasma levels of vitamin E (αTF, βTF, γTF, δTF, αT3, βT3, γT3, δT3)	1 year follow-up	Low levels of vitamin E isomers (αTF, γTF, αT3, βT3, γT3, δT3) was detected in Alzheimer’s disease patients
Mangialasche et al. (2013) [285]	Population-based cohort study	Subjects with Alzheimer’s disease (*n* = 40) and MCI (n = 24) (aged 71.5 ± 3.8 years)	Baseline serum vitamin E	8 years follow-up	High levels of TF and T3 were associated with reduced risk of cognitive impairment in older adults
Mangialasche et al. (2012) [287]	Cross-sectional study	Subjects with Alzheimer’s disease (*n* = 168; aged 77.4 ± 6.3 years) and MCI (*n* = 166; aged 75.8 ± 5.6 years)	Plasma vitamin E and markers of vitamin E damage (α-tocopherylquinone & 5-nitro-γTF)	-	Low levels of total TF, total T3 and total vitamin E in subjects with Alzheimer’s disease and MCI

**Table 13 nutrients-12-00259-t013:** Effect of T3 on wound healing.

**In Vitro Studies**					
**Researcher (Year)**	**Cell Type**	**Treatment & Dose**	**Findings**		
Xu et al. (2017) [295]	Human fibroblasts and human dermal microvascular endothelial cells	MeT3α (10 nmol)	KIF26A: ↑, TRIM54: ↑, TOR2A: ↑, COX7A: ↑, MYH14: ↑, lanosterol synthase expression: ↓, VEGFA: ↑, PDGFB: ↑, basal and reserve mitochondrial capacity: ↑		
Arffah et al. (2009) [302]	Fibroblasts from hypertrophic scar (HSc).	TRF (0 to 0.156 mg/ml)—incubated for 24, 48 and 72 h	Cell growth: ↑ (at lower concentration, 0.0025 mg/ml to 0.02 mg/ml), fibroblast cell death: ↑ (at higher concentration, 0.078 mg/ml to 0.156 mg/ml), cell growth of hypertrophic scar fibroblast: ↓		
Mahirah et al. 2006 [303]	Fibroblast and keratinocyte cultures from hypertrophic scar	TRF—incubated for 24, 48 and 72 h	Cell growth: ↑ (at lower concentration), cell growth: ↓ (at higher concentrations)		
**In Vivo Studies**					
**Researcher (Year)**	**Animal Model**	**Treatment & Dose**	**Mode**	**Treatment Duration**	**Findings**
Pierpaoli et al. (2017) [301]	BALB/c mice with excisional wounds inoculated with MRSA	Annatto T3 (12.9% γT3, 87.1% δT3) (100 mg/kg)	Oral	8 days	Bacterial load: ↓, natural killer cell cytotoxicity: ↑, fibronectin type III: ↑, IL-24: ↑
Elsy et al. (2017) [297]	Alloxan-induced diabetic albino rats with excisional wounds	d-δ-T3 (200 mg/kg)	Oral	3 weeks	Early regeneration of epidermal and dermal components: ↑, blood sugar: ↓, TC: ↓, TG: ↓, LDL: ↓, VLDL: ↓, HDL: ↑, serum total protein content: ↑, CAT: ↑, antioxidant capacity: ↑
Elsy et al. (2017) [298]	Alloxan-induced diabetic albino rats with incisional wounds	d-δ-TRF (200 mg/kg)	Oral	3 weeks	Regeneration and reorganization of epidermal and dermal components: ↑
Xu et al. (2017) [295]	Diabetic C57BL/KsJm/*Leptdb* (*db*/*db*) mice with excisional wounds	MeT3α (1 μmol)	Topical	21 days	Wound area: ↓
Musalmah et al. (2005) [299]	STZ-induced diabetic rats with excisional wounds	PVE (200 mg/kg)	Oral	10 days	Wound area: ↓, total protein content: ↑, collagen content: ↑, GPx: ↑, SOD: ↑, MDA: ↓,
Nurlaily et al. (2011) [300]	Diabetic Sprague-Dawley male rats with excisional wounds	0.5% TRF	Topical	Once daily for 10 days	Better wound contraction on 8th day post-treatment, total protein content: ↑

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
