# Peer review of "Potential Role of Tocotrienols on Non-Communicable Diseases: A Review of Current Evidence"

_nutrients, 2020, doi:10.3390/nu12010259_

Round 1

Reviewer 1 Report

Dear Editor(s), dear authors,

the review “Potential Role of Tocotrienol on Non-communicable Diseases: A Review of Current Evidence” from Wong et al. includes 382 references on in vivo and in vitro studies with tocotrienols (T3s) and mixtures thereof. The authors review current literature on the beneficial effects of tocotrienols on non-communicable diseases. With 12 tables and several figures, the manuscript comprehensively covers the effects of T3s on molecular basis.

The reviewer raises some minor questions.

Anti-inflammatory effects of T3s are mentioned in the chapters on ulcers, wound healing and CVD and muscle diseases. The reviewer suggests a more general paragraph on anti-inflammatory activity of T3s to dispose the reader for the specific chapters mentioned.

Regarding the pharmacokinetics of T3, recent literature suggests an increasing role of long chain metabolites of vitamin E on health effects. For example, the inhibitory activity of T3 long chain metabolites on 5-LOX (Pain et al. Nature Communication 2018). Also original references are missing on the first description of T3 metabolism by Birringer et al. JNutr2002.

In the chapter neurodegenerative disease, an animal study with guinea pigs is missing: Rice bran extract protects from mitochondrial dysfunction in guinea pig brains. Hagl et al. 2013 Pharmacol Res.

Style of table 7 and 12 should be consistent with other tables.

515: please comment on the results from human trials…

Author Response

Reviewer 1:

The review “Potential Role of Tocotrienol on Non-communicable Diseases: A Review of Current Evidence” from Wong et al. includes 382 references on in vivo and in vitro studies with tocotrienols (T3s) and mixtures thereof. The authors review current literature on the beneficial effects of tocotrienols on non-communicable diseases. With 12 tables and several figures, the manuscript comprehensively covers the effects of T3s on molecular basis.

The reviewer raises some minor questions.

 Comment: Anti-inflammatory effects of T3s are mentioned in the chapters on ulcers, wound healing and CVD and muscle diseases. The reviewer suggests a more general paragraph on anti-inflammatory activity of T3s to dispose the reader for the specific chapters mentioned.

Reply: Thank you for the comments. A more general paragraph on anti-inflammatory activity of T3 has been included (Page 58, Line 1211-1237)

Comment: Regarding the pharmacokinetics of T3, recent literature suggests an increasing role of long chain metabolites of vitamin E on health effects. For example, the inhibitory activity of T3 long chain metabolites on 5-LOX (Pein et al. Nature Communication 2018). Also original references are missing on the first description of T3 metabolism by Birringer et al. JNutr2002.

Reply: Thank you for the comments. Recent literature on the role of long chain metabolites of vitamin E on health effects has been included (Reference #28) (Page 3, Line 96-97). Original reference on the first description of T3 metabolism by Birringer et al. (2002) has been included (Reference #27) (Page 3, Line 93 & 96). We have added another paragraph on the anti-inflammatory properties of the metabolites of vitamin E (Page 58, Line 1214-1224).

Comment: In the chapter neurodegenerative disease, an animal study with guinea pigs is missing: Rice bran extract protects from mitochondrial dysfunction in guinea pig brains. Hagl et al. 2013 Pharmacol Res.

Reply: Thank you for the comments. Animal study with guinea pig done by Hagl et al. (2013) has been included in the chapter of neurodegenerative disease (Page 48, Table 12, Reference #289).

Comment: Style of table 7 and 12 should be consistent with other tables.

Reply: Thank you for the comments. Style of table 8 and 13 has been changed (Page 29 & 51)

Comment: 515: please comment on the results from human trials.

Reply: Thank you for the comments. The available evidence on humans has been summarized in each chapter and the overall comments on the results from human trials have been included in the Research gap and Future Perspective section (Page 60, Line 1314-1320).

Reviewer 2 Report

The review article is focused on an important topic “Potential Role of Tocotrienol on Non-communicable Diseases”. well written review article on tocotrienol.  Authors summarize the heath benefit of tocotrienol on wide range of diseases, including cardiovascular, musculoskeletal, metabolic, skin disorders and cancers. Most of the clinical trials testing the effect of Vitamin E in a wide range of major health disorders have studied alpha-tocopherol, which represent one-eight of natural vitamin E family. Providing information on studies published on tocotrienol is critical and informative.

Major and minor issues should be addressed:

Major issues:

It’s better to present effect of tocotrienol on cardiovascular, cerebrovascular and neuroprotection separately.

The review should also include detailed figure like 1 and 2 on other topics covered in article.

Please add flow chart or diagram on Tocotrienol sensitive molecular mechanisms for various diseases mentioned in this review.

Information on safety of tocotrienol and tocopherol must be provide under separate heading.

Pls add more figures like 1-4 on other specific disease mention in the article for their molecular mechanisms.  

Author Response

Reviewer 2:

Thank you for your meticulous review and your constructive comments.

The review article is focused on an important topic “Potential Role of Tocotrienol on Non-communicable Diseases”. well written review article on tocotrienol.  Authors summarize the heath benefit of tocotrienol on wide range of diseases, including cardiovascular, musculoskeletal, metabolic, skin disorders and cancers. Most of the clinical trials testing the effect of Vitamin E in a wide range of major health disorders have studied alpha-tocopherol, which represent one-eight of natural vitamin E family. Providing information on studies published on tocotrienol is critical and informative.

Comment: It’s better to present effect of tocotrienol on cardiovascular, cerebrovascular and neuroprotection separately

Reply: Thank you for the comments. The effects of tocotrienol on cardiovascular and cerebrovascular diseases have been separated (Page 21, Line 359-395)

Comment: The review should also include detailed figure like 1 and 2 on other topics covered in article.

Reply: Thank you for the comments. Detailed figures on topic covered in article have been included (Figure 1-11).

Comment: Please add flow chart or diagram on Tocotrienol sensitive molecular mechanisms for various diseases mentioned in this review.

Reply: Thank you for the comments. Diagram on tocotrienol sensitive molecular mechanisms for various diseases mentioned in this review has been included (Figure 1-11)

Comment: Information on safety of tocotrienol and tocopherol must be provided under separate heading.

Reply: Thank you for the comments. Information on safety of vitamin E has been included (Page 60, Line 1290-1307).

Comment: Pls add more figures like 1-4 on other specific disease mention in the article for their molecular mechanisms.

Reply: Thank you for the comments. Figures summarizing their molecular mechanisms of T3 in specific disease have been included (Figure 1-11)

Reviewer 3 Report

This is a very detailed review on the possible effects of tocotrienols (T3s) on non-communicable diseases (NCDs). The strengths of this article are the comprehensive, thoroughness, and tabulation of almost all the recent studies on the subject. The manuscript is generally well written. Some suggestions for the improvement of this manuscript are as follows. 

1. This review gives the impression of “descriptiveness”. Different forms of T3s were found to provide beneficial effects in animal models and studies in cell lines indicating that these compounds affect a variety of signaling molecules that supposedly contribute to the beneficial effects. Just tabulating everything that has been published would not help readers to understand what is exactly going on. The quality of the review would improve greatly if the authors pay attention to how a single chemical such as delta-tocotrienol can affect so many different signaling molecules. It is important to differentiate between the primary targets and secondary events. Are these activities related to the general antioxidant activities of T3s? This would provide a better understanding of the topic.

2. Many of the reviewed beneficial effects of T3s have been attributed to the anti-inflammatory activities. It is important to discuss the contribution of the side-chain degradation products of T3s in the anti-inflammatory action. There are interesting publications on this aspect.

3. It is well-known that, because of the bioavailability, metabolism and other physiological factors, results from in vitro studies could be quite different from animal studies, and the relevance of the information from cell line studies may be questionable. In this sense, to pay more attention to in vivo studies, especially in human studies, are important. This was not done in the studies in Section 12, Effects of tocotrienols on cancer. Quite a few recent excellent studies in animals and humans, as well as review articles, are missing.

4. The authors seem to pay great attention to studies with cell lines and describe the different observations by the different authors. The authors are encouraged to do more analysis on the structure-activity relationship – relating the biological activity to the chemical and physical properties of specific T3s. They may streamline the descriptions, since they are tabulated in the tables anyways, and pay more attention to analysis. 

5. There are some minor issues. For example: 

In the title, it should be "tocotrienols"; 's' should be added. On line 23, "vigorous studies"; how is “vigorous” defined? Page 2, lines 78-82, the distribution of T3s in fatty tissues need to be included; line 83-85, on the transport of vitamin E, please indicate that the function of alpha-TTP is to transfer vitamin E molecules from the liver to the blood. Page 15, line 294-295, "CVD... includes hypertension..."; hypertension is a risk factor for CVD and is not considered to be a disease. Line 552-553, "about 1-2% of skeletal muscle mass declined after 30 years old..."; is this per year or for a longer time? 

Author Response

Reviewer 3

Thank you for your meticulous review and your constructive comments.

This is a very detailed review on the possible effects of tocotrienols (T3s) on non-communicable diseases (NCDs). The strengths of this article are the comprehensive, thoroughness, and tabulation of almost all the recent studies on the subject. The manuscript is generally well written. Some suggestions for the improvement of this manuscript are as follows.

Comment: This review gives the impression of “descriptiveness”. Different forms of T3s were found to provide beneficial effects in animal models and studies in cell lines indicating that these compounds affect a variety of signaling molecules that supposedly contribute to the beneficial effects. Just tabulating everything that has been published would not help readers to understand what is exactly going on. The quality of the review would improve greatly if the authors pay attention to how a single chemical such as delta-tocotrienol can affect so many different signaling molecules. It is important to differentiate between the primary targets and secondary events. Are these activities related to the general antioxidant activities of T3s? This would provide a better understanding of the topic.

Reply: Thank you for the comment. We added Figure 1-11 to help the readers to understand the possible mechanisms underlying the protective effects of T3 in each of the disease discussed.

Comment: Many of the reviewed beneficial effects of T3s have been attributed to the anti-inflammatory activities. It is important to discuss the contribution of the side-chain degradation products of T3s in the anti-inflammatory action. There are interesting publications on this aspect.

Reply: We have added relevant information on the anti-inflammatory effects of vitamin E metabolites in the section “The anti-inflammatory property of tocotrienol” (Page 58, Line 1211-1237).

Comment: It is well-known that, because of the bioavailability, metabolism and other physiological factors, results from in vitro studies could be quite different from animal studies, and the relevance of the information from cell line studies may be questionable. In this sense, to pay more attention to in vivo studies, especially in human studies, are important. This was not done in the studies in Section 12, Effects of tocotrienols on cancer. Quite a few recent excellent studies in animals and humans, as well as review articles, are missing.

Reply: Thank you for the suggestion, we added an paragraph on clinical trial of T3 on cancer patients. There are numerous animal studies on the anticancer effects of T3, which has been reviewed previous. We have cited the recent review for the perusal of the readers (Page 57, Line 1195-1205). 

 Comment: The authors seem to pay great attention to studies with cell lines and describe the different observations by the different authors. The authors are encouraged to do more analysis on the structure-activity relationship – relating the biological activity to the chemical and physical properties of specific T3s. They may streamline the descriptions, since they are tabulated in the tables anyways, and pay more attention to analysis.

Reply: The structural-activity relationship of specific T3s and vitamin E metabolites has been mentioned throughout the manuscript. For example:

Line 127-128: The structure of T3 with its three double bonds is similar to farnesyl, the compound preceding the formation of squalene in cholesterol synthesis.

Line 142-143: A docking study showed that inhibition of HMGCR by T3 followed the order δ > γ > β > α.

Line 1004-1006: Molecular docking analysis also demonstrated that δT3 exhibited higher affinity and formed greater hydrogen bonding interaction with PPAR-δ and PPAR-γ which explaining its pan-PPAR agonist properties.

Line 1219-1221:  Docking studies showed that αT3 blocks the assess of arachidonic acid to the catalytic site of 12-lipoxygenase, thereby achieving its anti-inflammatory effects.

Line 1228-1229: In particular, 13’-carboxychromanol bound more strongly with COX-1 compared to 9’-carboxychromanol and exerted more potent COX-inhibitory effects.

Comment: There are some minor issues. For example:

In the title, it should be "tocotrienols"; 's' should be added. On line 23, "vigorous studies"; how is “vigorous” defined? Page 2, lines 78-82, the distribution of T3s in fatty tissues need to be included; line 83-85, on the transport of vitamin E, please indicate that the function of alpha-TTP is to transfer vitamin E molecules from the liver to the blood. Page 15, line 294-295, "CVD... includes hypertension..."; hypertension is a risk factor for CVD and is not considered to be a disease. Line 552-553, "about 1-2% of skeletal muscle mass declined after 30 years old..."; is this per year or for a longer time?

Reply:

Title: We have changed “Tocotrienol” to “Tocotrienols”.

Line 22: We delete the adjective “vigorous”.

Line 82-83: We added that “Multiple studies showed that T3 was preferably distributed in adipose tissue after supplementation”.

Line 84: We added that “… alpha-tocopherol transport protein (α-TTP), which transports vitamin E from the liver to the blood”

Line 297: We delete “hypertension” from the sentence.

Line 575: We added that “…about 1-2% of skeletal muscle mass declined annually after 30 years old…”

Round 2

Reviewer 3 Report

The authors have made many changes to improve the quality of the manuscript. The authors also indicated they have used Figures 1-11 to help the readers understand the mechanisms. However, the so-called "mechanisms" in the tables are still a tabulation of what has been observed in the experimental systems. The quality of the paper would improve if the initial biological activities could be related to the chemical and physical properties of the tocotrienols, for example, in Figure 11. How do tocotrienols increase ER stress, decrease H1F-1alpha and H1F-2alpha and decrease RAC1-wave2 signaling? There may not be a clear answer to these type of questions. However, an effort in this direction would help the progress of this research field. 

Author Response

Comment: The authors have made many changes to improve the quality of the manuscript. The authors also indicated they have used Figures 1-11 to help the readers understand the mechanisms. However, the so-called "mechanisms" in the tables are still a tabulation of what has been observed in the experimental systems. The quality of the paper would improve if the initial biological activities could be related to the chemical and physical properties of the tocotrienols, for example, in Figure 11. How do tocotrienols increase ER stress, decrease H1F-1alpha and H1F-2alpha and decrease RAC1-wave2 signaling? There may not be a clear answer to these type of questions. However, an effort in this direction would help the progress of this research field.

Reply: Thank you for the comments. We have summarised the available findings on the mechanisms of action of T3 in NCDs. The crucial information especially the upstream/initial biological activities, as well as the role of chemical/physical properties of T3 remain elusive. To date, there are limited studies on the structural-activity relationship between T3 and its biological targets. We had discussed some of the potential roles of chromanol ring and phytyl tail in antioxidant properties (line 1251-1258), anti-inflammation (line 1221-1223) and HMGCR inhibition (line 1241-1250). Besides, we have also mentioned in the text that there is limited information on the initial regulation for the T3-mediated ER stress (line 1120-1121), Rac1/WAVE inhibition (line 1181-1182) and HIF downregulation (line 1168). We also included more information on T3-mediated calcium release from the ER lumen as an early signal in ER stress initiation (line 1116-1121). Additionally, we have included more information to compare the anti-diabetic (line 1303-1306), anti-adipogenic (line 1306-1309) and anticancer properties (line 1310-1317) of each T3 isoforms. Lastly, we have also emphasized the needs for further investigation in relating the chemical and physical properties of T3 with their upstream biological activities (line 1352-1356).  Thank you.